# Stabilizing Dynamical Systems via Policy Gradient Methods

**Juan C. Perdomo**[*]
University of California, Berkeley

**Jack Umenberger**
MIT

**Max Simchowitz**
MIT

## Abstract

Stabilizing an unknown control system is one of the most fundamental problems in control systems engineering. In this paper, we provide a simple, model-free algorithm for stabilizing fully observed dynamical systems. While model-free methods have become increasingly popular in practice due to their simplicity and flexibility, stabilization via direct policy search has received surprisingly little attention. Our algorithm proceeds by solving a series of discounted LQR problems, where the discount factor is gradually increased. We prove that this method efficiently recovers a stabilizing controller for linear systems, and for smooth, nonlinear systems within a neighborhood of their equilibria. Our approach overcomes a significant limitation of prior work, namely the need for a pre-given stabilizing control policy. We empirically evaluate the effectiveness of our approach on common control benchmarks.

## 1  Introduction

Stabilizing an unknown control system is one of the most fundamental problems in control systems engineering. A wide variety of tasks - from maintaining a dynamical system around a desired equilibrium point, to tracking a reference signal (e.g a pilot's input to a plane) - can be recast in terms of stability. More generally, synthesizing an initial stabilizing controller is often a necessary first step towards solving more complex tasks, such as adaptive or robust control design [Sontag, 1999, 2009].

In this work, we consider the problem of finding a stabilizing controller for an unknown dynamical system via direct policy search methods. We introduce a simple procedure based off policy gradients which provably stabilizes a dynamical system around an equilibrium point. Our algorithm only requires access to a simulator which can return rollouts of the system under different control policies, and can efficiently stabilize both linear and smooth, nonlinear systems.

Relative to model-based approaches, model-free procedures, such as policy gradients, have two key advantages: they are conceptually simple to implement, and they are easily adaptable; that is, the same method can be applied in a wide variety of domains without much regard to the intricacies of the underlying dynamics. Due to their simplicity and flexibility, direct policy search methods have become increasingly popular amongst practitioners, especially in settings with complex, nonlinear dynamics which may be challenging to model. In particular, they have served as the main workhorse for recent breakthroughs in reinforcement learning and control [Silver et al., 2016, Mnih et al., 2015, Andrychowicz et al., 2020].

Despite their popularity amongst practitioners, model-free approaches for continuous control have only recently started to receive attention from the theory community [Fazel et al., 2018, Kakade et al., 2020, Tu and Recht, 2019]. While these analyses have begun to map out the computational and statistical tradeoffs that emerge in choosing between model-based and model-free approaches,

---

[*]Correspondence to jcperdomo@berkeley.edu

35th Conference on Neural Information Processing Systems (NeurIPS 2021).

they all share a common assumption: that the unknown dynamical system in question is stable, or that an initial stabilizing controller is known. As such, they do not address the perhaps more basic question, *how do we arrive at a stabilizing controller in the first place?*

## 1.1 Contributions

We establish a reduction from stabilizing an unknown dynamical system to solving a series of discounted, infinite-horizon LQR problems via policy gradients, for which no knowledge of an initial stable controller is needed. Our approach, which we call *discount annealing*, gradually increases the discount factor and yields a control policy which is near optimal for the undiscounted LQR objective. To the best of our knowledge, our algorithm is the first model-free procedure shown to provably stabilize unknown dynamical systems, thereby solving an open problem from Fazel et al. [2018].

We begin by studying linear, time-invariant dynamical systems with full state observation and assume access to *inexact* cost and gradient evaluations of the discounted, infinite-horizon LQR cost of a state-feedback controller $K$. Previous analyses (e.g., [Fazel et al., 2018]) establish how such evaluations can be implemented with access to (finitely many, finite horizon) trajectories sampled from a simulator. We show that our method recovers the controller $K_\star$ which is the optimal solution of the *undiscounted* LQR problem in a bounded number of iterations, up to optimization and simulator error. The stability of the resulting $K_\star$ is guaranteed by known stability margin results for LQR. In short, we prove the following guarantee:

**Theorem 1 (informal).** *For linear systems, discount annealing returns a stabilizing state-feedback controller which is also near-optimal for the LQR problem. It uses at most polynomially many, $\varepsilon$-inexact gradient and cost evaluations, where the tolerance $\varepsilon$ also depends polynomially on the relevant problem parameters.*

Since both the number of queries and error tolerance are polynomial, discount annealing can be efficiently implemented using at most polynomially many samples from a simulator.

Furthermore, our results extend to smooth, *nonlinear* dynamical systems. Given access to a simulator that can return damped system rollouts, we show that our algorithm finds a controller that attains near-optimal LQR cost for the Jacobian linearization of the nonlinear dynamics at the equilibrium. We then show that this controller stabilizes the nonlinear system within a neighborhood of its equilibrium.

**Theorem 2 (informal).** *Discount annealing returns a state-feedback controller which is exponentially stabilizing for smooth, nonlinear systems within a neighborhood of their equilibrium, using again only polynomially many samples drawn from a simulator.*

In each case, the algorithm returns a near optimal solution $K$ to the relevant LQR problem (or local approximation thereof). Hence, the stability properties of $K$ are, in theory, no better than those of the optimal LQR controller $K_\star$. Importantly, the latter may have worse stability guarantees than the optimal solution of a corresponding robust control objective (e.g. $\mathcal{H}_\infty$ synthesis). Nevertheless, we focus on the LQR subroutine in the interest of simplicity, clarity, and in order to leverage prior analyses of model-free methods for LQR. Extending our procedure to robust-control objectives is an exciting direction for future work.

Lastly, while our theoretical analysis only guarantees that the resulting controller will be stabilizing within a small neighborhood of the equilibrium, our simulations on nonlinear systems, such as the nonlinear cartpole, illustrate that discount annealing produces controllers that are competitive with established robust control procedures, such as $\mathcal{H}_\infty$ synthesis, without requiring any knowledge of the underlying dynamics.

## 1.2 Related work

Given its central importance to the field, stabilization of unknown and uncertain dynamical systems has received extensive attention within the controls literature. We review some of the relevant literature and point the reader towards classical texts for a more comprehensive treatment [Sontag, 2009, Sastry and Bodson, 2011, Zhou et al., 1996, Callier and Desoer, 2012, Zhou and Doyle, 1998].

**Model-based approaches.** Model-based methods construct approximate system models in order to synthesize stabilizing control policies. Traditional analyses consider stabilization of both linear and

nonlinear dynamical systems in the asymptotic limit of sufficient data [Sastry and Bodson, 2011, Sastry, 2013]. More recent, non-asymptotic studies have focused almost entirely on *linear* systems, where the controller is generated using data from multiple independent trajectories [Fiechter, 1997, Dean et al., 2019, Faradonbeh et al., 2018a, 2019]. Assuming the model is known, stabilizing policies may also be synthesized via convex optimization [Prajna et al., 2004] by combining a 'dual Lyapunov theorem' [Rantzer, 2001] with sum-of-squares programming [Parrilo, 2003]. Relative to these analysis our focus is on strengthening the theoretical foundations of model-free procedures and establishing rigorous guarantees that policy gradient methods can also be used to generate stabilizing controllers.

**Online control.** Online control studies the problem of adaptively *fine-tuning* the performance of an already-stabilizing control policy on a *single* trajectory [Dean et al., 2018, Faradonbeh et al., 2018b, Cohen et al., 2019, Mania et al., 2019, Simchowitz and Foster, 2020, Hazan et al., 2020, Simchowitz et al., 2020, Kakade et al., 2020]. Though early papers in this direction consider systems without pre-given stabilizing controllers [Abbasi-Yadkori and Szepesvári, 2011], their guarantees degrade exponentially in the system dimension (a penalty ultimately shown to be unavoidable by Chen and Hazan [2021]). Rather than fine-tuning an already stabilizing controller, we focus on the more basic problem of finding a controller which is stabilizing in the first place, and allow for the use of multiple independent trajectories.

**Model-free approaches.** Model-free approaches eschew trying to approximate the underlying dynamics and instead directly search over the space of control policies. The landmark paper of Fazel et al. [2018] proves that, despite the non-convexity of the problem, direct policy search on the infinite-horizon LQR objective efficiently converges to the globally optimal policy, assuming the search is initialized at an already stabilizing controller. Fazel et al. [2018] pose the synthesis of this initial stabilizing controller via policy gradients as an open problem; one that we solve in this work.

Following this result, there have been a large number of works studying policy gradients procedures in continuous control, see for example Feng and Lavaei [2020], Malik et al. [2019], Mohammadi et al. [2020, 2021], Zhang et al. [2021] just to name a few. Relative to our analysis, these papers consider questions of policy finite-tuning, derivative-free methods, and robust (or distributed) control which are important, yet somewhat orthogonal to the stabilization question considered herein. The recent analysis by Lamperski [2020] is perhaps the most closely related piece of prior work. It proposes a model-free, off-policy algorithm for computing a stabilizing controller for deterministic LQR systems. Much like discount annealing, the algorithm also works by alternating between policy optimization (in their case by a closed-form policy improvement step based on the Riccati update) and increasing a damping factor. However, whereas we provide precise finite-time convergence guarantees to a stabilizing controller for both linear and nonlinear systems, the guarantees in Lamperski [2020] are entirely asymptotic and restricted to linear systems. Furthermore, we pay special attention to quantifying the various error tolerances in the gradient and cost queries to ensure that the algorithm can be efficiently implemented in finite samples.

## 1.3 Background on stability of dynamical systems

Before introducing our results, we first review some of the basic concepts and definitions regarding stability of dynamical systems. In this paper, we study discrete-time, noiseless, time-invariant dynamical systems with states $\mathbf{x}_t \in \mathbb{R}^{d_x}$ and control inputs $\mathbf{u}_t \in \mathbb{R}^{d_u}$. In particular, given an initial state $\mathbf{x}_0$, the dynamics evolves according to $\mathbf{x}_{t+1} = G(\mathbf{x}_t, \mathbf{u}_t)$ where $G : \mathbb{R}^{d_x} \times \mathbb{R}^{d_u} \to \mathbb{R}^{d_x}$ is a state transition map. An equilibrium point of a dynamical system is a state $\mathbf{x}_\star \in \mathbb{R}^{d_x}$ such that $G(\mathbf{x}_\star, 0) = \mathbf{x}_\star$. As per convention, we assume that the origin $\mathbf{x}_\star = 0$ is the desired equilibrium point around which we wish to stabilize the system.

This paper restricts its attention to static state-feedback policies of the form $\mathbf{u}_t = K\mathbf{x}_t$ for a fixed matrix $K \in \mathbb{R}^{d_u \times d_x}$. Abusing notation slightly, we conflate the matrix $K$ with its induced policy. Our aim is to find a policy $K$ which is *exponentially stabilizing* around the equilbrium point.

Time-invariant, linear systems, where $G(\mathbf{x}, \mathbf{u}) = A\mathbf{x} + B\mathbf{u}$ are stabilizable if and only if there exists a $K$ such that $A+BK$ is a stable matrix [Callier and Desoer, 2012]. That is if $\rho(A+BK) < 1$, where $\rho(X)$ denotes the spectral radius, or the largest eigenvalue magnitude, of a matrix $X$. For general nonlinear systems, our goal is to find controllers which satisfy the following general, quantitative

definition of exponential stability (e.g Chapter 5.2 in Sastry [2013]). Throughout, $\|\cdot\|$ denotes the Euclidean norm.

**Definition 1.1.** A controller $K$ is $(m, \alpha)$-*exponentially stable* for dynamics $G$ if there exist constants $m, \alpha > 0$ such that if inputs are chosen according to $\mathbf{u}_t = K\mathbf{x}_t$, the sequence of states $\mathbf{x}_{t+1} = G(\mathbf{x}_t, \mathbf{u}_t)$ satisfy

$$\|\mathbf{x}_t\| \leq m \cdot \exp(-\alpha \cdot t)\|\mathbf{x}_0\|. \tag{1.1}$$

Likewise, $K$ is $(m, \alpha)$-*exponentially stable on radius* $r > 0$ if (1.1) holds for all $\mathbf{x}_0$ such that $\|\mathbf{x}_0\| \leq r$.

For linear systems, a controller $K$ is stabilizing if and only if it is stable over the entire state space, however, the restriction to stabilization over a particular radius is in general needed for nonlinear systems. Our approach for stabilizing nonlinear systems relies on analyzing their *Jacobian linearization* about the origin equilibrium. Given a continuously differentiable transition operator $G$, the local dynamics can be approximated by the Jacobian linearization $(A_{\mathrm{jac}}, B_{\mathrm{jac}})$ of $G$ about the zero equilibrium; that is

$$A_{\mathrm{jac}} := \nabla_{\mathbf{x}} G(\mathbf{x}, \mathbf{u})\big|_{(\mathbf{x}, \mathbf{u}) = (0,0)}, \quad B_{\mathrm{jac}} := \nabla_{\mathbf{u}} G(\mathbf{x}, \mathbf{u})\big|_{(\mathbf{x}, \mathbf{u}) = (0,0)}. \tag{1.2}$$

In particular, for $\mathbf{x}$ and $\mathbf{u}$ sufficiently small, $G(\mathbf{x}, \mathbf{u}) = A_{\mathrm{jac}}\mathbf{x} + B_{\mathrm{jac}}\mathbf{u} + f_{\mathrm{nl}}(\mathbf{x}, \mathbf{u})$, where $f_{\mathrm{nl}}(\mathbf{x}, \mathbf{u})$ is a nonlinear remainder from the Taylor expansion of $G$. To ensure stabilization via state-feedback is feasible, we assume throughout our presentation that the linearized dynamics $(A_{\mathrm{jac}}, B_{\mathrm{jac}})$ are stabilizable.

## 2 Stabilizing Linear Dynamical Systems

We now present our main results establishing how our algorithm, discount annealing, provably stabilizes linear dynamical systems via a reduction to direct policy search methods. We begin with the following preliminaries on the Linear Quadratic Regulator (LQR).

**Definition 2.1** (LQR Objective). For a given starting state $\mathbf{x}$, we define the LQR problem $J_{\mathrm{lin}}$ with discount factor $\gamma \in (0, 1]$, dynamic matrices $(A, B)$, and state feedback controller $K$ as,

$$J_{\mathrm{lin}}(K \mid \mathbf{x}, \gamma, A, B) := \sum_{t=0}^{\infty} \gamma^t \left(\mathbf{x}_t^\top Q \mathbf{x}_t + \mathbf{u}_t^\top R \mathbf{u}_t\right) \text{ s.t. } \mathbf{u}_t = K\mathbf{x}_t, \; \mathbf{x}_{t+1} = A\mathbf{x}_t + B\mathbf{u}_t, \; \mathbf{x}_0 = \mathbf{x}.$$

Here, $\mathbf{x}_t \in \mathbb{R}^{d_x}$, $\mathbf{u}_t \in \mathbb{R}^{d_u}$, and $Q, R$ are positive definite matrices. Slightly overloading notation, we define

$$J_{\mathrm{lin}}(K \mid \gamma, A, B) := \mathop{\mathbb{E}}_{\mathbf{x}_0 \sim \sqrt{d_x} \cdot \mathcal{S}^{d_x - 1}} [J_{\mathrm{lin}}(K \mid \mathbf{x}, \gamma, A, B)],$$

to be the same as the problem above, but where the initial state is now drawn from the uniform distribution over the sphere in $\mathbb{R}^{d_x}$ of radius $\sqrt{d_x}$.[2]

To simplify our presentation, we adopt the shorthand $J_{\mathrm{lin}}(K \mid \gamma) := J_{\mathrm{lin}}(K \mid \gamma, A, B)$ in cases where the system dynamics $(A, B)$ are understood from context. Furthermore, we assume that $(A, B)$ is stabilizable and that $\lambda_{\min}(Q), \lambda_{\min}(R) \geq 1$. It is a well-known fact that $K_{\star, \gamma} := \arg\min_K J_{\mathrm{lin}}(K \mid \gamma, A, B)$ achieves the minimum LQR cost over all possible control laws. We begin our analysis with the observation that the discounted LQR problem is equivalent to the undiscounted LQR problem with damped dynamics matrices.[3]

**Lemma 2.1.** *For all controllers $K$ such that $J_{\mathrm{lin}}(K \mid \gamma, A, B) < \infty$,*

$$J_{\mathrm{lin}}(K \mid \gamma, A, B) = J_{\mathrm{lin}}(K \mid 1, \sqrt{\gamma} \cdot A, \sqrt{\gamma} \cdot B).$$

From this equivalence, it follows from basic facts about LQR that a controller $K$ satisfies $J_{\mathrm{lin}}(0 \mid \gamma, A, B) < \infty$ if and only if $\sqrt{\gamma}(A + BK)$ is stable. Consequently, for $\gamma < \rho(A)^{-2}$, the zero

---

[2]This scaling is chosen so that the initial state distribution has identity covariance, and yields cost equivalent to $\mathbf{x}_0 \sim \mathcal{N}(0, I)$.

[3]This lemma is folklore within the controls community, see e.g. Lamperski [2020].



<p style="text-align:center">Discount Annealing</p>

**Initialize:** Objective $J(\cdot \mid \cdot), \gamma_0 \in (0, \rho(A)^{-2}), \; K_0 \leftarrow 0, \; \text{and } Q \leftarrow I, R \leftarrow I$

**For** $t = 0, 1, \ldots$

1. If $\gamma_t = 1$, run policy gradients once more as in Step 2, break, and return the resulting $K'$.

2. Using policy gradients (see Eq. (2.1)) initialized at $K_t$, find $K'$ such that:
$$J_{\lin}(K' \mid \gamma_t) - \min_K J_{\lin}(K \mid \gamma_t) \; \leq \; d_x. \tag{2.2}$$

3. Update initial controller $K_{t+1} \leftarrow K'$.

4. Using binary or random search, find a discount factor $\gamma' \in [\gamma_t, 1]$ such that
$$2.5J(K_{t+1} \mid \gamma_t) \; \leq \; J(K_{t+1} \mid \gamma') \; \leq \; 8J(K_{t+1} \mid \gamma_t). \tag{2.3}$$

5. Update the discount factor $\gamma_{t+1} \leftarrow \gamma'$.



Figure 1: Discount annealing algorithm. The procedure is identical for both linear and nonlinear systems. For linear, we initialize $J = J_{\lin}(\cdot \mid \gamma_0)$ and for nonlinear $J = J_{\nl}(\cdot \mid \gamma_0, r_\star)$ where $r_\star$ is chosen as in Theorem 2. See Theorem 1, Theorem 2, and Appendix C for details regarding policy gradients and binary (or random) search. The constants above are chosen for convenience, any constants $c_1, c_2$ such that $1 < c_1 < c_2$ suffice.

controller is stabilizing and one can solve the discounted LQR problem via direct policy search initialized at $K = 0$ [Fazel et al., 2018]. At this point, one may wonder whether the solution to this highly discounted problem yields a controller which stabilizes the undiscounted system. If this were true, running policy gradients (defined in Eq. (2.1)) to convergence, on a single discounted LQR problem, would suffice to find a stabilizing controller.

$$K_{t+1} = K_t - \eta \nabla_K J_{\lin}(K_t \mid \gamma), \quad K_0 = 0, \quad \eta > 0 \tag{2.1}$$

Unfortunately, the following proposition shows that this is not the case.

**Proposition 2.2** (Impossibility of Reward Shaping). *Fix $A = \diag(0, 2)$. For any positive definite cost matrices $Q, R$ and discount factor $\gamma$ such that $\sqrt{\gamma}A$ is stable, there exists a matrix $B$ such that $(A, B)$ is controllable (and thus stabilizable), yet the optimal controller $K_{\star,\gamma} := \arg\min_K J(K \mid \gamma, A, B)$ on the discounted problem is such that $A + BK_{\star,\gamma}$ is unstable.*

We now describe the discount annealing procedure for linear systems (Figure 1), which provably recovers a stabilizing controller $K$. For simplicity, we present the algorithm assuming access to noisy, bounded cost and gradient evaluations which satisfy the following definition. Employing standard arguments from [Fazel et al., 2018, Flaxman et al., 2005], we illustrate how these evaluations can be efficiently implemented using polynomially many samples drawn from a simulator in Appendix C.

**Definition 2.2** (Gradient and Cost Queries). *Given an error parameter $\varepsilon > 0$ and a function $J : \mathbb{R}^d \to \mathbb{R}$, $\varepsilon$-$\mathtt{Grad}(J, \mathbf{z})$ returns a vector $\widetilde{\nabla}$ such that $\|\widetilde{\nabla} - \nabla J(\mathbf{z})\|_F \leq \varepsilon$. Similarly, $\varepsilon$-$\mathtt{Eval}(J, \mathbf{z}, c)$ returns a scalar $v$ such that $|v - \min\{J(\mathbf{z}), c\}| \leq \varepsilon$.*

The procedure leverages the equivalence (Lemma 2.1) between discounted costs and damped dynamics for LQR, and the consequence that the zero controller is stabilizing if we choose $\gamma_0$ sufficiently small. Hence, for this discount factor, we may apply policy gradients initialized at the zero controller in order to recover a controller $K_1$ which is near-optimal for the $\gamma_0$ discounted objective.

Our key insight is that, due to known stability margins for LQR controllers, $K_1$ is stabilizing for the $\gamma_1$ discounted dynamics for some discount factor $\gamma_1 > (1 + c)\gamma_0$, where $c$ is a small constant that has a uniform lower bound. Therefore, $K_1$ has finite cost on the $\gamma_1$ discounted problem, so that we may again use policy gradients initialized at $K_1$ to compute a near-optimal controller $K_2$ for this larger discount factor. By iterating, we have that $\gamma_t \geq (1 + c)^t \gamma_0$ and can increase the discount factor up to 1, yielding a near-optimal stabilizing controller for the undiscounted LQR objective.

The rate at which we can increase the discount factors $\gamma_t$ depends on certain properties of the (unknown) dynamical system. Therefore, we opt for binary search to compute the desired $\gamma$ in the absence of system knowledge. This yields the following guarantee, which we state in terms of properties of the matrix $P_\star$, the optimal value function for the undiscounted LQR problem, which satisfies $\min_K J_{\text{lin}}(K \mid 1) = \text{tr}\,[P_\star]$ (see Appendix A for further details).

**Theorem 1** (Linear Systems). *Let $M_{\text{lin}} := \max\{16\text{tr}\,[P_\star]\,, J_{\text{lin}}(K_0 \mid \gamma_0)\}$. The following statements are true regarding the discount annealing algorithm when run on linear dynamical systems:*

    *a) Discount annealing returns a controller $\widehat{K}$ which is $(\sqrt{2\text{tr}[P_\star]}, (4\text{tr}\,[P_\star])^{-1})$-exponentially stable.*

    *b) If $\gamma_0 < 1$, the algorithm is guaranteed to halt whenever $t$ is greater than $64\text{tr}\,[P_\star]^4 \log(1/\gamma_0)$.*

*Furthermore, at each iteration $t$:*

    *c) Policy gradients as defined in Eq. (2.1) achieves the guarantee in Eq. (2.2) using only $\text{poly}(M_{\text{lin}}, \|A\|_{\text{op}}, \|B\|_{\text{op}})$ many queries to $\varepsilon$-$\texttt{Grad}(\cdot, J_{\text{lin}}(\cdot \mid \gamma))$ as long as $\varepsilon$ is less than $\text{poly}(M_{\text{lin}}^{-1}, \|A\|_{\text{op}}^{-1}, \|B\|_{\text{op}}^{-1})$.*

    *c) The noisy binary search algorithm (see Figure 2) returns a discount factor $\gamma'$ satisfying Eq. (2.3) using at most $\lceil 4\log(\text{tr}\,[P_\star])\rceil + 10$ many queries to $\varepsilon$-$\texttt{Eval}(\cdot, J_{\text{lin}}(\cdot \mid \gamma))$ for $\varepsilon = .1d_x$.*

We remark that since $\varepsilon$ need only be polynomially small in the relevant problem parameters, each call to $\varepsilon$-$\texttt{Grad}$ and $\varepsilon$-$\texttt{Eval}$ can be carried out using only polynomially many samples from a simulator which returns finite horizon system trajectories under various control policies. We make this claim formal in Appendix C.

*Proof.* We prove part $b)$ of the theorem and defer the proofs of the remaining parts of to Appendix A. Define $P_{K,\gamma}$ to be the solution to the discrete-time Lyapunov equation. That is for $\sqrt{\gamma}(A + BK)$ stable, $P_{K,\gamma}$ solves:

$$P_{K,\gamma} := Q + K^\top RK + \gamma(A + BK)^\top P_{K,\gamma}(A + BK). \tag{2.4}$$

Using this notation, $P_\star = P_{K_\star,1}$ is the solution to the above Lyapunov equation with $\gamma = 1$. The key step of the proof is Proposition A.4, which uses Lyapunov theory to verify the following: given the current discount factor $\gamma_t$, an idealized discount factor $\gamma'_{t+1}$ defined by

$$\gamma'_{t+1} := \left( \frac{1}{8\|P_{K_{t+1},\gamma_t}\|_{\text{op}}^4} + 1 \right)^2 \gamma_t,$$

satisfies $J_{\text{lin}}(K_{t+1} \mid \gamma'_{t+1}) = \text{tr}[P_{K_{t+1},\gamma'_{t+1}}] \leq 2\text{tr}\left[P_{K_{t+1},\gamma_t}\right] = 2J_{\text{lin}}(K_{t+1} \mid \gamma_t)$. Since the control cost is non-decreasing in $\gamma$, the binary search update in Step 4 ensures that the actual $\gamma_{t+1}$ also satisfies

$$\gamma_{t+1} \geq \left( \frac{1}{8\|P_{K_{t+1},\gamma_t}\|_{\text{op}}^4} + 1 \right)^2 \gamma_t \geq \left( \frac{1}{128\text{tr}\,[P_\star]^4} + 1 \right)^2 \gamma_t,$$

The following calculation (which uses $d_x \leq \text{tr}\,[P_\star]$ for $\lambda_{\min}(Q) \geq 1$) justifies the second inequality above:

$$\|P_{K_{t+1},\gamma_t}\|_{\text{op}}^4 \leq \text{tr}\left[P_{K_{t+1},\gamma_t}\right]^4 = J_{\text{lin}}(K_{t+1} \mid \gamma_t)^4 \leq (\min_K J_{\text{lin}}(K \mid \gamma_t) + d_x)^4 \leq 16\text{tr}\,[P_\star]^4.$$

Therefore, $\gamma_t \geq (1/(128\text{tr}\,[P_\star]^4) + 1)^{2t}\gamma_0$. The precise bound follows from taking logs of both sides and using the numerical inequality $\log(1 + x) \leq x$ to simplify the denominator.

                                                                         $\square$

# 3 Stabilizing Nonlinear Dynamical Systems

We now extend the guarantees of the discount annealing algorithm to smooth, nonlinear systems. Whereas our study of linear systems explicitly leveraged the equivalence of discounted costs and damped dynamics, our analysis for nonlinear systems *requires* access to system rollouts under damped dynamics, since the previous equivalence between discounting and damping breaks down in nonlinear settings.

More specifically, in this section, we assume access to a simulator which given a controller $K$, returns trajectories generated according to $\mathbf{x}_{t+1} = \sqrt{\gamma} G_{\mathrm{nl}}(\mathbf{x}_t, K\mathbf{x}_t)$ for any damping factor $\gamma \in (0, 1]$, where $G_{\mathrm{nl}}$ is the transition operator for the nonlinear system. While such trajectories may be infeasible to generate on a physical system, we believe these are reasonable to consider when dynamics are represented using software simulators, as is often the case in practice [Lewis et al., 2003, Peng et al., 2018].

The discount annealing algorithm for nonlinear systems is almost identical to the algorithm for linear systems. It again works by repeatedly solving a series of quadratic cost objectives on the nonlinear dynamics as defined below, and progressively increasing the damping factor $\gamma$.

**Definition 3.1** (Nonlinear Objective). For a state feedback controller $K : \mathbb{R}^{d_x} \to \mathbb{R}^{d_u}$, damping factor $\gamma \in (0, 1]$, and an initial state $\mathbf{x}$, we define:

$$J_{\mathrm{nl}}(K \mid \mathbf{x}, \gamma) := \sum_{t=0}^{\infty} \mathbf{x}_t^\top Q \mathbf{x}_t + \mathbf{u}_t^\top R \mathbf{u}_t \tag{3.1}$$

$$\text{s.t } \mathbf{u}_t = K\mathbf{x}_t, \quad \mathbf{x}_{t+1} = \sqrt{\gamma} \cdot G_{\mathrm{nl}}(\mathbf{x}_t, \mathbf{u}_t), \quad \mathbf{x}_0 = \mathbf{x}. \tag{3.2}$$

Overloading notation as before, we let $J_{\mathrm{nl}}(K \mid \gamma, r) := \mathbb{E}_{\mathbf{x} \sim r \cdot \mathcal{S}^{d_x-1}} [J_{\mathrm{nl}}(K \mid \gamma, \mathbf{x})] \times \frac{d_x}{r^2}$.

The normalization by $d_x/r^2$ above is chosen so that the nonlinear objective coincides with the LQR objective when $G_{\mathrm{nl}}$ is in fact linear. Relative to the linear case, the only algorithmic difference for nonlinear systems is that we introduce an extra parameter $r$ which determines the radius for the initial state distribution. As established in Theorem 2, this parameter must be chosen small enough to ensure that discount annealing succeeds. Our analysis pertains to dynamics which satisfy the following smoothness definition.

**Assumption 1** (Local Smoothness). The transition map $G_{\mathrm{nl}}$ is continuously differentiable. Furthermore, there exist $r_{\mathrm{nl}}, \beta_{\mathrm{nl}} > 0$ such that for all $(\mathbf{x}, \mathbf{u}) \in \mathbb{R}^{d_x+d_u}$ with $\|\mathbf{x}\| + \|\mathbf{u}\| \leq r_{\mathrm{nl}}$,

$$\|\nabla_{\mathbf{x},\mathbf{u}} G_{\mathrm{nl}}(\mathbf{x}, \mathbf{u}) - \nabla_{\mathbf{x},\mathbf{u}} G_{\mathrm{nl}}(0, 0)\|_{\mathrm{op}} \leq \beta_{\mathrm{nl}}(\|\mathbf{x}\| + \|\mathbf{u}\|).$$

For simplicity, we assume $\beta_{\mathrm{nl}} \geq 1$ and $r_{\mathrm{nl}} \leq 1$. Using Assumption 1, we can apply Taylor's theorem to rewrite $G_{\mathrm{nl}}$ as its Jacobian linearization around the equilibrium point, plus a nonlinear remainder term.

**Lemma 3.1.** *If $G_{\mathrm{nl}}$ satisfies Assumption 1, then all $\mathbf{x}, \mathbf{u}$ for which $\|\mathbf{x}\| + \|\mathbf{u}\| \leq r_{\mathrm{nl}}$,*

$$G_{\mathrm{nl}}(\mathbf{x}, \mathbf{u}) = A_{\mathrm{jac}} \mathbf{x} + B_{\mathrm{jac}} \mathbf{u} + f_{\mathrm{nl}}(\mathbf{x}, \mathbf{u}), \tag{3.3}$$

*where $\|f_{\mathrm{nl}}(\mathbf{x}, \mathbf{u})\| \leq \beta_{\mathrm{nl}}(\|\mathbf{x}\|^2 + \|\mathbf{u}\|^2)$, $\|\nabla f_{\mathrm{nl}}(\mathbf{x}, \mathbf{u})\| \leq \beta_{\mathrm{nl}}(\|\mathbf{x}\| + \|\mathbf{u}\|)$, and where $(A_{\mathrm{jac}}, B_{\mathrm{jac}})$ are the system's Jacobian linearization matrices defined in Eq. (1.2).*

Rather than trying to directly understand the behavior of stabilization procedures on the nonlinear system, the key insight of our nonlinear analysis is that we can reason about the performance of a state-feedback controller on the nonlinear system via its behavior on the system's Jacobian linearization. In particular, the following lemma establishes how any controller which achieves finite discounted LQR cost for the Jacobian linearization is guaranteed to be exponentially stabilizing on the damped nonlinear system for initial states that are small enough. Throughout the remainder of this section, we define $J_{\mathrm{lin}}(\cdot \mid \gamma) := J_{\mathrm{lin}}(\cdot \mid \gamma, A_{\mathrm{jac}}, B_{\mathrm{jac}})$ as the LQR objective from Definition 2.1 where $(A, B) = (A_{\mathrm{jac}}, B_{\mathrm{jac}})$.

**Lemma 3.2** (Restatement of Lemma B.2). *Suppose that $\mathcal{C}_J = J_{\mathrm{lin}}(K \mid \gamma) < \infty$, then $K$ is $(\mathcal{C}_J^{1/2}, (4\mathcal{C}_J)^{-1})$ exponentially stable on the damped system $\sqrt{\gamma} G_{\mathrm{nl}}$ over radius $r = r_{\mathrm{nl}} / (\beta_{\mathrm{nl}} \mathcal{C}_J^{3/2})$.*

The second main building block of our nonlinear analysis is the observation that if the dynamics are locally smooth around the equilibrium point, then by Lemma 3.1, decreasing the radius $r$ of the initial state distribution $\mathbf{x}_0 \sim r \cdot \mathcal{S}^{d_x-1}$ reduces the magnitude of the nonlinear remainder term $f_{\mathrm{nl}}$. Hence, the nonlinear system smoothly approximates its Jacobian linearization. More precisely, we establish that the difference in gradients and costs between $J_{\mathrm{nl}}(K \mid \gamma, r)$ and $J_{\mathrm{lin}}(K \mid \gamma)$ decrease linearly with the radius $r$.

**Proposition 3.3.** *Assume $J_{\mathrm{lin}}(K \mid \gamma) < \infty$. Then, for $P_{K,\gamma}$ defined as in Eq. (2.4):*

a) *If $r \leq \frac{r_{\mathrm{nl}}}{2\beta_{\mathrm{nl}}\|P_{K,\gamma}\|_{\mathrm{op}}^2}$, then $\left| J_{\mathrm{nl}}(K \mid \gamma, r) - J_{\mathrm{lin}}(K \mid \gamma) \right| \leq 8d_x\beta_{\mathrm{nl}}\|P_{K,\gamma}\|_{\mathrm{op}}^4 \cdot r.$*

b) *If $r \leq \frac{1}{12\beta_{\mathrm{nl}}\|P_{K,\gamma}\|_{\mathrm{op}}^{5/2}}$, then, $\|\nabla_K J_{\mathrm{nl}}(K \mid \gamma, r) - \nabla_K J_{\mathrm{lin}}(K \mid \gamma)\|_{\mathrm{F}} \leq 48d_x\beta_{\mathrm{nl}}(1 + \|B\|_{\mathrm{op}})\|P_{K,\gamma}\|_{\mathrm{op}}^7 \cdot r$*

Lastly, because policy gradients on linear dynamical systems is robust to inexact gradient queries, we show that for $r$ sufficiently small, running policy gradients on $J_{\mathrm{nl}}$ converges to a controller which has performance close to the optimal controller for the LQR problem with dynamic matrices $(A_{\mathrm{jac}}, B_{\mathrm{jac}})$. As noted previously, we can then use Lemma 3.2 to translate the performance of the optimal LQR controller for the Jacobian linearization to an exponential stability guarantee for the nonlinear dynamics. Using these insights, we establish the following theorem regarding discount annealing for nonlinear dynamics.

**Theorem 2** (Nonlinear Systems). *Let $M_{\mathrm{nl}} := \max\{21\mathrm{tr}\left[P_\star\right], J_{\mathrm{lin}}(K_0 \mid \gamma_0)\}$. The following statements are true regarding the discount annealing algorithm for nonlinear dynamical systems when $r_\star$ is less than a fixed quantity that is $\mathrm{poly}(1/M_{\mathrm{nl}}, 1/\|A\|_{\mathrm{op}}, 1/\|B\|_{\mathrm{op}}, r_{\mathrm{nl}}/\beta_{\mathrm{nl}})$*

a) *Discount annealing returns a controller $\widehat{K}$ which is $(\sqrt{2\mathrm{tr}[P_\star]}, (8\mathrm{tr}\left[P_\star\right])^{-1})$-exponentially stable over a radius $r = r_{\mathrm{nl}} / (8\beta_{\mathrm{nl}}\mathrm{tr}\left[P_\star\right]^2)$*

b) *If $\gamma_0 < 1$, the algorithm is guaranteed to halt whenever $t$ is greater than $64\mathrm{tr}\left[P_\star\right]^4 \log(1/\gamma_0)$.*

*Furthermore, at each iteration $t$:*

c) *Policy gradients achieves the guarantee in Eq. (2.2) using only $\mathrm{poly}(M_{\mathrm{nl}}, \|A\|_{\mathrm{op}}, \|B\|_{\mathrm{op}})$ many queries to $\varepsilon\text{-}\mathtt{Grad}(\cdot, J_{\mathrm{nl}}(\cdot \mid \gamma))$ as long as $\varepsilon$ is less than some fixed polynomial $\mathrm{poly}(M_{\mathrm{nl}}^{-1}, \|A\|_{\mathrm{op}}^{-1}, \|B\|_{\mathrm{op}}^{-1})$.*

c) *Let $c_0$ denote a universal constant. With probability $1 - \delta$, the noisy random search algorithm (see Figure 2) returns a discount factor $\gamma'$ satisfying Eq. (2.3) using at most $c_0 \cdot \mathrm{tr}\left[P_\star\right]^4 \log(1/\delta)$ queries to $\varepsilon\text{-}\mathtt{Eval}(\cdot, J_{\mathrm{nl}}(\cdot \mid \gamma, r_\star))$ for $\varepsilon = .01d_x$.*

We note that while our theorem only guarantees that the controller is stabilizing around a polynomially small neighborhood of the equilibrium, in experiments, we find that the resulting controller successfully stabilizes the dynamics for a wide range of initial conditions. Relative to the case of linear systems where we leveraged the monotonicity of the LQR cost to search for discount factors using binary search, this monotonicity breaks down in the case of nonlinear systems and we instead analyze a random search algorithm to simplify the analysis.

## 4 Experiments

In this section, we evaluate the ability of the discount annealing algorithm to stabilize a simulated nonlinear system. Specifically, we consider the familiar cart-pole, with $d_x = 4$ (positions and velocities of the cart and pole), and $d_u = 1$ (horizontal force applied to the cart). The goal is to stabilize the system with the pole in the unstable 'upright' equilibrium position. For further details, including the precise dynamics, see Appendix D.1. The system was simulated in discrete-time with a simple forward Euler discretization, i.e., $\mathbf{x}_{t+1} = \mathbf{x}_t + T_s\dot{\mathbf{x}}_t$, where $\dot{\mathbf{x}}_t$ is given by the continuous time dynamics, and $T_s = 0.05$ (20Hz). Simulations were carried out in PyTorch [Paszke et al., 2019] and run on a single GPU.

Table 1: Final region of attraction radius $r_{\mathrm{roa}}$ as a function of the initial state radius $r$ used during training (discount annealing). We report the [min, max] values of $r_{\mathrm{roa}}$ over 5 independent trials. The optimal LQR policy for the linearized system achieved $r_{\mathrm{roa}} = 0.703$ when applied to the nonlinear system. We also synthesized an $\mathcal{H}_\infty$ optimal controller for the linearized dynamics, which achieved $r_{\mathrm{roa}} = 0.506$.

| $r$ | 0.1 | 0.3 | 0.5 | 0.6 | 0.7 |
|---|---|---|---|---|---|
| $r_{\mathrm{roa}}$ | [0.702, 0.704] | [0.711, 0.713] | [0.727, 0.734] | [0.731, 0.744] | [0.769, 0.777] |

**Setup**. The discounted annealing algorithm of Figure 1 was implemented as follows. In place of the true infinite horizon discounted cost $J_{\mathrm{nl}}(K \mid \gamma, r)$ in Eq. (C.4) we use a finite horizon, finite sample Monte Carlo approximation as described in Appendix C,

$$J_{\mathrm{nl}}(K \mid \gamma) \approx \frac{1}{N} \sum_{i=1}^{N} J_{\mathrm{nl}}^{(H)}(K \mid \gamma, \mathbf{x}^{(i)}), \quad \mathbf{x}^{(i)} \sim r \cdot \mathcal{S}^{d_x - 1}.$$

Here, $J_{\mathrm{nl}}^{(H)}(K \mid \gamma, \mathbf{x}) = \sum_{j=0}^{H-1} \mathbf{x}_t^\top Q \mathbf{x}_t + \mathbf{u}_t^\top R \mathbf{u}_t$, is the length $H$, finite horizon cost of a controller $K$ in which the states evolve according to the $\sqrt{\gamma}$ damped dynamics from Eq. (C.5) and $\mathbf{u}_t = K \mathbf{x}_t$. We used $N = 5000$ and $H = 1000$ in our experiments. For the cost function, we used $Q = T_s \cdot I$ and $R = T_s$. We compute unbiased approximations of the gradients using automatic differentiation on the finite horizon objective $J_{\mathrm{nl}}^{(H)}$.

Instead of using SGD updates for policy gradients, we use Adam [Kingma and Ba, 2014] with a learning rate of $\eta = 0.01/r$. Furthermore, we replace the policy gradient termination criteria in Step 2 (Eq. (2.2)) by instead halting after a fixed number ($M = 200$) of gradient descent steps. We wish to emphasize that the hyperparameters $(N, H, \eta, M)$ were not optimized for performance. In particular, for $r = 0.1$, we found that as few as $M = 40$ iterations of policy gradient and horizons as short as $H = 400$ were sufficient. Finally, we used an initial discount factor $\gamma_0 = 0.9 \cdot \|A_{\mathrm{jac}}\|_2^{-2}$, where $A_{\mathrm{jac}}$ denotes the linearization of the (discrete-time) cart-pole about the vertical equilibrium.

**Results**. We now proceed to discuss the performance of the algorithm, focusing on three main properties of interest: i) the number of iterations of discount annealing required to find a stabilizing controller (that is, increase $\gamma_t$ to 1), ii) the maximum radius $r$ of the ball of initial conditions $\mathbf{x}_0 \sim r \cdot \mathcal{S}^{d_x - 1}$ for which discount annealing succeeds at stabilizing the system, and iii) the radius $r_{\mathrm{roa}}$ of the largest ball contained within the region of attraction (ROA) for the policy returned by discount annealing. Although the true ROA (the set of all initial conditions such that the closed-loop system converges asymptotically to the equilibrium point) is not necessarily shaped like a ball (as the system is more sensitive to perturbations in the position and velocity of the pole than the cart), we use the term region of attraction radius to refer to the radius of the largest ball contained in the ROA.

Concerning (i), discount annealing reliably returned a stabilizing policy in less than 9 iterations. Specifically, over 5 independent trials for each initial radius $r \in \{0.1, 0.3, 0.5, 0.7\}$ (giving 20 independent trials, in total) the algorithm never required more than 9 iterations to return a stabilizing policy.

Concerning (ii), discount annealing reliably stabilized the system for $r \leq 0.7$. For $r \approx 0.75$, we observed trials in which the state of the damped system ($\gamma < 1$) diverged to infinity. For such a rollout, the gradient of the cost is not well-defined, and policy gradient is unable to improve the policy, which prevents discount annealing from finding a stabilizing policy.

Concerning (iii), in Table 1 we report the final radius $r_{\mathrm{roa}}$ for the region of attraction of the final controller returned by discount annealing as a function of the training radius $r$. We make the following observations. Foremost, the policy returned by discount annealing extends the radius of the ROA beyond the radius used during training, i.e. $r_{\mathrm{roa}} > r$. Moreover, for each $r > .1$, the $r_{\mathrm{roa}}$ achieved by discount annealing is greater than the $r_{\mathrm{roa}} = 0.703$ achieved by the exact optimal LQR controller *and* the $r_{\mathrm{roa}} = 0.506$ achieved by the exact optimal $\mathcal{H}_\infty$ controller for the system's Jacobian linearization (see Table 1). (The $\mathcal{H}_\infty$ optimal controller mitigates the effect of worst-case additive state disturbances on the cost; cf. Appendix D.2 for details).

One may hypothesize that this is due to the fact that discount annealing directly operates on the true nonlinear dynamics whereas the other baselines (LQR and $\mathcal{H}_\infty$ control), find the optimal controller

for an idealized linearization of the dynamics. Indeed, there is evidence to support this hypothesis. In Figure 3 presented in Appendix D, we plot the error $\|K_{\mathrm{pg}}^{\star}(\gamma_t) - K_{\mathrm{lin}}^{\star}(\gamma_t)\|_F$ between the policy $K_{\mathrm{pg}}^{\star}(\gamma_t)$ returned by policy gradients, and the optimal LQR policy $K_{\mathrm{lin}}^{\star}(\gamma_t)$ for the (damped) linearized system, as a function of the discount factor $\gamma_t$ used in each iteration of the discount annealing algorithm. For small training radius, such as $r = 0.05$, $K_{\mathrm{pg}}^{\star}(\gamma_t) \approx K_{\mathrm{lin}}^{\star}(\gamma_t)$ for all $\gamma_t$. However, for larger radii (i.e $r = 0.7$), we see that $\|K_{\mathrm{pg}}^{\star}(\gamma_t) - K_{\mathrm{lin}}^{\star}(\gamma_t)\|_F$ steadily increases as $\gamma_t$ increases.

That is, as discount annealing increases the discount factor $\gamma$ and the closed-loop trajectories explore regions of the state space where the dynamics are increasingly nonlinear, $K_{\mathrm{pg}}^{\star}$ begins to diverge from $K_{\mathrm{lin}}^{\star}$. Moreover, at the conclusion of discount annealing $K_{\mathrm{pg}}^{\star}(1)$ achieves a lower cost, namely [15.2, 15.4] vs [16.5, 16.8] (here $[a, b]$ denotes [min, max] over 5 trials) and larger $r_{\mathrm{roa}}$, namely [0.769, 0.777] vs [0.702, 0.703], than $K_{\mathrm{lin}}^{\star}(1)$, suggesting that the method has indeed adapted to the nonlinearity of the system. Similar observations as to the behavior of controllers fine tuned via policy gradient methods are predicted by the theoretical results from Qu et al. [2020].

# 5 Discussion

This works illustrates how one can provably stabilize a broad class of dynamical systems via a simple model-free procedure based off policy gradients. In line with the simplicity and flexibility that have made model-free methods so popular in practice, our algorithm works under relatively weak assumptions and with little knowledge of the underlying dynamics. Furthermore, we solve an open problem from previous work [Fazel et al., 2018] and take a step towards placing model-free methods on more solid theoretical footing. We believe that our results raise a number of interesting questions and directions for future work.

In particular, our theoretical analysis states that discount annealing returns a controller whose stability properties are similar to those of the optimal LQR controller for the system's Jacobian linearization. We were therefore quite surprised when in experiments, the resulting controller had a significantly better radius of attraction than the exact optimal LQR and $\mathcal{H}_{\infty}$ controllers for the linearization of the dynamics. It is an interesting and important direction for future work to gain a better understanding of exactly when and how model-free procedures are adaptive to the nonlinearities of the system and improve upon these model-based baselines. Furthermore, for our analysis of nonlinear systems, we require access to damped system trajectories. It would be valuable to understand whether this is indeed necessary or whether our analysis could be extended to work without access to damped trajectories.

As a final note, in this work we reduce the problem of stabilizing dynamical systems to running policy gradients on a discounted LQR objective. This choice of reducing to LQR was in part made for simplicity to leverage previous analyses. However, it is possible that overall performance could be improved if rather than reducing to LQR, we instead attempted to run a model-free method that directly tries to optimize a robust control objective (which explicitly deals with uncertainty in the system dynamics). We believe that understanding these tradeoffs in objectives and their relevant sample complexities is an interesting avenue for future inquiry.

## Acknowledgments

We would like to thank Peter Bartlett for many helpful conversations and comments throughout the course of this project, and Russ Tedrake for support with the numerical experiments. JCP is supported by an NSF Graduate Research Fellowship. MS is supported by an Open Philanthropy Fellowship grant. JU is supported by the National Science Foundation, Award No. EFMA-1830901, and the Department of the Navy, Office of Naval Research, Award No. N00014-18-1-2210.

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
