^\star_{\text{pg}}(\gamma_t) - K^\star_{\text{lin}}(\gamma_t)\|_F$ between the policy $K^\star_{\text{pg}}(\gamma_t)$ returned by policy gradients, and the optimal LQR policy $K^\star_{\text{lin}}(\gamma_t)$ for the (damped) linearized system, as a function of the discount factor $\gamma_t$ used in each iteration of the discount annealing algorithm. For small training radius, such as $r = 0.05$, $K^\star_{\text{pg}}(\gamma_t) \approx K^\star_{\text{lin}}(\gamma_t)$ for all $\gamma_t$. However, for larger radii (i.e $r = 0.7$), we see that $\|K^\star_{\text{pg}}(\gamma_t) - K^\star_{\text{lin}}(\gamma_t)\|_F$ steadily increases as $\gamma_t$ increases.

That is, as discount annealing increases the discount factor $\gamma$ and the closed-loop trajectories explore regions of the state space where the dynamics are increasingly nonlinear, $K^\star_{\text{pg}}$ begins to diverge from $K^\star_{\text{lin}}$. Moreover, at the conclusion of discount annealing $K^\star_{\text{pg}}(1)$ achieves a lower cost, namely [15.2, 15.4] vs [16.5, 16.8] (here $[a, b]$ denotes [min, max] over 5 trials) and larger $r_{\text{roa}}$, namely [0.769, 0.777] vs [0.702, 0.703], than $K^\star_{\text{lin}}(1)$, suggesting that the method has indeed adapted to the nonlinearity of the system. Similar observations as to the behavior of controllers fine tuned via policy gradient methods are predicted by the theoretical results from Qu et al. [2020].

## 5 Discussion

This works illustrates how one can provably stabilize a broad class of dynamical systems via a simple model-free procedure based off policy gradients. In line with the simplicity and flexibility that have made model-free methods so popular in practice, our algorithm works under relatively weak assumptions and with little knowledge of the underlying dynamics. Furthermore, we solve an open problem from previous work [Fazel et al., 2018] and take a step towards placing model-free methods on more solid theoretical footing. We believe that our results raise a number of interesting questions and directions for future work.

In particular, our theoretical analysis states that discount annealing returns a controller whose stability properties are similar to those of the optimal LQR controller for the system's Jacobian linearization. We were therefore quite surprised when in experiments, the resulting controller had a significantly better radius of attraction than the exact optimal LQR and $\mathcal{H}_\infty$ controllers for the linearization of the dynamics. It is an interesting and important direction for future work to gain a better understanding of exactly when and how model-free procedures are adaptive to the nonlinearities of the system and improve upon these model-based baselines. Furthermore, for our analysis of nonlinear systems, we require access to damped system trajectories. It would be valuable to understand whether this is indeed necessary or whether our analysis could be extended to work without access to damped trajectories.

As a final note, in this work we reduce the problem of stabilizing dynamical systems to running policy gradients on a discounted LQR objective. This choice of reducing to LQR was in part made for simplicity to leverage previous analyses. However, it is possible that overall performance could be improved if rather than reducing to LQR, we instead attempted to run a model-free method that directly tries to optimize a robust control objective (which explicitly deals with uncertainty in the system dynamics). We believe that understanding these tradeoffs in objectives and their relevant sample complexities is an interesting avenue for future inquiry.

## Acknowledgments

We would like to thank Peter Bartlett for many helpful conversations and comments throughout the course of this project, and Russ Tedrake for support with the numerical experiments. JCP is supported by an NSF Graduate Research Fellowship. MS is supported by an Open Philanthropy Fellowship grant. JU is supported by the National Science Foundation, Award No. EFMA-1830901, and the Department of the Navy, Office of Naval Research, Award No. N00014-18-1-2210.

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

## Table of Contents: Appendix

## A  Deferred Proofs and Analysis for the Linear Setting

**Preliminaries on Linear Quadratic Control.**    The cost a state-feedback controller $J_{\mathrm{lin}}(K \mid \gamma)$ is intimately related to the solution of the discrete-time Lyapunov equation. Given a stable matrix $A_{\mathrm{cl}}$ and a symmetric positive definite matrix $\Sigma$, we define $\mathsf{dlyap}(A_{\mathrm{cl}}, \Sigma)$ to be the unique solution (over $X$) to the matrix equation,

$$X = \Sigma + A_{\mathrm{cl}}^\top X A_{\mathrm{cl}}.$$

A classical result in Lyapunov theory states that $\mathsf{dlyap}(A_{\mathrm{cl}}, \Sigma) = \sum_{j=0}^\infty (A_{\mathrm{cl}}^j)^\top X A_{\mathrm{cl}}^j$. Recalling our earlier definition, for a controller $K$ such that $\sqrt{\gamma}(A+BK)$ is stable, we let $P_{K,\gamma} := \mathsf{dlyap}(\sqrt{\gamma}(A+BK), Q+K^\top RK)$, where $Q, R$ are the cost matrices for the LQR problem defined in Definition 2.1. As a special case, we let $P_\star := \mathsf{dlyap}(A + BK_\star, Q + K_\star^\top RK_\star)$ where $K_\star := K_{\star,1}$ is the optimal controller for the undiscounted LQR problem. Using these definitions, we have the following facts:

**Fact A.1.**  $J_{\mathrm{lin}}(K \mid \gamma, A, B) < \infty$ *if and only if* $\sqrt{\gamma} \cdot (A + BK)$ *is a stable matrix.*

**Fact A.2.**  *If* $J_{\mathrm{lin}}(K \mid \gamma, A, B) < \infty$ *then for all* $\mathbf{x} \in \mathbb{R}^{d_x}$, $J_{\mathrm{lin}}(K \mid \mathbf{x}, \gamma, A, B) = \mathbf{x}^\top P_{K,\gamma}\mathbf{x}$. *Furthermore,*

$$J_{\mathrm{lin}}(K \mid \gamma, A, B) = \mathop{\mathbb{E}}_{\mathbf{x}_0 \sim \sqrt{d_x}\mathcal{S}^{d_x-1}} \mathbf{x}_0^\top P_{K,\gamma}\mathbf{x}_0 = \mathrm{tr}\left[P_{K,\gamma}\right].$$

Employing these identities, we can now restate and prove Lemma 2.1:

**Lemma 2.1.**  *For all $K$ such that* $J(K \mid \gamma, A, B) < \infty$, $J_{\mathrm{lin}}(K \mid \gamma, A, B) = J_{\mathrm{lin}}(K \mid 1, \sqrt{\gamma} \cdot A, \sqrt{\gamma} \cdot B)$.

*Proof.* From the definition of the LQR cost and linear dynamics in Definition 2.1, for $A_{\mathrm{cl}} := A + BK$,

$$J_{\mathrm{lin}}(K \mid \gamma, A, B) = \mathbb{E}_{\mathbf{x}_0}\left[\sum_{t=0}^{\infty} \gamma^t \cdot \mathbf{x}_0^\top \left(A_{\mathrm{cl}}^t\right)^\top \left(Q + K^\top R K\right) A_{\mathrm{cl}}^t \mathbf{x}_0\right]$$

$$= \mathbb{E}_{\mathbf{x}_0}\left[\sum_{t=0}^{\infty} \mathbf{x}_0^\top \left((\sqrt{\gamma}A_{\mathrm{cl}})^t\right)^\top \left(Q + K^\top R K\right) (\sqrt{\gamma}A_{\mathrm{cl}})^t \mathbf{x}_0\right]$$

$$= J_{\mathrm{lin}}(K \mid 1, \sqrt{\gamma} \cdot A, \sqrt{\gamma} \cdot B).$$

$\square$

Therefore, since LQR with a discount factor is equivalent to LQR with damped dynamics, it follows that for $\gamma_0 < \rho(A)^{-2}$, (noisy) policy gradients initialized at the zero controller converges to the global optimum of the discounted LQR problem. The following lemma is essentially a restatement of the finite sample convergence result for gradient descent on LQR (Theorem 31) from Fazel et al. [2018], where we have set $Q = R = I$ and $\mathbb{E}\left[\mathbf{x}_0\mathbf{x}_0^\top\right] = I$ as in the discount annealing algorithm. We include the proof of this result in Appendix A.2 for the sake of completeness.

**Lemma A.3** (Fazel et al. [2018]). *For $K_0$ such that $J_{\mathrm{lin}}(K_0 \mid \gamma) < \infty$, define (noisy) policy gradients as the procedure which computes updates according to,*

$$K_{t+1} = K_t - \eta \widetilde{\nabla}_t,$$

*for some matrix $\widetilde{\nabla}_t$. There exists a choice of a constant step size $\eta > 0$ and a fixed polynomial,*

$$\mathcal{C}_{\mathrm{PG}} := \mathrm{poly}\left(\frac{1}{J_{\mathrm{lin}}(K_0 \mid \gamma)}, \frac{1}{\|A\|_{\mathrm{op}}}, \frac{1}{\|B\|_{\mathrm{op}}}\right),$$

*such that if the following inequality holds for all $t = 1 \ldots T$,*

$$\left\|\nabla_K J_{\mathrm{lin}}(K_t \mid \gamma) - \widetilde{\nabla}_t\right\|_{\mathrm{F}} \leq \mathcal{C}_{\mathrm{PG}} \cdot \varepsilon, \tag{A.1}$$

*then $J_{\mathrm{lin}}(K_T \mid \gamma) - \min_K J_{\mathrm{lin}}(K \mid \gamma) \leq \varepsilon$ whenever*

$$T \geq \mathrm{poly}\left(\|A\|_{\mathrm{op}}, \|B\|_{\mathrm{op}}, J_{\mathrm{lin}}(K_0 \mid \gamma)\right) \log\left(\frac{J_{\mathrm{lin}}(K_0 \mid \gamma) - \min_K J_{\mathrm{lin}}(K \mid \gamma)}{\varepsilon}\right).$$

With this lemma, we can now present the proof of Theorem 1.

## A.1 Discount annealing on linear systems: Proof of Theorem 1

We organize the proof into two main parts. First, we prove statements $c$) and $d$) by an inductive argument. Then, having already proved part $b$) in the main body of the paper, we finish the proof by establishing the stability guarantees of the resulting controller outlined in part $a$).

### A.1.1 Proof of $c$) and $d$)

**Base case.** Recall that at iteration $t$ of discount annealing, policy gradients is initialized at $K_{t,0} := K_t$ and (in the case of linear systems) computes updates according to:

$$K_{t,j+1} = K_{t,j} - \eta \cdot \varepsilon\text{-}\mathtt{Grad}\left(J_{\mathrm{lin}}(\cdot \mid \gamma_t), K_{t,j}\right).$$

From Lemma A.3, if $J_{\mathrm{lin}}(K_{t,0} \mid \gamma_t) < \infty$ and

$$\|\varepsilon\text{-}\mathtt{Grad}\left(J_{\mathrm{lin}}(\cdot \mid \gamma_t), K_{t,j}\right) - \nabla J_{\mathrm{lin}}(K_{t,j} \mid \gamma_t)\|_{\mathrm{F}} \leq \mathcal{C}_{\mathrm{pg},t} d_x \quad \forall j \geq 0, \tag{A.2}$$

where $\mathcal{C}_{\mathrm{pg},t} = \mathrm{poly}(1/J_{\mathrm{lin}}(K_{t,0} \mid \gamma_t), 1/\|A\|_{\mathrm{op}}, 1/\|B\|_{\mathrm{op}})$ then policy gradients will converge to a $K_{t,j}$ such that $J_{\mathrm{lin}}(K_{t,j} \mid \gamma_t) - \min_K J_{\mathrm{lin}}(K \mid \gamma_t) \leq d_x$ after $\mathrm{poly}(\|A\|_{\mathrm{op}}, \|B\|_{\mathrm{op}}, J_{\mathrm{lin}}(K_{t,0}))$ many iterations.

By our choice of discount factor, we have that $J_{\mathrm{lin}}(0 \mid \gamma_0) < \infty$. Furthermore, since $\varepsilon \leq \mathrm{poly}(1/J_{\mathrm{lin}}(0 \mid \gamma_0), 1/\|A\|_{\mathrm{op}}, 1/\|B\|_{\mathrm{op}})$, then the condition outlined in Eq. (A.2) holds and policy gradients achieves the desired guarantee when $t = 0$.

The correctness of binary search at iteration $t = 0$ follows from Lemma C.6. In particular, we instantiate the lemma with $f = J_{\mathrm{lin}}(K_{t+1} \mid \cdot)$ of the LQR objective, which is nondecreasing in $\gamma$ by definition, $f_1 = 2.5 J_{\mathrm{lin}}(K_{t+1} \mid \gamma_t)$ and $f_2 = 8 J_{\mathrm{lin}}(K_{t+1} \mid \gamma_t)$. The algorithm requires auxiliary values $\overline{f}_1 \in [2.5 J_{\mathrm{lin}}(K_{t+1} \mid \gamma_t), 3 J_{\mathrm{lin}}(K_{t+1} \mid \gamma_t)]$ and $\overline{f}_2 \in [7 J_{\mathrm{lin}}(K_{t+1} \mid \gamma_t), 7.5 J_{\mathrm{lin}}(K_{t+1} \mid \gamma_t)]$ which we can always compute by using $\varepsilon$-Eval to estimate the cost $J_{\mathrm{lin}}(K_{t+1} \mid \gamma_t)$ to precision $.1 d_x$ (recall that $J_{\mathrm{lin}}(K \mid \gamma) \geq d_x$ for any $K$ and $\gamma$). The last step needed to apply the lemma is to lower bound the width of the feasible region of $\gamma$'s which satisfy the desired criterion that $J_{\mathrm{lin}}(K_{t+1} \mid \gamma) \in [\overline{f}_1, \overline{f}_2]$.

Let $\gamma' \geq \gamma_t$ be such that $J_{\mathrm{lin}}(K_{t+1} \mid \gamma') = 3 J_{\mathrm{lin}}(K_{t+1} \mid \gamma_t)$. Such a $\gamma$ is guaranteed to exist since $J_{\mathrm{lin}}(K_{t+1} \mid \cdot)$ is nondecreasing in $\gamma$ and it is a continuous function for all $\gamma < \overline{\gamma} := \sup\{\gamma : J_{\mathrm{lin}}(K_{t+1} \mid \gamma) < \infty\}$. By the calculation from the proof of part $b$) presented in the main body of the paper, for

$$\gamma'' := \left( \frac{1}{8 \| P_{K_{t+1}, \gamma'} \|_{\mathrm{op}}^4} + 1 \right)^2 \gamma',$$

we have that $J_{\mathrm{lin}}(K_{t+1} \mid \gamma'') \leq 2 J_{\mathrm{lin}}(K_{t+1} \mid \gamma') = 6 J_{\mathrm{lin}}(K_{t+1} \mid \gamma_t)$. By monotonicity and continuity of $J_{\mathrm{lin}}(K_{t+1} \mid \cdot)$, when restricted to $\gamma \leq \overline{\gamma}$, all $\gamma \in [\gamma', \gamma'']$ satisfy $J_{\mathrm{lin}}(K_{t+1} \mid \gamma) \in [3 J_{\mathrm{lin}}(K_{t+1} \mid \gamma_t), 6 J_{\mathrm{lin}}(K_{t+1} \mid \gamma_t)]$. Moreover,

$$\gamma'' - \gamma' = \left[ \left( \frac{1}{8 \| P_{K_{t+1}, \gamma'} \|_{\mathrm{op}}^4} + 1 \right)^2 - 1 \right] \gamma' \geq \frac{1}{4 \| P_{K_{t+1}, \gamma'} \|_{\mathrm{op}}^4} \gamma' \geq \frac{1}{4 \mathrm{tr} \left[ P_{K_{t+1}, \gamma'} \right]^4} \gamma_t,$$

where the last line follows from the fact that $\gamma' \geq \gamma_t$ and that the trace of a PSD matrix is always at least as large as the operator norm. Lastly, since $\mathrm{tr}\left[ P_{K_{t+1}, \gamma'} \right] = J_{\mathrm{lin}}(K_{t+1} \mid \gamma') = 3 J_{\mathrm{lin}}(K_{t+1} \mid \gamma_t)$, by the guarantee of policy gradients, we have that for $t = 0$, $J_{\mathrm{lin}}(K_1 \mid \gamma_0) \leq 2 \mathrm{tr}\left[ P_\star \right]$. Therefore, for $t = 0$:

$$\gamma'' - \gamma' \geq \frac{1}{4 (6 \mathrm{tr}\left[ P_\star \right])^4}$$

Hence the width of the feasible region is at least $\frac{1}{5184 \mathrm{tr}[P_\star]^4} \gamma_0$.

**Inductive step.** To show that policy gradients achieves the desired guarantee at iteration $t + 1$, we can repeat the exact same argument as in the base case. The only difference is that the need to argue that the cost of the initial controller $\sup_{t \geq 0} J_{\mathrm{lin}}(K_t \mid \gamma_t)$ is uniformly bounded across iterations. By the inductive hypothesis on the success of the binary search algorithm at iteration $t - 1$, we have that,

$$
\begin{aligned}
J_{\mathrm{lin}}(K_t \mid \gamma_t) &\leq 8 J_{\mathrm{lin}}(K_t \mid \gamma_{t-1}) \\
&\leq 8 \left( \min_K J_{\mathrm{lin}}(K \mid \gamma_{t-1}) + d_x \right) \\
&\leq 8 \left( \min_K J_{\mathrm{lin}}(K \mid 1) + d_x \right) \\
&\leq 16 \mathrm{tr}\left[ P_\star \right].
\end{aligned}
$$

Hence, by Lemma A.3, policy gradients achieves the desired guarantee using $\mathrm{poly}(M_{\mathrm{lin}}, \|A\|_{\mathrm{op}}, \|B\|_{\mathrm{op}})$ many queries to $\varepsilon$-Grad$(\cdot, J_{\mathrm{lin}}(\cdot \mid \gamma))$ as long as $\varepsilon$ is less than $\mathrm{poly}(M_{\mathrm{lin}}^{-1}, \|A\|_{\mathrm{op}}^{-1}, \|B\|_{\mathrm{op}}^{-1})$.

Likewise, the argument for the correctness of the binary search procedure is identical to that of the base case. Because of the success of policy gradients and binary search at the previous iteration, we can upper bound, $\mathrm{tr}\left[ P_{K_{t+1}, \gamma'} \right]$ by $6 \mathrm{tr}\left[ P_\star \right]$ and get a uniform lower bound on the width of the feasible region.

### A.1.2 Proof of $a$)

After halting, we see that discount annealing returns a controller $\widehat{K}$ satisfying the stated condition from Step 2 requiring that,

$$J_{\mathrm{lin}}(\widehat{K} \mid 1) - J_{\mathrm{lin}}(K_\star \mid 1) = \mathrm{tr}\left[ \widehat{P} - P_\star \right] \leq d_x.$$

Here, we have used Fact A.2 to rewrite $J_{\mathrm{lin}}(\widehat{K} \mid 1)$ as $\mathrm{tr}[\widehat{P}]$ for $\widehat{P} := \mathsf{dlyap}(A + B\widehat{K}, Q + \widehat{K}^\top R\widehat{K})$ (and likewise for $P_\star$). Since $\mathrm{tr}\,[P_\star] \geq d_x$, we conclude that $\mathrm{tr}[\widehat{P}] \leq 2\mathrm{tr}\,[P_\star]$. Now, by properties of the Lyapunov equation (see Lemma A.5) the following holds for $A_{\mathrm{cl}} := A + B\widehat{K}$:

$$\|A_{\mathrm{cl}}^2\|_{\mathrm{op}}^t \;\leq\; \|\widehat{P}\|_{\mathrm{op}} \left(1 - \frac{1}{\|\widehat{P}\|_{\mathrm{op}}}\right)^t \;\leq\; \mathrm{tr}[\widehat{P}] \exp\left(-t/\mathrm{tr}[\widehat{P}]\right).$$

Hence, we conclude that,

$$\|\mathbf{x}_t\| = \|A_{\mathrm{cl}}^t \mathbf{x}_0\| \;\leq\; \|A_{\mathrm{cl}}^t\|_{\mathrm{op}} \|\mathbf{x}_0\| \;\leq\; \sqrt{2\mathrm{tr}[P_\star]} \exp\left(-\frac{1}{4\mathrm{tr}\,[P_\star]} \cdot t\right) \|\mathbf{x}_0\|.$$

## A.2  Convergence of policy gradients for LQR: Proof of Lemma A.3

*Proof.* Note that, by Lemma 2.1, proving the above result for $J_{\mathrm{lin}}(\cdot \mid \gamma, A, B)$ is the same as proving it for $J_{\mathrm{lin}}(\cdot \mid 1, \sqrt{\gamma}A, \sqrt{\gamma}B)$. We start by defining the following idealized updates,

$$K' = K - \eta \nabla_K J_{\mathrm{lin}}(K_t \mid \gamma)$$
$$K'' = K - \eta \widetilde{\nabla}_t.$$

From Lemmas 13, 24, and 25 from Fazel et al. [2018], there exists a fixed polynomial,

$$\mathcal{C}_\eta := \mathrm{poly}\left(\frac{1}{\sqrt{\gamma}\|A\|_{\mathrm{op}}}, \frac{1}{\sqrt{\gamma}\|B\|_{\mathrm{op}}}, \frac{1}{J_{\mathrm{lin}}(K_0 \mid \gamma)}\right)$$

Such that, for $\eta \leq \mathcal{C}_\eta$, the following inequality holds,

$$J_{\mathrm{lin}}(K' \mid \gamma) - J_{\mathrm{lin}}(K_\star \mid \gamma) \;\leq\; \left(1 - \eta \frac{1}{\|\Sigma_{K_\star}\|_{\mathrm{op}}}\right) (J_{\mathrm{lin}}(K \mid \gamma) - J_{\mathrm{lin}}(K_\star \mid \gamma)),$$

where $\Sigma_{K_\star} = \mathbb{E}_{\mathbf{x}_0}[\sum_{t=0}^\infty \mathbf{x}_{t,\star}^\top \mathbf{x}_{t,\star}]$ and $\{\mathbf{x}_{t,\star}\}_{t=0}^\infty$ is the sequence of states generated by the controller $K_\star$. Therefore, if $J_{\mathrm{lin}}(K'' \mid \gamma)$ and $J_{\mathrm{lin}}(K' \mid \gamma)$ satisfy,

$$|J_{\mathrm{lin}}(K'' \mid \gamma) - J_{\mathrm{lin}}(K' \mid \gamma)| \;\leq\; \frac{1}{2\|\Sigma_{K_\star}\|_{\mathrm{op}}} \eta\varepsilon \tag{A.3}$$

then, as long as $J_{\mathrm{lin}}(K \mid \gamma) - J_{\mathrm{lin}}(K_\star \mid \gamma) \geq \varepsilon$, this following inequality also holds:

$$J_{\mathrm{lin}}(K'' \mid \gamma) - J_{\mathrm{lin}}(K_\star \mid \gamma) \;\leq\; \left(1 - \eta \frac{1}{\|2\Sigma_{K_\star}\|_{\mathrm{op}}}\right) (J_{\mathrm{lin}}(K \mid \gamma) - J_{\mathrm{lin}}(K_\star \mid \gamma)).$$

The proof then follows by unrolling the recursion and simplifying. We now focus on establishing Eq. (A.3). By Lemma 27 in Fazel et al. [2018], if

$$\|K'' - K'\|_{\mathrm{op}} = \eta\|\nabla_K J_{\mathrm{lin}}(K_j \mid \gamma) - \widetilde{\nabla}_j\|_{\mathrm{op}} \;\leq\; \mathcal{C}_K$$

where $\mathcal{C}_K$ is a fixed polynomial $\mathcal{C}_K := \mathrm{poly}(\frac{1}{J_{\mathrm{lin}}(K_0|\gamma)}, \frac{1}{\sqrt{\gamma}\|A\|_{\mathrm{op}}}, \frac{1}{\sqrt{\gamma}\|B\|_{\mathrm{op}}})$, then

$$|J_{\mathrm{lin}}(K'' \mid \gamma) - J_{\mathrm{lin}}(K' \mid \gamma)| \;\leq\; \mathcal{C}_{\mathrm{cost}}\|K'' - K'\|_{\mathrm{op}} = \mathcal{C}_{\mathrm{cost}} \cdot \eta\|\nabla_K J_{\mathrm{lin}}(K_j \mid \gamma) - \widetilde{\nabla}_j\|_{\mathrm{op}},$$

where $\mathcal{C}_{\mathrm{cost}} := \mathrm{poly}(d_x, \|R\|_{\mathrm{op}}, \|B\|_{\mathrm{op}}, J_{\mathrm{lin}}(K_0))$. Therefore, Eq. (A.3) holds if

$$\|\nabla_K J_{\mathrm{lin}}(K_j \mid \gamma) - \widetilde{\nabla}_j\|_{\mathrm{op}} \;\leq\; \min\left\{\frac{1}{2\|\Sigma_{K_\star}\|_{\mathrm{op}}\mathcal{C}_{\mathrm{cost}}}\varepsilon,\ \mathcal{C}_K\right\}.$$

The exact statement follows from using $d_x \leq J_{\mathrm{lin}}(K_0 \mid \gamma)$ and $\|\Sigma_{K_\star,\gamma}\|_{\mathrm{op}} \leq J_{\mathrm{lin}}(K_{\star,\gamma}) \leq J_{\mathrm{lin}}(K_0 \mid \gamma)$ by Lemma 13 in [Fazel et al., 2018] and taking the polynomial in the proposition statement to be the minimum of $\mathcal{C}_K$ and $1/\mathcal{C}_{\mathrm{cost}}$. $\qquad\square$

## A.3   Impossibility of reward shaping: Proof of Proposition 2.2

*Proof.* Consider the linear dynamical system with dynamics matrices,

$$A = \begin{bmatrix} 0 & 0 \\ 0 & 2 \end{bmatrix}, \quad B = \begin{bmatrix} 1 \\ \beta \end{bmatrix}$$

where $\beta > 0$ is a parameter to be chosen later. Note that a linear dynamical system of these dimensions is controllable (and hence stabilizable) [Callier and Desoer, 2012], since the matrix

$$[B \quad AB] = \begin{bmatrix} 1 & 0 \\ \beta & 2\beta \end{bmatrix}$$

is full rank. For any controller $K = [k_1 \quad k_2]$, the closed loop system $A_{\mathrm{cl}} := A + BK$ has the form,

$$A_{\mathrm{cl}} = \begin{bmatrix} k_1 & k_2 \\ \beta k_1 & 2 + \beta k_2 \end{bmatrix}.$$

By Gershgorin's circle theorem, $A_{\mathrm{cl}}$ has an eigenvalue $\lambda$ which satisfies,

$$|\lambda| \geq |2 + \beta k_2| - |\beta k_1| \geq 2 - 2\beta \max\{|k_1|, |k_2|\}.$$

Therefore, any controller $K$ for which the closed-loop system $A + BK$ is stable must have the property that,

$$\max\{|k_1|, |k_2|\} \geq \frac{1}{2\beta}.$$

Using this observation and Fact A.2, for any discount factor $\gamma$, a stabilizing controller $K$ must satisfy,

$$\begin{aligned}
J_{\mathrm{lin}}(K \mid \gamma) = \mathrm{tr}\,[P_{K,\gamma}] \\
\geq \mathrm{tr}\,[Q] + \mathrm{tr}\,[K^\top R K] \\
\geq (k_1^2 + k_2^2) \cdot \sigma_{\min}(R) \\
\geq \frac{1}{4\beta^2} \cdot \sigma_{\min}(R).
\end{aligned}$$

In the above calculation, we have used the identity $P_{K,\gamma} = Q + K^\top R K + \gamma A_{\mathrm{cl}}^\top P_{K,\gamma} A_{\mathrm{cl}}$ as well as the assumption that $R$ is positive definite. Next, we observe that for a discount factor $\gamma = c^2 \cdot \rho(A)^{-2}$, where $c \in (0, 1)$ as chosen in the initial iteration of our algorithm, the cost of the 0 controller has the following upper bound:

$$\begin{aligned}
J_{\mathrm{lin}}(0 \mid \gamma_0) = \sum_{j=0}^{\infty} c^{2j} \rho(A)^{-2j} \cdot \mathrm{tr}\,[(A^\top)^j Q A^j] \\
\leq \|Q\|_{\mathrm{op}} \sum_{j=0}^{\infty} c^{2j} \rho(A)^{-2j} \|A^j\|_{\mathrm{F}}^2 \\
= \|Q\|_{\mathrm{op}} \sum_{j=0}^{\infty} \left\| \left( \frac{c}{\rho(A)} \cdot A \right)^j \right\|_{\mathrm{F}}^2.
\end{aligned}$$

Using standard Lyapunov arguments (see for example Section D.2 in Perdomo et al. [2021]) the sum in the last line is a geometric series and is equal to some function $f(c, A) < \infty$, which depends *only* on $c$ and $A$, for all $c \in (0, 1)$. Using this calculation, it follows that

$$\min_K J_{\mathrm{lin}}(K \mid \gamma_0) \leq J_{\mathrm{lin}}(0 \mid \gamma) \leq \|Q\|_{\mathrm{op}} f(c, A)$$

Hence, for any $Q$, $R$, and discount factor $\gamma \in (0, \rho(A)^{-2})$, we can choose $\beta$ small enough such that,

$$\|Q\|_{\mathrm{op}} f(c, A) < \frac{1}{4\beta^2} \sigma_{\min}(R)$$

implying that the optimal controller $K_{\star,\gamma}$ for the discounted problem cannot be stabilizing for $(A, B)$. $\square$

### A.4 Auxiliary results for linear systems

**Proposition A.4.** *Let $\sqrt{\gamma_1}(A+BK)$ be a stable matrix and define $P_1 := \mathsf{dlyap}(\sqrt{\gamma_1}(A+BK), Q+K^\top RK)$, then for $c$ defined as*

$$c := \frac{1}{8\|P_1\|_{\mathrm{op}}^4} + 1,$$

*the following holds for $\gamma_2 := c^2\gamma_1$ and $P_2 := \mathsf{dlyap}(\sqrt{\gamma_2}(A+BK), Q+K^\top RK)$:*

$$\mathrm{tr}\,[P_2 - P_1] \;\leq\; \mathrm{tr}\,[P_1].$$

*Proof.* The proof is a direct consequence of Proposition C.7 in Perdomo et al. [2021]. In particular, we use their results for the trace norm and use the following substitutions,

$$
\begin{aligned}
A_1 &\leftarrow \sqrt{\gamma_1}(A+BK) & \Sigma &\leftarrow Q+K^\top RK \\
A_2 &\leftarrow \sqrt{\gamma_2}(A+BK) & \alpha &\leftarrow 1/2
\end{aligned}
$$

where $A_1, A_2, \Sigma$, and $\alpha$ are defined as in Perdomo et al. [2021]. Note that for $c$ satisfying,

$$c \;\leq\; \frac{1}{8\|\sqrt{\gamma_1}(A+BK)\|_{\mathrm{op}}} \min\left\{\frac{1}{\|P_1\|_{\mathrm{op}}^{3/2}};\; \frac{\mathrm{tr}\,[P_1]}{d_x\|P_1\|_{\mathrm{op}}^{7/2}}\right\} + 1$$

we get that,

$$\|A_1 - A_2\|_{\mathrm{op}}^2 = \gamma_1(c-1)^2\|A+BK\|_{\mathrm{op}}^2 \;\leq\; \frac{1}{64\|P_1\|_{\mathrm{op}}^3} = \frac{\alpha^2}{16\|P_1\|_{\mathrm{op}}^3}.$$

Therefore, Proposition C.7 states that, for $\mathcal{C} := \mathrm{tr}\left[P_1^{-1/2}(Q+K^\top RK)P_1^{-1/2}\right] \;\leq\; \mathrm{tr}\,[I] = d_x$,

$$
\begin{aligned}
\mathrm{tr}\,[P_2 - P_1] &\;\leq\; 8\mathcal{C}\sqrt{\gamma_1}(c-1)\|A+BK\|_{\mathrm{op}}\|P_1\|_{\mathrm{op}}^{7/2} \\
&\;\leq\; \mathrm{tr}\,[P_1].
\end{aligned}
$$

Lastly, noting that,

$$P_1 = \mathsf{dlyap}(\sqrt{\gamma_1}(A+BK), Q+K^\top RK) \succeq \gamma(A+BK)^\top(A+BK)$$

we have that $\|\sqrt{\gamma_1}(A+BK)\|_{\mathrm{op}} \;\leq\; \|P_1\|_{\mathrm{op}}^{1/2}$ and $\mathrm{tr}\,[P_1] \;\geq\; d_x$. Therefore, since $\|P_1\|_{\mathrm{op}} \;\geq\; 1$, in order to apply Proposition C.7 from Perdomo et al. [2021] it suffices for $c$ to satisfy,

$$c \;\leq\; \frac{1}{8\|P_1\|_{\mathrm{op}}^4} + 1.$$

$\qquad\square$

**Lemma A.5** (Lemma D.9 in Perdomo et al. [2021]). *Let $A$ be a stable matrix, $Q \succeq I$, and define $P := \mathsf{dlyap}(A, Q)$. Then, for all $j \;\geq\; 0$,*

$$(A^\top)^j A^j \;\preceq\; (A^\top)^j P A^j \;\preceq\; P\left(1 - \frac{1}{\|P\|_{\mathrm{op}}}\right)^j.$$

**Lemma A.6.** *Let $A$ be a stable matrix and define $P := \mathsf{dlyap}(A, Q)$ where $Q \succeq I$. Then, for any matrix $\Delta$ such that $\|\Delta\|_{\mathrm{op}} \;\leq\; \frac{1}{6\|P\|_{\mathrm{op}}^2}$, it holds that for all $j \;\geq\; 0$*

$$\left((A+\Delta)^\top\right)^j P(A+\Delta)^j \;\preceq\; P\left(1 - \frac{1}{2\|P\|_{\mathrm{op}}}\right)^j.$$

*Proof.* Expanding out, we have that

$$
\begin{aligned}
(A+\Delta)^\top P(A+\Delta) &= A^\top PA + A^\top P\Delta + \Delta^\top PA + \Delta^\top P\Delta \\
&\preceq P\left(1 - \frac{1}{\|P\|_{\mathrm{op}}}\right) + \|A^\top P\Delta + \Delta^\top PA + \Delta^\top P\Delta\|_{\mathrm{op}}I,
\end{aligned}
$$

where in the second line we have used properties of the Lyapunov function, Lemma A.5. Next, we observe that

$$\|\Delta^\top P A\|_{\mathrm{op}} \leq \|\Delta^\top P^{1/2}\|_{\mathrm{op}}\|P^{1/2}A\|_{\mathrm{op}} \leq \|\Delta^\top P^{1/2}\|_{\mathrm{op}}\|P^{1/2}\|_{\mathrm{op}} \leq \|\Delta\|_{\mathrm{op}}\|P\|_{\mathrm{op}},$$

where we have again used Lemma A.5 to conclude that $A^\top P A \preceq P$. Note that the exact same calculation holds for $\|A^\top P \Delta\|_{\mathrm{op}}$. Hence, we can conclude that for $\Delta$ such that $\|\Delta\|_{\mathrm{op}} \leq 1$,

$$(A + \Delta)^\top P(A + \Delta) \preceq P\left(1 - \frac{1}{\|P\|_{\mathrm{op}}}\right) + 3\|P\|_{\mathrm{op}}\|\Delta\|_{\mathrm{op}} \cdot I.$$

Using the fact that, $P \succeq I$ and that $\|\Delta\|_{\mathrm{op}} \leq 1/(6\|P\|_{\mathrm{op}}^2)$, we get that,

$$3\|P\|_{\mathrm{op}}\|\Delta\|_{\mathrm{op}} \cdot I \preceq \frac{1}{2\|P\|_{\mathrm{op}}}P,$$

which finishes the proof. $\qquad\square$

# B   Deferred Proofs and Analysis for the Nonlinear Setting

**Establishing Lyapunov functions.** Our analysis for nonlinear systems begins with the observation that any state-feedback controller $K$ which achieves finite cost on the $\gamma$-discounted LQR problem has an associated value function $P_{K,\gamma}$ which can be used as a Lyapunov function for the $\sqrt{\gamma}$-damped nonlinear dynamics, for small enough initial states. We present the proof of this result in Appendix B.3.

**Lemma B.1.** *Let $J_{\mathrm{lin}}(K \mid \gamma) < \infty$. Then, for all $\mathbf{x} \in \mathbb{R}_x^d$ such that,*

$$\mathbf{x}^\top P_{K,\gamma}\mathbf{x} \leq \frac{r_{\mathrm{nl}}^2}{4\beta_{\mathrm{nl}}^2\|P_{K,\gamma}\|_{\mathrm{op}}^3},$$

*the following inequality holds:*

$$\gamma \cdot G_{\mathrm{nl}}(\mathbf{x}, K\mathbf{x})^\top P_{K,\gamma}G_{\mathrm{nl}}(\mathbf{x}, K\mathbf{x}) \leq \mathbf{x}^\top P_{K,\gamma}\mathbf{x} \cdot \left(1 - \frac{1}{2\|P_{K,\gamma}\|_{\mathrm{op}}}\right).$$

Using this observation, we can then show that any controller which has finite discounted LQR cost is exponentially stabilizing over states in a sufficiently small region of attraction.

**Lemma B.2.** *Assume $J_{\mathrm{lin}}(K \mid \gamma) < \infty$ and define $\{\mathbf{x}_{t,\mathrm{nl}}\}_{t=0}^\infty$ be the sequence of states generated according to $\mathbf{x}_{t+1,\mathrm{nl}} = \sqrt{\gamma}G_{\mathrm{nl}}(\mathbf{x}_{t,\mathrm{nl}}, K\mathbf{x}_{t,\mathrm{nl}})$ where $\mathbf{x}_{t,\mathrm{nl}} = \mathbf{x}_0$. If $\mathbf{x}_0$ is such that,*

$$V_0 := \mathbf{x}_0^\top P_{K,\gamma}\mathbf{x}_0 \leq \frac{r_{\mathrm{nl}}^2}{4\beta_{\mathrm{nl}}^2\|P_{K,\gamma}\|_{\mathrm{op}}^3},$$

*then for all $t \geq 0$ and for $V_0$ defined as above,*

  *a) The norm of the state $\|\mathbf{x}_{t,\mathrm{nl}}\|^2$ is bounded by*

$$\|\mathbf{x}_{t,\mathrm{nl}}\|^2 \leq \mathbf{x}_{t,\mathrm{nl}}^\top P_{K,\gamma}\mathbf{x}_{t,\mathrm{nl}} \leq V_0\left(1 - \frac{1}{2\|P_{K,\gamma}\|_{\mathrm{op}}}\right)^t. \tag{B.1}$$

  *b) The norms of $f_{\mathrm{nl}}(\mathbf{x}_{t,\mathrm{nl}}, K\mathbf{x}_{t,\mathrm{nl}})$ and $\nabla f_{\mathrm{nl}}(\mathbf{x}_{t,\mathrm{nl}}, K\mathbf{x}_{t,\mathrm{nl}})$ are bounded by*

$$\|f_{\mathrm{nl}}(\mathbf{x}_{t,\mathrm{nl}}, K\mathbf{x}_{t,\mathrm{nl}})\| \leq \beta_{\mathrm{nl}}(1 + \|K\|_{\mathrm{op}}^2)V_0\left(1 - \frac{1}{2\|P_{K,\gamma}\|_{\mathrm{op}}}\right)^t. \tag{B.2}$$

$$\|\nabla f_{\mathrm{nl}}(\mathbf{x}_{t,\mathrm{nl}}, K\mathbf{x}_{t,\mathrm{nl}})\|_{\mathrm{op}} \leq \beta(1 + \|K\|_{\mathrm{op}})V_0^{1/2}\left(1 - \frac{1}{2\|P_{K,\gamma}\|_{\mathrm{op}}}\right)^{t/2}. \tag{B.3}$$

*Proof.* The proof of $(a)$ follows by repeatedly applying Lemma B.1. Part $(b)$ follows from the first after using Lemma 3.1. The statement of the lemma in the main body follows from using $\|P_{K,\gamma}\|_{\mathrm{op}} \leq \mathrm{tr}\,[P_{K,\gamma}] = J_{\mathrm{lin}}(K \mid \gamma)$ and simplifying $(1 - x)^t \leq \exp(-tx)$. $\qquad\square$

**Relating $G_{\mathrm{nl}}$ to its Jacobian Linearization.** Having established how any controller that achieves finite LQR cost is guaranteed to be stabilizing for the nonlinear system, we now go a step further and illustrate how this stability guarantee can be used to prove that the difference in costs and gradients between $G_{\mathrm{nl}}$ and its Jacobian linearization are guaranteed to be small.

**Proposition 3.3 (restated).** *Assume $J_{\mathrm{lin}}(K \mid \gamma) < \infty$. Then,*

a) *If $r \leq \frac{r_{\mathrm{nl}}}{2\beta_{\mathrm{nl}}\|P_{K,\gamma}\|_{\mathrm{op}}^2}$, then $\left| J_{\mathrm{nl}}(K \mid \gamma, r) - J_{\mathrm{lin}}(K \mid \gamma) \right| \leq 8d_x\beta_{\mathrm{nl}}\|P_{K,\gamma}\|_{\mathrm{op}}^4 \cdot r$.*

b) *If $r \leq \frac{r_{\mathrm{nl}}}{12\beta_{\mathrm{nl}}\|P_{K,\gamma}\|_{\mathrm{op}}^{5/2}}$, then,*

$$\|\nabla_K J_{\mathrm{nl}}(K \mid \gamma, r) - \nabla_K J_{\mathrm{lin}}(K \mid \gamma)\|_{\mathrm{F}} \leq 48d_x\beta_{\mathrm{nl}}(1 + \|B\|_{\mathrm{op}})\|P_{K,\gamma}\|_{\mathrm{op}}^7 r.$$

*Proof.* Due to our assumption on $r = \|\mathbf{x}_0\|$, we have that,

$$\mathbf{x}_0^\top P_{K,\gamma}\mathbf{x}_0 \leq \|P_{K,\gamma}\|_{\mathrm{op}}\|\mathbf{x}_0\|^2 \leq \frac{1}{4\beta_{\mathrm{nl}}^2\|P_{K,\gamma}\|_{\mathrm{op}}^3}.$$

Therefore, we can apply Lemma B.5 to conclude that,

$$\left| J_{\mathrm{nl}}(K \mid \gamma, \mathbf{x}_0) - J_{\mathrm{lin}}(K \mid \gamma, \mathbf{x}_0) \right| \leq 8\beta_{\mathrm{nl}}\|\mathbf{x}_0\|^3\|P_{K,\gamma}\|_{\mathrm{op}}^4.$$

Next, we multiply both sides by $d_x/r^2$, take expectations, and apply Jensen's inequality to get that,

$$\left| \frac{d_x}{r^2}\mathbb{E}_{\mathbf{x}_0 \sim r \cdot \mathcal{S}^{d_x-1}} J_{\mathrm{nl}}(K \mid \gamma, \mathbf{x}_0) - \frac{d_x}{r^2}\mathbb{E}_{\mathbf{x}_0 \sim r \cdot \mathcal{S}^{d_x-1}} J_{\mathrm{lin}}(K \mid \gamma, \mathbf{x}_0) \right| \leq 8\frac{d_x}{r^2}\beta_{\mathrm{nl}}\|P_{K,\gamma}\|_{\mathrm{op}}^4\mathbb{E}_{\mathbf{x}_0 \sim r \cdot \mathcal{S}^{d_x-1}}\|\mathbf{x}_0\|^3.$$

Given our definitions of the linear objective in Definition 2.1, we have that,

$$\frac{d_x}{r^2}\mathbb{E}_{\mathbf{x}_0 \sim r \cdot \mathcal{S}^{d_x-1}} J_{\mathrm{lin}}(K \mid \gamma, \mathbf{x}_0) = J_{\mathrm{lin}}(K \mid \gamma),$$

for all $r > 0$. Therefore, we can rewrite the inequality above as,

$$\left| J_{\mathrm{nl}}(K \mid \gamma, r) - J_{\mathrm{lin}}(K \mid \gamma) \right| \leq 8d_x\beta_{\mathrm{nl}}\|P_{K,\gamma}\|_{\mathrm{op}}^4 \cdot r.$$

The second part of the proposition uses the same argument as part a, but this time employing Lemma B.6 to bound the difference in gradients (pointwise). $\qquad\square$

In short, this previous lemma states that if the cost on the linear system is bounded, then the costs and gradients between the nonlinear objective and its Jacobian linearization are close. We can also prove the analogous statement which establishes closeness while assuming that the cost on the nonlinear system is bounded.

**Lemma B.3.** *Let $\alpha > 1$ be such that $80d_x^2 J_{\mathrm{nl}}(K \mid \gamma, r) \leq \alpha$.*

1. *If $r \leq \frac{r_{\mathrm{nl}}^2}{64\beta_{\mathrm{nl}}\alpha^2(1+\|K\|_{\mathrm{op}})}$, then $|J_{\mathrm{nl}}(K \mid \gamma, r) - J_{\mathrm{lin}}(K \mid \gamma)| \leq 8d_x\beta_{\mathrm{nl}}\alpha^4 r$.*

2. *If $r \leq \frac{1}{12\beta_{\mathrm{nl}}\alpha^{5/2}}$, then $\|\nabla_K J_{\mathrm{nl}}(K \mid \gamma, r) - \nabla_K J_{\mathrm{lin}}(K \mid \gamma)\|_{\mathrm{F}} \leq 48d_x\beta_{\mathrm{nl}}(1+\|B\|_{\mathrm{op}})\alpha^7 \cdot r$.*

*Proof.* The lemma is a consequence of combining Proposition 3.3 and Proposition C.5. In particular, from Proposition C.5 if $r \leq \min\{\alpha r_{\mathrm{nl}}^2, \frac{d_x}{64\alpha^2\beta_{\mathrm{nl}}(1+\|K\|_{\mathrm{op}})}\}$, then

$$80d_x^2 J_{\mathrm{nl}}(K \mid \gamma, r) \geq \min\{J_{\mathrm{lin}}(K \mid \gamma), \alpha\}$$

However, since $\alpha \geq 80d_x^2 J_{\mathrm{nl}}(K \mid \gamma, r)$, we conclude that $80d_x^2 J_{\mathrm{nl}}(K \mid \gamma, r) \geq J_{\mathrm{lin}}(K \mid \gamma)$. Having shown that the linear cost is bounded, we can now plug in Proposition 3.3. In particular, if

$$r \leq \frac{r_{\mathrm{nl}}}{2\beta_{\mathrm{nl}}\alpha^2} \leq \frac{r_{\mathrm{nl}}}{2\beta_{\mathrm{nl}} J_{\mathrm{lin}}(K \mid \gamma)^2}$$

then, Proposition 3.3 states that

$$|J_{\mathrm{nl}}(K \mid \gamma, r) - J_{\mathrm{lin}}(K \mid \gamma)| \leq 8d_x\beta_{\mathrm{nl}} J_{\mathrm{lin}}(K \mid \gamma)^4 r \leq 8d_x\beta_{\mathrm{nl}}\alpha^4 r.$$

To prove the second part of the statement, we again use Proposition 3.3. In particular, since

$$r \leq \frac{1}{12\beta_{\mathrm{nl}}\alpha^{5/2}} \leq \frac{1}{12\beta_{\mathrm{nl}} J_{\mathrm{lin}}(K \mid \gamma)^{5/2}}$$

we can hence conclude that

$$\|\nabla_K J_{\mathrm{nl}}(K \mid \gamma, r) - \nabla_K J_{\mathrm{lin}}(K \mid \gamma)\|_{\mathrm{F}} \leq 48d_x\beta_{\mathrm{nl}}(1 + \|B\|_{\mathrm{op}})\alpha^7 \cdot r.$$

$\qquad\square$

## B.1 Discount annealing on nonlinear systems: Proof of Theorem 2

As in Theorem 1, we first prove parts $c)$ and $d)$ by induction and then prove parts $a)$ and $b)$ separately.

### B.1.1 Proof of $c)$ and $d)$

**Base case.** As before, at each iteration $t$ of discount annealing, policy gradients is initialized at $K_{t,0} := K_t$ and computes updates according to,

$$K_{t,j+1} = K_{t,j} - \eta \cdot \varepsilon\text{-}\texttt{Grad}\left(J_{\mathrm{nl}}(\cdot \mid \gamma_t, r_\star), K_{t,j}\right).$$

To prove correctness, we show that the noisy gradients on the nonlinear system are close to the true gradients on the *linear* system. That is,

$$\|\varepsilon\text{-}\texttt{Grad}\left(J_{\mathrm{nl}}(\cdot \mid \gamma_t, r_\star), K_{t,j}\right) - \nabla J_{\mathrm{lin}}(K_{t,j} \mid \gamma_t)\|_{\mathrm{F}} \leq \mathcal{C}_{\mathrm{pg},t} d_x \quad \forall j \geq 0, \tag{B.4}$$

where $\mathcal{C}_{\mathrm{pg},t} = \mathrm{poly}(1/\|A\|_{\mathrm{op}}, 1/\|B\|_{\mathrm{op}}, 1/J_{\mathrm{lin}}(K_t \mid \gamma_t))$ is again a fixed polynomial from Lemma A.3.

Consider the first iteration of discount annealing, by choice of $\gamma_0$, we have that $J_{\mathrm{lin}}(K_0 \mid \gamma_0) < \infty$. Therefore, by Proposition 3.3 if

$$r_\star \leq \min\left\{\frac{r_{\mathrm{nl}}}{12\beta_{\mathrm{nl}}J_{\mathrm{lin}}(K_{0,0} \mid \gamma_0)}, \frac{\mathcal{C}_{\mathrm{pg},0}}{100\beta_{\mathrm{nl}}(1 + \|B\|_{\mathrm{op}})J_{\mathrm{lin}}(K_{0,0} \mid \gamma_0)^7}\right\}$$

it must hold that $\|\nabla J_{\mathrm{nl}}(K_{0,0} \mid \gamma_0, r_\star) - \nabla J_{\mathrm{lin}}(K_{0,0} \mid \gamma_0)\|_{\mathrm{F}} \leq .5\mathcal{C}_{\mathrm{pg},0} d_x$. Likewise, if we choose the tolerance parameter $\varepsilon \leq .5\mathcal{C}_{\mathrm{pg},0} d_x$ in $\varepsilon\text{-}\texttt{Grad}$ then we have that

$$\|\varepsilon\text{-}\texttt{Grad}\left(J_{\mathrm{nl}}(\cdot \mid \gamma_0, r_\star), K_{0,0}\right) - J_{\mathrm{nl}}(K_{0,0} \mid \gamma_0, r_\star)\|_{\mathrm{F}} \leq .5\mathcal{C}_{\mathrm{pg},0} d_x.$$

By the triangle inequality, the inequality in Eq. (B.4) holds for $t = 0$ and $j = 0$. However, because Lemma A.3 shows that policy gradients is a descent method, that is $J_{\mathrm{lin}}(K_{0,j} \mid \gamma_0) \leq J_{\mathrm{lin}}(K_{0,0} \mid \gamma_0)$ for all $j \geq 0$, Eq. (B.4) also holds for all $j \geq 0$ for the same choice of $r_\star$ and tolerance parameter for $\varepsilon\text{-}\texttt{Grad}$. By guarantee of Lemma A.3, for $t = 0$, policy gradients achieves the guarantee outlined in Step 2 using at most $\mathrm{poly}(\|A\|_{\mathrm{op}}, \|B\|_{\mathrm{op}}, J_{\mathrm{lin}}(K_0 \mid \gamma_0))$ many queries.

To prove that random search achieves the guarantee outlines in Step 4 at iteration 0 of discount annealing, we appeal to Lemma C.7. In particular, we instantiate the lemma with $f \leftarrow J_{\mathrm{nl}}(K_1 \mid \cdot, r_\star)$, $f_1 \leftarrow 8J_{\mathrm{nl}}(K_1 \mid \gamma_0, r_\star)$, $f_2 \leftarrow 2.5J_{\mathrm{nl}}(K_1 \mid \gamma_0, r_\star)$. As before, the algorithm requires values $\overline{f}_1 \in [2.9J_{\mathrm{nl}}(K_1 \mid \gamma_0, r_\star), 3J_{\mathrm{nl}}(K_1 \mid \gamma_0)]$ and $\overline{f}_2 \in [6J_{\mathrm{nl}}(K_1 \mid \gamma_0, r_\star), 6.1J_{\mathrm{nl}}(K_1 \mid \gamma_0)]$. These can be estimated via two calls to $\varepsilon\text{-}\texttt{Eval}$ with tolerance parameter $.01d_x$.

To show the lemma applies we only need to lower bound the width of feasible $\gamma$ such that

$$2.9J_{\mathrm{nl}}(K_1 \mid \gamma_0, r_\star) \leq J_{\mathrm{nl}}(K_1 \mid \gamma, r_\star) \leq 6.1J_{\mathrm{nl}}(K_1 \mid \gamma_0, r_\star) \tag{B.5}$$

From the guarantee from policy gradients, we know that $J_{\mathrm{lin}}(K_1 \mid \gamma_0) \leq 2\mathrm{tr}\,[P_\star]$. Furthermore, from the proof of Theorem 1, we know that there exists $\gamma'', \gamma' \in [0,1]$ satisfying, $\gamma'' - \gamma' \geq \frac{1}{5200\mathrm{tr}[P_\star]^4}\gamma_0$, such that for all $\gamma \in [\gamma', \gamma'']$

$$3J_{\mathrm{lin}}(K_1 \mid \gamma_0) \leq J_{\mathrm{lin}}(K_1 \mid \gamma) \leq 6J_{\mathrm{lin}}(K_1 \mid \gamma_0). \tag{B.6}$$

To finish the proof of correctness, we show that any $\gamma$ that satisfies Eq. (B.6) must also satisfy Eq. (B.5). In particular, since $J_{\mathrm{lin}}(K_1 \mid \gamma_0) \leq 2\mathrm{tr}\,[P_\star]$ and $J_{\mathrm{lin}}(K_1 \mid \gamma) \leq 12\mathrm{tr}\,[P_\star]$ for

$$r_\star \leq \min\left\{\frac{r_{\mathrm{nl}}}{2\beta_{\mathrm{nl}}(12\mathrm{tr}\,[P_\star])^2}, \frac{.01}{8\beta_{\mathrm{nl}}(12\mathrm{tr}\,[P_\star])^4}\right\},$$

it holds that $|J_{\mathrm{nl}}(K_1 \mid \gamma_0, r_\star) - J_{\mathrm{lin}}(K_1 \mid \gamma_0)| \leq .01d_x$ and $|J_{\mathrm{nl}}(K_1 \mid \gamma, r_\star) - J_{\mathrm{lin}}(K_1 \mid \gamma)| \leq .01d_x$. Using these two inequalities along with Eq. (B.6) implies that Eq. (B.5) must also hold. Therefore, the width of the feasible region is at least $1/(5200\mathrm{tr}\,[P_\star]^4)$ and random search must return a discount factor using at most 1 over this many iterations by Lemma C.7.

**Inductive step.** To show that policy gradients converges, we can use the exact same argument as when arguing the base case where instead of referring to $J_{\text{lin}}(K_0 \mid \gamma_0)$ and $J_{\text{nl}}(K_0 \mid \gamma_0, r_\star)$ we use $J_{\text{lin}}(K_t \mid \gamma_t)$ and $J_{\text{nl}}(K_t \mid \gamma_t, r_\star)$. In particular, we can reuse the same argument as in the base case, but need to ensure that is that $\sup_{t \geq 1} J_{\text{lin}}(K_t \mid \gamma_t)$ is uniformly bounded.

To prove this, from the inductive hypothesis on the correctness of binary search at previous iterations, we know that $J_{\text{nl}}(K_t \mid \gamma_t) \leq 8 J_{\text{nl}}(K_t \mid \gamma_{t-1}, r_\star)$. Again by the inductive hypothesis, at time step $t-1$ policy gradients achieves the desired guarantee from Step 2, implying that $J_{\text{lin}}(K_t \mid \gamma_{t-1}) \leq 2\text{tr}\,[P_\star]$. By choice of $r_\star$ this implies that

$$|J_{\text{lin}}(K_t \mid \gamma_{t-1}) - J_{\text{nl}}(K_t \mid \gamma_{t-1}, r_\star)| \leq .01 d_x$$

and hence $J_{\text{nl}}(K_t \mid \gamma_t) \leq 20\text{tr}\,[P_\star]$. Now, we can apply [Lemma B.3] to conclude that for $\alpha := (80 \times 20) d_x^2 \text{tr}\,[P_\star]$ and

$$r_\star \leq \min\left\{ \frac{r_{\text{nl}}^2}{64 \beta_{\text{nl}} \alpha^2 (1 + \|K\|_{\text{op}})}, \frac{.01}{8 \beta_{\text{nl}} \alpha^4} \right\}$$

it holds that $|J_{\text{nl}}(K_t \mid \gamma_t, r_\star) - J_{\text{lin}}(K_t \mid \gamma_t)| \leq .01 d_x$ and hence $J_{\text{lin}}(K_t \mid \gamma_t) \leq 21\text{tr}\,[P_\star]$. Therefore, $\sup_{t \geq 1} J_{\text{lin}}(K_t \mid \gamma_t) \leq 21\text{tr}\,[P_\star]$.

Similarly, the inductive step for the random search procedure follows from noting that the exact same argument can be repeated by replacing $J_{\text{nl}}(K_1 \mid \gamma_0, r_\star)$ with $J_{\text{nl}}(K_t \mid \gamma_{t-1})$ and $J_{\text{lin}}(K_1 \mid \gamma_0)$ with $J_{\text{lin}}(K_t \mid \gamma_{t-1})$ since (by the inductive hypothesis) $J_{\text{lin}}(K_t \mid \gamma_{t-1}) \leq 2\text{tr}\,[P_\star]$.

### B.1.2 Proof of $a$)

By the guarantee from Step 2, the algorithm returns a $\widehat{K}$ which satisfies the following guarantee on the *linear* system:

$$J_{\text{lin}}(\widehat{K} \mid 1) - \min_K J_{\text{lin}}(K \mid 1) \leq d_x.$$

Therefore, $J_{\text{lin}}(\widehat{K} \mid 1) \leq 2\text{tr}\,[P_\star]$. Now by, [Lemma B.2], the following holds,

$$\|\mathbf{x}_{t,\text{nl}}\|^2 \leq \|\widehat{P}\|_{\text{op}} \|\mathbf{x}_0\|^2 \left( 1 - \frac{1}{2\|\widehat{P}\|_{\text{op}}} \right)^t$$

$$\leq 2\text{tr}\,[P_\star] \|\mathbf{x}_0\|^2 \exp\left( -\frac{t}{4\text{tr}\,[P_\star]} \right),$$

for $\widehat{P} := \text{dlyap}(A + B\widehat{K}, Q + \widehat{K}^\top R \widehat{K})$ and all $\mathbf{x}_0$ such that $\|\mathbf{x}_0\| \leq r_{\text{nl}}/(8 \beta_{\text{nl}} \text{tr}\,[P_\star]^2)$.

### B.1.3 Proof of $b$)

The bound for the number of subproblems solved by the discount annealing algorithms is similar to that of the linear case. The crux of the argument for part $b$) is to show that any $\gamma' \in [\gamma_t, 1]$ such that

$$2.5 J_{\text{nl}}(K_{t+1} \mid \gamma_t, r_\star) \leq J_{\text{nl}}(K_{t+1} \mid \gamma', r_\star) \leq 8 J_{\text{nl}}(K_{t+1} \mid \gamma_t, r_\star)$$

the following inequality must also hold: $J_{\text{lin}}(K_{t+1} \mid \gamma') \geq 2 J_{\text{lin}}(K_{t+1} \mid \gamma_t)$. Once we've lower bounded the cost on the *linear* system, we can repeat the same argument as in Theorem 1. Since the cost on the linear system in nondecreasing in $\gamma$, it must be the case that $\gamma'$ satisfies

$$\gamma' \geq \left( \frac{1}{8\|P_{K_{t+1}, \gamma_t}\|_{\text{op}}^4} + 1 \right)^2 \gamma_t \geq \left( \frac{1}{128\text{tr}\,[P_\star]^4} + 1 \right)^2 \gamma_t.$$

Here, we have again we have used the calculation that,

$$\|P_{K_{t+1}, \gamma_t}\|_{\text{op}}^4 \leq \text{tr}\,\left[ P_{K_{t+1}, \gamma_t} \right]^4 = J_{\text{lin}}(K_{t+1} \mid \gamma_t)^4 \leq (\min_K J_{\text{lin}}(K \mid \gamma_t) + d_x)^4 \leq 16\text{tr}\,[P_\star]^4,$$

which follows from the guarantee that (for our choice of $r_\star$) policy gradients on the nonlinear system converges to a near optimal controller for the system's Jacobian linearization. Hence, as in the linear

setting, we conclude that $\gamma_t \geq (1/(128\mathrm{tr}\,[P_\star]^4)+1)^{2t}\gamma_0$ and discount annealing achieves the same rate as for linear systems.

We now focus on establishing that $J_{\mathrm{lin}}(K_{t+1} \mid \gamma') \geq 2J_{\mathrm{lin}}(K_{t+1} \mid \gamma_t)$. By guarantee from policy gradients, we have that $J_{\mathrm{lin}}(K_{t+1} \mid \gamma_t) \leq \min_K J_{\mathrm{lin}}(K \mid \gamma_t) + d_x \leq 2\mathrm{tr}\,[P_\star]$. Therefore, by [Proposition 3.3](#) since

$$r_\star \leq \frac{.01 r_{\mathrm{nl}}}{8 \times 2^4 \cdot \mathrm{tr}\,[P_\star]^4 \beta_{\mathrm{nl}}} \leq \min\left\{\frac{r_{\mathrm{nl}}}{2\beta_{\mathrm{nl}}\|P_{K_{t+1},\gamma_t}\|_{\mathrm{op}}^2}, \frac{.01}{8\beta_{\mathrm{nl}}\|P_{K_{t+1},\gamma_t}\|_{\mathrm{op}}^4}\right\}$$

it holds that $|J_{\mathrm{lin}}(K_{t+1} \mid \gamma_t) - J_{\mathrm{nl}}(K_{t+1} \mid \gamma_t, r_\star)| \leq .01 d_x$.

Next, we show that $|J_{\mathrm{lin}}(K_{t+1} \mid \gamma') - J_{\mathrm{nl}}(K_{t+1} \mid \gamma', r_\star)|$ is also small. In particular, the previous statement, together with the guarantee from Step 4, implies that

$$J_{\mathrm{nl}}(K_{t+1} \mid \gamma') \leq 8J_{\mathrm{nl}}(K_{t+1} \mid \gamma_t) \leq 8(J_{\mathrm{lin}}(K_{t+1} \mid \gamma_t) + .01 d_x) \leq 8.08 J_{\mathrm{lin}}(K_{t+1} \mid \gamma_t) \leq 16.16\mathrm{tr}\,[P_\star].$$

Therefore, for $\alpha := (80 \times 17)\mathrm{tr}\,[P_\star]\,d_x^2$, [Lemma B.3](#) implies that if,

$$r \leq \frac{r_{\mathrm{nl}}^2}{64\beta_{\mathrm{nl}}\alpha^2(1 + \|K_{t+1}\|_{\mathrm{op}})},$$

it holds that $|J_{\mathrm{nl}}(K_{t+1} \mid \gamma', r) - J_{\mathrm{lin}}(K_{t+1} \mid \gamma')| \leq 8d_x\beta_{\mathrm{nl}}\alpha^4 r$. Hence, if $r \leq .01/(8\beta_{\mathrm{nl}}\alpha^4)$ we get that $|J_{\mathrm{lin}}(K_{t+1} \mid \gamma') - J_{\mathrm{nl}}(K_{t+1} \mid \gamma', r_\star)| \leq .01 d_x$. Using again the fact that $\min\{J_{\mathrm{lin}}(K \mid \gamma), J_{\mathrm{nl}}(K \mid \gamma, r)\} \geq d_x$ for all $K, \gamma, r$ we hence conclude that

$$\begin{aligned}
J_{\mathrm{lin}}(K_{t+1} \mid \gamma') &\geq .99 J_{\mathrm{nl}}(K_{t+1} \mid \gamma', r_\star) \\
&\geq 2.5 \times .99 \cdot J_{\mathrm{nl}}(K_{t+1} \mid \gamma_t, r_\star) \\
&\geq 2.5 \times .99^2 \cdot J_{\mathrm{lin}}(K_{t+1} \mid \gamma_t),
\end{aligned}$$

which finished the proof of the fact that $J_{\mathrm{lin}}(K_{t+1} \mid \gamma') \geq 2J_{\mathrm{lin}}(K_{t+1} \mid \gamma_t)$.

## B.2 Relating costs and gradients to the linear system: Proof of [Proposition 3.3](#)

In order to relate the properties of the nonlinear system to its Jacobian linearization, we employ the following version of the performance difference lemma.

**Lemma B.4** (Performance Difference). *Assume $J_{\mathrm{lin}}(K \mid \gamma) < \infty$ and define $\{\mathbf{x}_{t,\mathrm{nl}}\}_{t=0}^\infty$ be the sequence of states generated according to $\mathbf{x}_{t+1,\mathrm{nl}} = \sqrt{\gamma}G_{\mathrm{nl}}(\mathbf{x}_{t,\mathrm{nl}}, K\mathbf{x}_{t,\mathrm{nl}})$ where $\mathbf{x}_{t,\mathrm{nl}} = \mathbf{x}_0$. Then,*

$$J_{\mathrm{nl}}(K \mid \gamma, \mathbf{x}) - J_{\mathrm{lin}}(K \mid \gamma, \mathbf{x}) = \sum_{t=0}^\infty \gamma \cdot f_{\mathrm{nl}}(\mathbf{x}_{t,\mathrm{nl}}, K\mathbf{x}_{t,\mathrm{nl}})^\top P_{K,\gamma}\left(G_{\mathrm{nl}}(\mathbf{x}_{t,\mathrm{nl}}, K\mathbf{x}_{t,\mathrm{nl}}) + G_{\mathrm{lin}}(\mathbf{x}_{t,\mathrm{nl}}, K\mathbf{x}_{t,\mathrm{nl}})\right).$$

*Proof.* From the definition of the relevant objectives, and [Fact A.2](#), we get that,

$$\begin{aligned}
J_{\mathrm{nl}}(K \mid \gamma, \mathbf{x}) - J_{\mathrm{lin}}(K \mid \gamma, \mathbf{x}) &= \left(\sum_{t=0}^\infty \mathbf{x}_{t,\mathrm{nl}}^\top(Q + K^\top RK)\mathbf{x}_{t,\mathrm{nl}}\right) - \mathbf{x}^\top P_{K,\gamma}\mathbf{x} \\
&= \left(\sum_{t=0}^\infty \mathbf{x}_{t,\mathrm{nl}}^\top(Q + K^\top RK)\mathbf{x}_{t,\mathrm{nl}} \pm \mathbf{x}_{t,\mathrm{nl}}^\top P_{K,\gamma}\mathbf{x}_{t,\mathrm{nl}}\right) - \mathbf{x}^\top P_{K,\gamma}\mathbf{x} \\
&= \sum_{t=0}^\infty \mathbf{x}_{t,\mathrm{nl}}^\top(Q + K^\top RK)\mathbf{x}_{t,\mathrm{nl}} + \mathbf{x}_{t+1,\mathrm{nl}}^\top P_{K,\gamma}\mathbf{x}_{t+1,\mathrm{nl}} - \mathbf{x}_{t,\mathrm{nl}}^\top P_{K,\gamma}\mathbf{x}_{t,\mathrm{nl}},
\end{aligned}$$

$$\tag{B.7}$$

where in the last line we have used the fact that $\mathbf{x}_{0,\mathrm{nl}} = \mathbf{x}$. The proof then follows from the following two observations. First, by definition of state sequence, $\mathbf{x}_{t,\mathrm{nl}}$,

$$\mathbf{x}_{t+1,\mathrm{nl}} = \sqrt{\gamma} \cdot G_{\mathrm{nl}}(\mathbf{x}_{t,\mathrm{nl}}, K\mathbf{x}_{t,\mathrm{nl}}).$$

Second, since $P_{K,\gamma}$ is the solution to a Lyapunov equation,

$$\begin{aligned}
\mathbf{x}_{t,\mathrm{nl}}^\top P_{K,\gamma}\mathbf{x}_{t,\mathrm{nl}} &= \mathbf{x}_{t,\mathrm{nl}}^\top(Q + K^\top RK)\mathbf{x}_{t,\mathrm{nl}} + \gamma \cdot \mathbf{x}_{t,\mathrm{nl}}^\top(A + BK)^\top P_{K,\gamma}(A + BK)\mathbf{x}_{t,\mathrm{nl}} \\
&= \mathbf{x}_{t,\mathrm{nl}}^\top(Q + K^\top RK)\mathbf{x}_{t,\mathrm{nl}} + \gamma \cdot G_{\mathrm{lin}}(\mathbf{x}_{t,\mathrm{nl}}, K\mathbf{x}_{t,\mathrm{nl}})^\top P_{K,\gamma}G_{\mathrm{lin}}(\mathbf{x}_{t,\mathrm{nl}}, K\mathbf{x}_{t,\mathrm{nl}}).
\end{aligned}$$

Plugging these last two lines into Eq. (B.7), we get that $J_{\mathrm{nl}}(K \mid \gamma, \mathbf{x}) - J_{\mathrm{lin}}(K \mid \gamma, \mathbf{x})$ is equal to,

$$= \sum_{t=0}^{\infty} \gamma \cdot G_{\mathrm{nl}}(\mathbf{x}_{t,\mathrm{nl}}, K\mathbf{x}_{t,\mathrm{nl}})^{\top} P_{K,\gamma} G_{\mathrm{nl}}(\mathbf{x}_{t,\mathrm{nl}}, K\mathbf{x}_{t,\mathrm{nl}}) - \gamma \cdot G_{\mathrm{lin}}(\mathbf{x}_{t,\mathrm{nl}}, K\mathbf{x}_{t,\mathrm{nl}})^{\top} P_{K,\gamma} G_{\mathrm{lin}}(\mathbf{x}_{t,\mathrm{nl}}, K\mathbf{x}_{t,\mathrm{nl}})$$

$$= \sum_{t=0}^{\infty} \gamma \cdot \left( G_{\mathrm{nl}}(\mathbf{x}_{t,\mathrm{nl}}, K\mathbf{x}_{t,\mathrm{nl}}) - G_{\mathrm{lin}}(\mathbf{x}_{t,\mathrm{nl}}, K\mathbf{x}_{t,\mathrm{nl}}) \right) P_{K,\gamma} \left( G_{\mathrm{nl}}(\mathbf{x}_{t,\mathrm{nl}}, K\mathbf{x}_{t,\mathrm{nl}}) + G_{\mathrm{lin}}(\mathbf{x}_{t,\mathrm{nl}}, K\mathbf{x}_{t,\mathrm{nl}}) \right)$$

$$= \sum_{t=0}^{\infty} \gamma \cdot f_{\mathrm{nl}}(\mathbf{x}_{t,\mathrm{nl}}, K\mathbf{x}_{t,\mathrm{nl}}) P_{K,\gamma} \left( G_{\mathrm{nl}}(\mathbf{x}_{t,\mathrm{nl}}, K\mathbf{x}_{t,\mathrm{nl}}) + G_{\mathrm{lin}}(\mathbf{x}_{t,\mathrm{nl}}, K\mathbf{x}_{t,\mathrm{nl}}) \right).$$

$\square$

### B.2.1 Establishing similarity of costs

The following lemma follows by bounding the terms appearing in the performance difference lemma.

**Lemma B.5** (Similarity of Costs). *Assume $J_{\mathrm{lin}}(K \mid \gamma) < \infty$ and define $\{\mathbf{x}_{t,\mathrm{nl}}\}_{t=0}^{\infty}$ be the sequence of states generated according to $\mathbf{x}_{t+1,\mathrm{nl}} = \sqrt{\gamma} G_{\mathrm{nl}}(\mathbf{x}_{t,\mathrm{nl}}, K\mathbf{x}_{t,\mathrm{nl}})$ where $\mathbf{x}_{t,\mathrm{nl}} = \mathbf{x}_0$. For $\mathbf{x}_0$ such that,*

$$\mathbf{x}_0^{\top} P_{K,\gamma} \mathbf{x}_0 \leq \frac{r_{\mathrm{nl}}^2}{4\beta_{\mathrm{nl}}^2 \|P_{K,\gamma}\|_{\mathrm{op}}^3},$$

*then,*

$$\left| J_{\mathrm{nl}}(K \mid \gamma, \mathbf{x}_0) - J_{\mathrm{lin}}(K \mid \gamma, \mathbf{x}_0) \right| \leq 8\beta_{\mathrm{nl}} \|\mathbf{x}_0\|^3 \|P_{K,\gamma}\|_{\mathrm{op}}^4.$$

*Proof.* We begin with the following observation. Due to our assumption on $\mathbf{x}_0$, we can use Lemma B.2 to conclude that for all $t \geq 0$, the following relationship holds for $V_0 := \mathbf{x}_{0,\mathrm{nl}}^{\top} P_{K,\gamma} \mathbf{x}_{0,\mathrm{nl}}$,

$$\|\mathbf{x}_{t,\mathrm{nl}}\|^2 \leq \mathbf{x}_{t,\mathrm{nl}}^{\top} P_{K,\gamma} \mathbf{x}_{t,\mathrm{nl}} \leq V_0 \cdot \left( 1 - \frac{1}{2\|P_{K,\gamma}\|_{\mathrm{op}}} \right)^t. \tag{B.8}$$

Now, from the performance difference lemma (Lemma B.4), we get that $J_{\mathrm{nl}}(K \mid \gamma, \mathbf{x}) - J_{\mathrm{lin}}(K \mid \gamma, \mathbf{x})$ is equal to:

$$\sum_{t=0}^{\infty} \gamma \cdot f_{\mathrm{nl}}(\mathbf{x}_{t,\mathrm{nl}}, K\mathbf{x}_{t,\mathrm{nl}})^{\top} P_{K,\gamma} \left( G_{\mathrm{nl}}(\mathbf{x}_{t,\mathrm{nl}}, K\mathbf{x}_{t,\mathrm{nl}}) + G_{\mathrm{lin}}(\mathbf{x}_{t,\mathrm{nl}}, K\mathbf{x}_{t,\mathrm{nl}}) \right).$$

Therefore, the difference $\left| J_{\mathrm{nl}}(K \mid \gamma, \mathbf{x}_0) - J_{\mathrm{lin}}(K \mid \gamma, \mathbf{x}_0) \right|$ can be bounded by,

$$\sum_{t=0}^{\infty} \underbrace{\|\sqrt{\gamma} \cdot P_{K,\gamma}^{1/2} f_{\mathrm{nl}}(\mathbf{x}_{t,\mathrm{nl}}, K\mathbf{x}_{t,\mathrm{nl}})\|}_{:=T_1} \cdot \underbrace{\|\sqrt{\gamma} \cdot P_{K,\gamma}^{1/2} \left( G_{\mathrm{nl}}(\mathbf{x}_{t,\mathrm{nl}}, K\mathbf{x}_{t,\mathrm{nl}}) + G_{\mathrm{lin}}(\mathbf{x}_{t,\mathrm{nl}}, K\mathbf{x}_{t,\mathrm{nl}}) \right)\|}_{:=T_2}. \tag{B.9}$$

Now, we analyze each of $T_1$ and $T_2$ separately. For $T_1$, by Lemma 3.1, and the fact that $\gamma \leq 1$,

$$\|\sqrt{\gamma} \cdot P_{K,\gamma}^{1/2} f_{\mathrm{nl}}(\mathbf{x}_{t,\mathrm{nl}}, K\mathbf{x}_{t,\mathrm{nl}})\| \leq \|P_{K,\gamma}\|_{\mathrm{op}}^{1/2} \|f_{\mathrm{nl}}(\mathbf{x}_{t,\mathrm{nl}}, K\mathbf{x}_{t,\mathrm{nl}})\|$$

$$= \|P_{K,\gamma}\|_{\mathrm{op}}^{1/2} \cdot \beta_{\mathrm{nl}}(\|\mathbf{x}_{t,\mathrm{nl}}\|^2 + \|K\mathbf{x}_{t,\mathrm{nl}}\|^2)$$

$$\leq \|P_{K,\gamma}\|_{\mathrm{op}}^{1/2} \beta_{\mathrm{nl}} \cdot (\|K\|_{\mathrm{op}}^2 + 1) V_0 \cdot \left( 1 - \frac{1}{2\|P_{K,\gamma}\|_{\mathrm{op}}} \right)^t,$$

where in the last line, we have used our assumption on the initial state and Lemma B.2. Moving onto $T_2$, we use the triangle inequality to get that

$$T_2 \leq \|\sqrt{\gamma} \cdot P_{K,\gamma}^{1/2} G_{\mathrm{nl}}(\mathbf{x}_{t,\mathrm{nl}}, K\mathbf{x}_{t,\mathrm{nl}})\| + \|\sqrt{\gamma} \cdot P_{K,\gamma}^{1/2} G_{\mathrm{lin}}(\mathbf{x}_{t,\mathrm{nl}}, K\mathbf{x}_{t,\mathrm{nl}})\|.$$

For the second term above, by Lemma A.5 and Lemma B.2, we have that

$$\|\sqrt{\gamma} \cdot P_{K,\gamma}^{1/2} G_{\lin}(\mathbf{x}_{t,\nl}, K\mathbf{x}_{t,\nl})\| = \sqrt{\gamma \cdot \mathbf{x}_{t,\nl}^{\top}(A+BK)^{\top} P_{K,\gamma}(A+BK)\mathbf{x}_{t,\nl}}$$
$$\leq \sqrt{\mathbf{x}_{t,\nl}^{\top} P_{K,\gamma}\mathbf{x}_{t,\nl}}$$
$$\leq V_0^{1/2}\left(1 - \frac{1}{2\|P_{K,\gamma}\|_{\op}}\right)^{t/2}.$$

Lastly, we bound the first term by again using Lemma B.2,

$$\|\sqrt{\gamma} \cdot P_{K,\gamma}^{1/2} G_{\nl}(\mathbf{x}_{t,\nl}, K\mathbf{x}_{t,\nl})\| = \|P_{K,\gamma}^{1/2}\mathbf{x}_{t+1,\nl}\| \leq V_0^{1/2}\left(1 - \frac{1}{2\|P_{K,\gamma}\|_{\op}}\right)^{\frac{t+1}{2}}.$$

Therefore, $T_2$ is bounded by $2V_0^{1/2}(1 - 1/(2\|P_{K,\gamma}\|_{\op}))^{t/2}$. Going back to Eq. (B.9), we can combine our bounds on $T_1$ and $T_2$ to conclude that,

$$J_{\nl}(K \mid \gamma, \mathbf{x}) - J_{\lin}(K \mid \gamma, \mathbf{x}) \leq V_0^{3/2} \cdot 2\beta_{\nl}\|P_{K,\gamma}\|_{\op}^{1/2}(\|K\|_{\op}^2 + 1) \cdot \sum_{t=0}^{\infty}\left(1 - \frac{1}{2\|P_{K,\gamma}\|_{\op}}\right)^t$$
$$= V_0^{3/2} \cdot 4\beta_{\nl}\|P_{K,\gamma}\|_{\op}^{3/2}(\|K\|_{\op}^2 + 1).$$

Using the fact that $1 + \|K\|_{\op}^2 \leq 2\|P_{K,\gamma}\|_{\op}$ and $V_0 \leq \|\mathbf{x}_0\|^2\|P_{K,\gamma}\|_{\op}$, we get that

$$V_0^{3/2} \cdot 4\beta_{\nl}\|P_{K,\gamma}\|_{\op}^{3/2}(\|K\|_{\op}^2 + 1) \leq 8\beta_{\nl}\|\mathbf{x}_0\|^3\|P_{K,\gamma}\|_{\op}^4.$$

$\square$

### B.2.2  Establishing similarity of gradients

Much like the previous lemma which bounds the costs between the linear and nonlinear system via the performance difference lemma, this next lemma differentiates the performance difference lemma to bound the difference between gradients.

**Lemma B.6.** *Assume $J_{\lin}(K \mid \gamma) < \infty$. If $\mathbf{x}_0$ is such that*

$$\mathbf{x}_0 P_{K,\gamma}\mathbf{x}_0 \leq \frac{r_{\nl}^2}{144\beta_{\nl}^2\|P_{K,\gamma}\|_{\op}^4}$$

*then,*

$$\|\nabla_K J_{\nl}(K \mid \gamma, \mathbf{x}_0) - \nabla_K J_{\lin}(K \mid \gamma, \mathbf{x}_0)\|_{\F} \leq 48\beta_{\nl}(1 + \|B\|_{\op})\|P_{K,\gamma}\|_{\op}^7\|\mathbf{x}_0\|^3.$$

*Proof.* Using the variational definition of the Frobenius norm,

$$\|\nabla_K J_{\nl}(K \mid \gamma, \mathbf{x}) - \nabla_K J_{\lin}(K \mid \gamma, \mathbf{x})\|_{\F} = \sup_{\|\Delta\|_{\F} \leq 1} \tr\left[(\nabla_K J_{\nl}(K \mid \gamma, \mathbf{x}) - \nabla_K J_{\lin}(K \mid \gamma, \mathbf{x}))^{\top}\Delta\right]$$
$$= \sup_{\|\Delta\|_{\F} \leq 1} D_K J_{\nl}(K \mid \gamma, \mathbf{x}) - D_K J_{\lin}(K \mid \gamma, \mathbf{x}),$$

where $D_K$ is the directional derivative operator in the direction $\Delta$. The argument follows by bounding the directional derivative appearing above. From the performance difference lemma, Lemma B.4, we have that

$$D_K J_{\nl}(K \mid \gamma, \mathbf{x}) - D_K J_{\lin}(K \mid \gamma, \mathbf{x}) = \sum_{t=0}^{\infty} \gamma \cdot D_K\left[\cdot f_{\nl}(\mathbf{x}_{t,\nl}, K\mathbf{x}_{t,\nl})^{\top} P_{K,\gamma}\mathbf{x}_{t+1,\nl}\right].$$

Each term appearing in the sum above can be decomposed into the following three terms,

$$\underbrace{\gamma \cdot (D_K \cdot f_{\nl}(\mathbf{x}_{t,\nl}, K\mathbf{x}_{t,\nl}))^{\top} P_{K,\gamma}\mathbf{x}_{t+1,\nl}}_{:=T_1}$$
$$+ \underbrace{\gamma \cdot f_{\nl}(\mathbf{x}_{t,\nl}, K\mathbf{x}_{t,\nl})(D_K P_{K,\gamma})\mathbf{x}_{t+1,\nl}}_{:=T_2}$$
$$+ \underbrace{\gamma \cdot f_{\nl}(\mathbf{x}_{t,\nl}, K\mathbf{x}_{t,\nl})P_{K,\gamma}(D_K\mathbf{x}_{t+1,\nl})}_{:=T_3}.$$

In order to bound each of these three terms, we start by bounding the directional derivatives appearing above. Throughout the remainder of the proof, we will make repeated use of the following inequalities which follow from Lemma 3.1, Lemma B.2, and our assumption on $\mathbf{x}_0$. For all $t \geq 0$,

$$\|\mathbf{x}_{t,\mathrm{nl}}\|^2 \leq V_0 \left(1 - \frac{1}{2\|P_{K,\gamma}\|_{\mathrm{op}}}\right)^t \tag{B.10}$$

$$\|\mathbf{x}_{t,\mathrm{nl}}\| + \|K\mathbf{x}_{t,\mathrm{nl}}\| \leq r_{\mathrm{nl}}. \tag{B.11}$$

**Lemma B.7** (Bounding $D_K\mathbf{x}_{t,\mathrm{nl}}$.)**.** *Let* $\{\mathbf{x}_{t,\mathrm{nl}}\}$ *be the sequence of states generated according to* $\mathbf{x}_{t+1,\mathrm{nl}} = \sqrt{\gamma}G_{\mathrm{nl}}(\mathbf{x}_{t,\mathrm{nl}}, K\mathbf{x}_{t,\mathrm{nl}})$. *Under the same assumptions as in Lemma B.6, for all* $t \geq 0$

$$\|D_K\mathbf{x}_{t,\mathrm{nl}}\| \leq t \cdot \|P_{K,\gamma}\|_{\mathrm{op}}^{1/2} \left(\frac{1}{6\|P_{K,\gamma}\|_{\mathrm{op}}^2} + \|B\|_{\mathrm{op}}\right) \left(1 - \frac{1}{2\|P_{K,\gamma}\|_{\mathrm{op}}}\right)^{\frac{t-1}{2}} V_0^{1/2}.$$

*Proof.* Taking derivatives, we get that

$$D_K f_{\mathrm{nl}}(\mathbf{x}_{t,\mathrm{nl}}, K\mathbf{x}_{t,\mathrm{nl}}) = \nabla_{\mathbf{x},\mathbf{u}} f_{\mathrm{nl}}(\mathbf{x}, \mathbf{u})\Big|_{(\mathbf{x},\mathbf{u})=(\mathbf{x}_{t,\mathrm{nl}}, K\mathbf{x}_{t,\mathrm{nl}})} \begin{bmatrix} D_K\mathbf{x}_{t,\mathrm{nl}} \\ D_K(K\mathbf{x}_{t,\mathrm{nl}}) \end{bmatrix}.$$

Since, $D_K(K\mathbf{x}_{t,\mathrm{nl}}) = \Delta\mathbf{x}_{t,\mathrm{nl}} + K(D_K\mathbf{x}_{t,\mathrm{nl}})$, we can rewrite the expression above as,

$$\begin{aligned}
D_K f_{\mathrm{nl}}(\mathbf{x}_{t,\mathrm{nl}}, K\mathbf{x}_{t,\mathrm{nl}}) &= \nabla_{\mathbf{x},\mathbf{u}} f_{\mathrm{nl}}(\mathbf{x}, \mathbf{u})\Big|_{(\mathbf{x},\mathbf{u})=(\mathbf{x}_{t,\mathrm{nl}}, K\mathbf{x}_{t,\mathrm{nl}})} \begin{bmatrix} 0 \\ \Delta \end{bmatrix} \mathbf{x}_{t,\mathrm{nl}} \\
&\quad + \nabla_{\mathbf{x},\mathbf{u}} f_{\mathrm{nl}}(\mathbf{x}, \mathbf{u})\Big|_{(\mathbf{x},\mathbf{u})=(\mathbf{x}_{t,\mathrm{nl}}, K\mathbf{x}_{t,\mathrm{nl}})} \begin{bmatrix} I \\ K \end{bmatrix} D_K\mathbf{x}_{t,\mathrm{nl}}.
\end{aligned} \tag{B.12}$$

Next, we compute $D_K\mathbf{x}_{t,\mathrm{nl}}$,

$$\begin{aligned}
D_K\mathbf{x}_{t,\mathrm{nl}} &= \sqrt{\gamma} \cdot (D_K f_{\mathrm{nl}}(\mathbf{x}_{t-1,\mathrm{nl}}, K\mathbf{x}_{t-1,\mathrm{nl}}) + D_K[(A + BK)\mathbf{x}_{t-1,\mathrm{nl}}]) \\
&= \sqrt{\gamma} \cdot (D_K f_{\mathrm{nl}}(\mathbf{x}_{t-1,\mathrm{nl}}, K\mathbf{x}_{t-1,\mathrm{nl}}) + B\Delta\mathbf{x}_{t-1,\mathrm{nl}} + (A + BK)D_K\mathbf{x}_{t-1,\mathrm{nl}}).
\end{aligned}$$

Plugging in our earlier calculation for $D_K f_{\mathrm{nl}}(\mathbf{x}_{t,\mathrm{nl}}, K\mathbf{x}_{t,\mathrm{nl}})$, we get that the following recursion holds for $\psi_t := D_K\mathbf{x}_{t,\mathrm{nl}}$,

$$\psi_t := D_K\mathbf{x}_{t,\mathrm{nl}} = \underbrace{\sqrt{\gamma} \cdot \left[\nabla_{\mathbf{x},\mathbf{u}} f_{\mathrm{nl}}(\mathbf{x}, \mathbf{u})\Big|_{(\mathbf{x},\mathbf{u})=(\mathbf{x}_{t-1,\mathrm{nl}}, K\mathbf{x}_{t-1,\mathrm{nl}})} \begin{bmatrix} 0 \\ \Delta \end{bmatrix} + B\Delta\right] \mathbf{x}_{t-1,\mathrm{nl}}}_{:=m_{t-1}} \tag{B.13}$$

$$+ \underbrace{\sqrt{\gamma} \cdot \left[\nabla_{\mathbf{x},\mathbf{u}} f_{\mathrm{nl}}(\mathbf{x}, \mathbf{u})\Big|_{(\mathbf{x},\mathbf{u})=(\mathbf{x}_{t-1,\mathrm{nl}}, K\mathbf{x}_{t-1,\mathrm{nl}})} \begin{bmatrix} I \\ K \end{bmatrix} + (A + BK)\right] \psi_{t-1}}_{:=N_{t-1}}. \tag{B.14}$$

Using the shorthand introduced above, we can re-express the recursion as,

$$\psi_t = m_{t-1} + N_{t-1}\psi_{t-1}.$$

Unrolling this recursion, with the base case that $\psi_0 = D_K\mathbf{x}_{0,\mathrm{nl}} = 0$, we get that

$$\psi_t = \sum_{j=0}^{t-1} \left(\prod_{i=j+1}^{t-1} N_i\right) m_j.$$

Therefore,

$$\|\psi_t\| \leq \sum_{j=0}^{t-1} \left\|\prod_{i=j+1}^{t-1} N_i\right\|_{\mathrm{op}} \|m_j\|. \tag{B.15}$$

Next, we prove that each matrix $N_i$ is stable so that the product of the $N_i$ is small. By Lemma 3.1 and our earlier inequalities, Eq. (B.10) & Eq. (B.11), we have that,

$$
\left\| \nabla_{\mathbf{x},\mathbf{u}} f_{\mathrm{nl}}(\mathbf{x},\mathbf{u}) \Big|_{(\mathbf{x},\mathbf{u})=(\mathbf{x}_{t-1,\mathrm{nl}}, K\mathbf{x}_{t-1,\mathrm{nl}})} \begin{bmatrix} I \\ K \end{bmatrix} \right\|_{\mathrm{op}} \leq \left\| \nabla_{\mathbf{x},\mathbf{u}} f_{\mathrm{nl}}(\mathbf{x},\mathbf{u}) \Big|_{(\mathbf{x},\mathbf{u})=(\mathbf{x}_{t-1,\mathrm{nl}}, K\mathbf{x}_{t-1,\mathrm{nl}})} \right\|_{\mathrm{op}} \left\| \begin{bmatrix} I \\ K \end{bmatrix} \right\|_{\mathrm{op}}
$$
$$
\leq \beta_{\mathrm{nl}}(1 + \|K\|_{\mathrm{op}})^2 \|\mathbf{x}_{t-1,\mathrm{nl}}\|
$$
$$
\leq \frac{1}{6\|P_{K,\gamma}\|_{\mathrm{op}}^2}, \tag{B.16}
$$

where we have used our initial condition on $V_0$. Therefore, $N_i = \sqrt{\gamma}(A + BK) + \Delta_{N_i}$ where $\|\Delta_{N_i}\|_{\mathrm{op}} \leq 1/(6\|P_{K,\gamma}\|_{\mathrm{op}}^2)$ for all $i$. By definition of the operator norm, Lemma A.6, and the fact that $P_{K,\gamma} \succeq I$:

$$
\left\| \prod_{i=j+1}^{t-1} N_i \right\|_{\mathrm{op}}^2 = \left\| \left( \prod_{i=j+1}^{t-1} N_i \right)^\top \left( \prod_{i=j+1}^{t-1} N_i \right) \right\|_{\mathrm{op}}
$$
$$
\leq \left\| \left( \prod_{i=j+1}^{t-1} N_i \right)^\top P_{K,\gamma} \left( \prod_{i=j+1}^{t-1} N_i \right) \right\|_{\mathrm{op}}
$$
$$
\leq \|P_{K,\gamma}\|_{\mathrm{op}} \left( 1 - \frac{1}{2\|P_{K,\gamma}\|_{\mathrm{op}}} \right)^{t-j-1}.
$$

Then, using our bound on $\nabla_{\mathbf{x},\mathbf{u}} f_{\mathrm{nl}}$ from Eq. (B.16), and Lemma B.2, we bound $m_j$ (defined in Eq. (B.13)) as,

$$
\|m_j\| \leq \left\| \sqrt{\gamma} \left[ \nabla_{\mathbf{x},\mathbf{u}} f_{\mathrm{nl}}(\mathbf{x},\mathbf{u}) \Big|_{(\mathbf{x},\mathbf{u})=(\mathbf{x}_{j,\mathrm{nl}}, K\mathbf{x}_{j,\mathrm{nl}})} \begin{bmatrix} 0 \\ \Delta \end{bmatrix} + B\Delta \right] \right\|_{\mathrm{op}} \|\mathbf{x}_{j,\mathrm{nl}}\|
$$
$$
\leq \left( \frac{1}{6\|P_{K,\gamma}\|_{\mathrm{op}}^2} + \|B\|_{\mathrm{op}} \right) V_0^{1/2} \left( 1 - \frac{1}{2\|P_{K,\gamma}\|_{\mathrm{op}}} \right)^{j/2}.
$$

Returning to our earlier bound on $\|\psi_t\|$ in Eq. (B.15), we conclude that,

$$
\|\psi_t\| \leq \sum_{j=0}^{t-1} \|P_{K,\gamma}\|_{\mathrm{op}}^{1/2} \left( 1 - \frac{1}{2\|P_{K,\gamma}\|_{\mathrm{op}}} \right)^{(t-j-1)/2} \left( \frac{1}{6\|P_{K,\gamma}\|_{\mathrm{op}}^2} + \|B\|_{\mathrm{op}} \right) V_0^{1/2} \left( 1 - \frac{1}{2\|P_{K,\gamma}\|_{\mathrm{op}}} \right)^{j/2}
$$
$$
\leq t \cdot \|P_{K,\gamma}\|_{\mathrm{op}}^{1/2} \left( \frac{1}{6\|P_{K,\gamma}\|_{\mathrm{op}}^2} + \|B\|_{\mathrm{op}} \right) \left( 1 - \frac{1}{2\|P_{K,\gamma}\|_{\mathrm{op}}} \right)^{\frac{t-1}{2}} V_0^{1/2},
$$

concluding the proof. $\qquad\square$

Using this, we can now return to bounding $\mathrm{D}_K f_{\mathrm{nl}}(\mathbf{x}_{t,\mathrm{nl}}, K\mathbf{x}_{t,\mathrm{nl}})$.

**Lemma B.8** (Bounding $\mathrm{D}_K f_{\mathrm{nl}}(\mathbf{x}_{t,\mathrm{nl}}, K\mathbf{x}_{t,\mathrm{nl}})$)**.** *For all $t \geq 0$, the following bound holds:*

$$
\|\mathrm{D}_K f_{\mathrm{nl}}(\mathbf{x}_{t,\mathrm{nl}}, K\mathbf{x}_{t,\mathrm{nl}})\| \leq 8\beta_{\mathrm{nl}}\|P_{K,\gamma}\|_{\mathrm{op}}^2 (1 + \|B\|_{\mathrm{op}}) \cdot t \left( 1 - \frac{1}{2\|P_{K,\gamma}\|_{\mathrm{op}}} \right)^{t-1/2} \|\mathbf{x}_0\|^2.
$$

*Proof.* From Eq. (B.12), $\|\mathrm{D}_K f_{\mathrm{nl}}(\mathbf{x}_{t,\mathrm{nl}}, K\mathbf{x}_{t,\mathrm{nl}})\|$ is less than,

$$
\left\| \nabla_{\mathbf{x},\mathbf{u}} f_{\mathrm{nl}}(\mathbf{x},\mathbf{u}) \Big|_{(\mathbf{x},\mathbf{u})=(\mathbf{x}_{t,\mathrm{nl}}, K\mathbf{x}_{t,\mathrm{nl}})} \right\|_{\mathrm{op}} \left\| \begin{bmatrix} 0 \\ \Delta \end{bmatrix} \right\|_{\mathrm{op}} \|\mathbf{x}_{t,\mathrm{nl}}\|
$$
$$
+ \left\| \nabla_{\mathbf{x},\mathbf{u}} f_{\mathrm{nl}}(\mathbf{x},\mathbf{u}) \Big|_{(\mathbf{x},\mathbf{u})=(\mathbf{x}_{t,\mathrm{nl}}, K\mathbf{x}_{t,\mathrm{nl}})} \right\|_{\mathrm{op}} \left\| \begin{bmatrix} I \\ K \end{bmatrix} \right\|_{\mathrm{op}} \|\mathrm{D}_K \mathbf{x}_{t,\mathrm{nl}}\|.
$$

Again using the assumption on the nonlinear dynamics, we can bound the gradient terms as in Eq. (B.16) and get that $\|D_K f_{\mathrm{nl}}(\mathbf{x}_{t,\mathrm{nl}}, K\mathbf{x}_{t,\mathrm{nl}})\|$ is no larger than,

$$\beta_{\mathrm{nl}}(1 + \|K\|_{\mathrm{op}})\|\mathbf{x}_{t,\mathrm{nl}}\|^2 + \beta_{\mathrm{nl}}(1 + \|K\|_{\mathrm{op}})^2\|\mathbf{x}_{t,\mathrm{nl}}\|\|D_K\mathbf{x}_{t,\mathrm{nl}}\|$$
$$\leq\ 2\beta_{\mathrm{nl}}(1 + \|K\|_{\mathrm{op}})^2\|\mathbf{x}_{t,\mathrm{nl}}\|\max\{\|\mathbf{x}_{t,\mathrm{nl}}\|,\ \|D_K\mathbf{x}_{t,\mathrm{nl}}\|\}.$$

Seeing as how our upper bound on $\|D_K\mathbf{x}_{t,\mathrm{nl}}\|$ in Lemma B.7 is always larger than the bound for $\|\mathbf{x}_{t,\mathrm{nl}}\|$ in Eq. (B.10), we can bound $\max\{\|\mathbf{x}_{t,\mathrm{nl}}\|,\ \|D_K\mathbf{x}_{t,\mathrm{nl}}\|\}$ by the former. Consequently,

$$\|D_K f_{\mathrm{nl}}(\mathbf{x}_{t,\mathrm{nl}}, K\mathbf{x}_{t,\mathrm{nl}})\|\ \leq\ 2\beta_{\mathrm{nl}}(1 + \|K\|_{\mathrm{op}})^2\|P_{K,\gamma}\|_{\mathrm{op}}^{1/2}V_0\left(\frac{1}{6\|P_{K,\gamma}\|_{\mathrm{op}}^2} + \|B\|_{\mathrm{op}}\right)\cdot t\left(1 - \frac{1}{2\|P_{K,\gamma}\|_{\mathrm{op}}}\right)^{t-1/2}$$

$$\leq\ 8\beta_{\mathrm{nl}}\|P_{K,\gamma}\|_{\mathrm{op}}^2(1 + \|B\|_{\mathrm{op}})\cdot t\left(1 - \frac{1}{2\|P_{K,\gamma}\|_{\mathrm{op}}}\right)^{t-1/2}\|\mathbf{x}_0\|^2.$$

$\square$

Finally, we bound $D_K P_{K,\gamma}$.

**Lemma B.9** (Bounding $D_K P_{K,\gamma}$)**.** *The following bound holds:*

$$\|D_K P_{K,\gamma}\|_{\mathrm{op}}\ \leq\ 2\|P_{K,\gamma}\|_{\mathrm{op}}^2(\|B\|_{\mathrm{op}} + \|K\|_{\mathrm{op}}).$$

*Proof.* By definition of the discrete time Lyapunov equation,

$$P_{K,\gamma} = Q + K^\top R K + \gamma(A + BK)^\top P_{K,\gamma}(A + BK).$$

Therefore, the directional derivative $D_K P_{K,\gamma}$ satisfies another Lyapunov equation,

$$D_K P_{K,\gamma} = \underbrace{\Delta^\top R K + K^\top R\Delta + \gamma\cdot(B\Delta)^\top P_{K,\gamma}(A + BK) + \gamma\cdot(A + BK)^\top P_{K,\gamma}B\Delta}_{:=E_K}$$
$$+ \gamma(A + BK)^\top(D_K P_{K,\gamma})(A + BK),$$

implying that $D_K P_{K,\gamma} = \mathsf{dlyap}(\sqrt{\gamma}(A + BK), E_K)$. By properties of the Lyapunov equation,

$$D_K P_{K,\gamma} \preceq \|E_K\|_{\mathrm{op}}\mathsf{dlyap}(A + BK, I)$$
$$\preceq \|E_K\|_{\mathrm{op}}\mathsf{dlyap}(\sqrt{\gamma}(A + BK), K^\top R K + Q) = \|E_K\|_{\mathrm{op}}P_{K,\gamma}.$$

Therefore, to bound $\|D_K P_{K,\gamma}\|_{\mathrm{op}}$ it suffices to bound, $\|E_k\|_{\mathrm{op}}$. Using the fact that $\|P_{K,\gamma}^{1/2}\sqrt{\gamma}(A + BK)\|_{\mathrm{op}} \leq \|P_{K,\gamma}^{1/2}\|_{\mathrm{op}}$ and that $\|\Delta\|_{\mathrm{op}} \leq 1$, a short calculation reveals that,

$$\|E_K\|_{\mathrm{op}}\ \leq\ 2\|P_{K,\gamma}\|_{\mathrm{op}}(\|B\|_{\mathrm{op}} + \|K\|_{\mathrm{op}}),$$

which together with our previous bound on $D_K P_{K,\gamma}$ implies that,

$$\|D_K P_{K,\gamma}\|_{\mathrm{op}}\ \leq\ 2\|P_{K,\gamma}\|_{\mathrm{op}}^2(\|B\|_{\mathrm{op}} + \|K\|_{\mathrm{op}}).$$

$\square$

With Lemmas B.7 to B.9 in place, we now return to bounding terms $T_1, T_2, T_3$.

**Bounding $T_1$**    Recall $T_1 := \gamma\cdot(D_K\cdot f_{\mathrm{nl}}(\mathbf{x}_{t,\mathrm{nl}}, K\mathbf{x}_{t,\mathrm{nl}}))^\top P_{K,\gamma}\mathbf{x}_{t+1,\mathrm{nl}}$. We then have

$$\|T_1\|\ \leq\ \|(D_K f_{\mathrm{nl}}(\mathbf{x}_{t,\mathrm{nl}}, K\mathbf{x}_{t,\mathrm{nl}}))^\top P_{K,\gamma}\mathbf{x}_{t+1,\mathrm{nl}}\|\ \leq\ \|P_{K,\gamma}\|_{\mathrm{op}}\|D_K f_{\mathrm{nl}}(\mathbf{x}_{t,\mathrm{nl}}, K\mathbf{x}_{t,\mathrm{nl}})\|\|\mathbf{x}_{t+1,\mathrm{nl}}\|.$$

Using the bound on $\|\mathbf{x}_{t+1,\mathrm{nl}}\|$ stated in Eq. (B.10) and on $\|D_K f_{\mathrm{nl}}(\mathbf{x}_{t,\mathrm{nl}}, K\mathbf{x}_{t,\mathrm{nl}})\|$ from Lemma B.8, the above simplifies to

$$8\beta_{\mathrm{nl}}\|P_{K,\gamma}\|_{\mathrm{op}}^{7/2}(1 + \|B\|_{\mathrm{op}})\cdot t\left(1 - \frac{1}{2\|P_{K,\gamma}\|_{\mathrm{op}}}\right)^{1.5t}\|\mathbf{x}_0\|^3.$$

**Bounding $T_2$**    Recall $T_2 = f_{\mathrm{nl}}(\mathbf{x}_{t,\mathrm{nl}}, K\mathbf{x}_{t,\mathrm{nl}})\mathrm{D}_K P_{K,\gamma}\mathbf{x}_{t+1,\mathrm{nl}}$, so that

$$\|T_2\| \;\le\; \|f_{\mathrm{nl}}(\mathbf{x}_{t,\mathrm{nl}}, K\mathbf{x}_{t,\mathrm{nl}})\|\|\mathrm{D}_K P_{K,\gamma}\|_{\mathrm{op}}\|\mathbf{x}_{t+1,\mathrm{nl}}\|.$$

Using the bound on $\|f_{\mathrm{nl}}(\mathbf{x},\mathbf{u})\|$ from [Lemma 3.1](),

$$
\begin{aligned}
\|f_{\mathrm{nl}}(\mathbf{x}_{t,\mathrm{nl}}, K\mathbf{x}_{t,\mathrm{nl}})\| &\;\le\; \beta_{\mathrm{nl}}(1 + \|K\|_{\mathrm{op}}^2)\|\mathbf{x}_{t,\mathrm{nl}}\|^2 \\
&\;\le\; \beta_{\mathrm{nl}}(1 + \|K\|_{\mathrm{op}}^2)\|P_{K,\gamma}\|_{\mathrm{op}}\left(1 - \frac{1}{2\|P_{K,\gamma}\|_{\mathrm{op}}}\right)^t \|\mathbf{x}_0\|^2 \\
&\;\le\; 2\beta_{\mathrm{nl}}\|P_{K,\gamma}\|_{\mathrm{op}}^2\left(1 - \frac{1}{2\|P_{K,\gamma}\|_{\mathrm{op}}}\right)^t \|\mathbf{x}_0\|^2.
\end{aligned}
\tag{B.17}
$$

Therefore, using [Eq. (B.10)]() again and [Lemma B.9](),

$$
\begin{aligned}
\|T_2\| &\;\le\; 4\beta_{\mathrm{nl}}\|P_{K,\gamma}\|_{\mathrm{op}}^{9/2}(\|B\|_{\mathrm{op}} + \|K\|_{\mathrm{op}})\left(1 - \frac{1}{2\|P_{K,\gamma}\|_{\mathrm{op}}}\right)^{\frac{3}{2}t+\frac{1}{2}} \|\mathbf{x}_0\|^3 \\
&\;\le\; 4\beta_{\mathrm{nl}}\|P_{K,\gamma}\|_{\mathrm{op}}^5(\|B\|_{\mathrm{op}} + 1)\left(1 - \frac{1}{2\|P_{K,\gamma}\|_{\mathrm{op}}}\right)^{\frac{3}{2}t+\frac{1}{2}} \|\mathbf{x}_0\|^3.
\end{aligned}
$$

**Bounding $T_3$.**    Recall $T_3 := f_{\mathrm{nl}}(\mathbf{x}_{t,\mathrm{nl}}, K\mathbf{x}_{t,\mathrm{nl}})P_{,K,\gamma}\mathrm{D}_K\mathbf{x}_{t+1,\mathrm{nl}}$. From [Eq. (B.17)]() and [Lemma B.7](), we have that

$$
\begin{aligned}
\|T_3\| &\;\le\; \|f_{\mathrm{nl}}(\mathbf{x}_{t,\mathrm{nl}}, K\mathbf{x}_{t,\mathrm{nl}})\|\|P_{,K,\gamma}\|_{\mathrm{op}}\|\mathrm{D}_K\mathbf{x}_{t+1,\mathrm{nl}}\|_{\mathrm{op}} \\
&\;\le\; 2\beta_{\mathrm{nl}}\|P_{K,\gamma}\|_{\mathrm{op}}^4(1 + \|B\|_{\mathrm{op}})\cdot(t+1)\left(1 - \frac{1}{2\|P_{K,\gamma}\|_{\mathrm{op}}}\right)^{1.5t} \|\mathbf{x}_0\|^3.
\end{aligned}
$$

**Wrapping up**    Therefore,

$$\|T_1\| + \|T_2\| + \|T_3\| \;\le\; 12\beta_{\mathrm{nl}}(1 + \|B\|_{\mathrm{op}})\|P_{K,\gamma}\|_{\mathrm{op}}^5(t+1)\left(1 - \frac{1}{2\|P_{K,\gamma}\|_{\mathrm{op}}}\right)^{\frac{3}{2}t} \|\mathbf{x}_0\|^3.$$

And hence,

$$
\begin{aligned}
\sum_{t=0}^{\infty}\mathrm{D}_K\left[f_{\mathrm{nl}}(\mathbf{x}_{t,\mathrm{nl}}, K\mathbf{x}_{t,\mathrm{nl}})^\top P_{K,\gamma}\mathbf{x}_{t+1,\mathrm{nl}}\right] &\;\le\; 12\beta_{\mathrm{nl}}(1 + \|B\|_{\mathrm{op}})\|P_{K,\gamma}\|_{\mathrm{op}}^5\|\mathbf{x}_0\|^3\sum_{t=0}^{\infty}(t+1)\left(1 - \frac{1}{2\|P_{K,\gamma}\|_{\mathrm{op}}}\right)^t \\
&= 48\beta_{\mathrm{nl}}(1 + \|B\|_{\mathrm{op}})\|P_{K,\gamma}\|_{\mathrm{op}}^7\|\mathbf{x}_0\|^3.
\end{aligned}
$$

$\square$

## B.3    Establishing Lyapunov functions: Proof of [Lemma B.1]()

*Proof.* To be concise, we use the shorthand, $\mathbf{z} = (\mathbf{x}, K\mathbf{x})$. We start by expanding out,

$$\gamma\cdot G_{\mathrm{nl}}(\mathbf{z})^\top P_K G_{\mathrm{nl}}(\mathbf{z}) = \gamma\left(G_{\mathrm{lin}}(\mathbf{z})^\top P_K G_{\mathrm{lin}}(\mathbf{z}) + f_{\mathrm{nl}}(\mathbf{z})^\top P_K f_{\mathrm{nl}}(\mathbf{z}) + 2f_{\mathrm{nl}}(\mathbf{z})^\top P_K G_{\mathrm{lin}}(\mathbf{z})\right).$$

By the AM-GM inequality for vectors, the following holds for any $\tau > 0$,

$$2f_{\mathrm{nl}}(\mathbf{z})^\top P_K G_{\mathrm{lin}}(\mathbf{z}) \;\le\; \tau\cdot G_{\mathrm{lin}}(\mathbf{z})^\top P_K G_{\mathrm{lin}}(\mathbf{z}) + \frac{1}{\tau}\cdot f_{\mathrm{nl}}(\mathbf{z})^\top P_K f_{\mathrm{nl}}(\mathbf{z}).$$

Combining these two relationships, we get that,

$$\gamma\cdot G_{\mathrm{nl}}(\mathbf{z})^\top P_K G_{\mathrm{nl}}(\mathbf{z}) \;\le\; \gamma\cdot(1 + \tau)\cdot G_{\mathrm{lin}}(\mathbf{z})^\top P_K G_{\mathrm{lin}}(\mathbf{z}) + \gamma\cdot\left(1 + \frac{1}{\tau}\right)\cdot f_{\mathrm{nl}}(\mathbf{z})^\top P_K f_{\mathrm{nl}}(\mathbf{z}).$$

$$\tag{B.18}$$

Next, by properties of the Lyapunov function [Lemma A.5](), we have that

$$\gamma G_{\mathrm{lin}}(\mathbf{z})^\top P_K G_{\mathrm{lin}}(\mathbf{z}) = \gamma\cdot\mathbf{x}^\top(A + BK)^\top P_{K,\gamma}(A + BK)\mathbf{x} \;\le\; \mathbf{x}^\top P_{K,\gamma}\mathbf{x}\left(1 - \frac{1}{\|P_{K,\gamma}\|_{\mathrm{op}}}\right).$$

Letting, $V_{\mathbf{x}} := \mathbf{x}^\top P_{K,\gamma}\mathbf{x}$, we can plug in the previous expression into Eq. (B.18) and optimize over $\tau$ to get that,

$$\gamma \cdot G_{\mathrm{nl}}(\mathbf{z})^\top P_{K,\gamma} G_{\mathrm{nl}}(\mathbf{z}) \ \leq \ V_{\mathbf{x}}\left(1 - \frac{1}{\|P_{K,\gamma}\|_{\mathrm{op}}}\right) + f_{\mathrm{nl}}(\mathbf{z})^\top P_{K,\gamma} f_{\mathrm{nl}}(\mathbf{z}) + 2\sqrt{V_{\mathbf{x}} f_{\mathrm{nl}}(\mathbf{z})^\top P_{K,\gamma} f_{\mathrm{nl}}(\mathbf{z})}$$

$$\leq \ V_{\mathbf{x}}\left(1 - \frac{1}{\|P_{K,\gamma}\|_{\mathrm{op}}}\right) + \|P_{K,\gamma}\|_{\mathrm{op}}\|f_{\mathrm{nl}}(\mathbf{z})\|^2 + 2\sqrt{V_{\mathbf{x}}\|P_{K,\gamma}\|_{\mathrm{op}}}\|f_{\mathrm{nl}}(\mathbf{z})\|,$$

where we have dropped a factor of $\gamma$ from the last two terms since $\gamma \leq 1$. Next, the proof follows by noting that this following inequality is satisfied,

$$\|P_{K,\gamma}\|_{\mathrm{op}}\|f_{\mathrm{nl}}(\mathbf{z})\|^2 + 2\sqrt{V_{\mathbf{x}}\|P_{K,\gamma}\|_{\mathrm{op}}}\|f_{\mathrm{nl}}(\mathbf{z})\| \ \leq \ \frac{V_{\mathbf{x}}}{2\|P_{K,\gamma}\|_{\mathrm{op}}}, \tag{B.19}$$

whenever,

$$\|f_{\mathrm{nl}}(\mathbf{z})\| \ \leq \ \sqrt{\frac{V_{\mathbf{x}}}{2\|P_{K,\gamma}\|_{\mathrm{op}}^2} + \frac{V_{\mathbf{x}}}{\|P_{K,\gamma}\|_{\mathrm{op}}}} - \sqrt{\frac{V_{\mathbf{x}}}{\|P_{K,\gamma}\|_{\mathrm{op}}}}$$

$$= \sqrt{\frac{V_{\mathbf{x}}}{\|P_{K,\gamma}\|_{\mathrm{op}}}}\left(\sqrt{\frac{1}{2\|P_{K,\gamma}\|_{\mathrm{op}}} + 1} - 1\right). \tag{B.20}$$

Therefore, assuming the inequality above Eq. (B.20), we get our desired result showing that

$$G_{\mathrm{nl}}(\mathbf{z})^\top P_K G_{\mathrm{nl}}(\mathbf{z}) \ \leq \ \mathbf{x}^\top P_K \mathbf{x} \cdot \left(1 - \frac{1}{2\|P_K\|_{\mathrm{op}}}\right).$$

We conclude the proof by showing that Eq. (B.20) is satisfied for all $\mathbf{x}$ small enough. In particular, using our bounds on $f_{\mathrm{nl}}$ from Lemma 3.1, if $\|\mathbf{x}\| + \|K\mathbf{x}\| \leq r_{\mathrm{nl}}$,

$$\|f_{\mathrm{nl}}(\mathbf{z})\| \ \leq \ \beta_{\mathrm{nl}}(\|\mathbf{x}\|^2 + \|K\mathbf{x}\|^2) \ \leq \ \beta_{\mathrm{nl}}(1 + \|K\|_{\mathrm{op}}^2)\|\mathbf{x}\|^2.$$

Since $V_{\mathbf{x}} \geq \|\mathbf{x}\|^2$, in order for Eq. (B.20) to hold, it suffices for $\|\mathbf{x}\|$ to satisfy

$$\|\mathbf{x}\| \ \leq \ \min\left\{\frac{r_{\mathrm{nl}}}{1 + \|K\|_{\mathrm{op}}}, \ \frac{1}{\beta_{\mathrm{nl}}\|P_{K,\gamma}\|_{\mathrm{op}}^{1/2}(1 + \|K\|_{\mathrm{op}}^2)}\right\}.$$

Using the fact that $\|K\|_{\mathrm{op}}^2 \leq \|P_{K,\gamma}\|_{\mathrm{op}}$, we can simplify upper bound on $\|\mathbf{x}\|$ to be,

$$\|\mathbf{x}\| \ \leq \ \frac{r_{\mathrm{nl}}}{2\beta_{\mathrm{nl}}\|P_{K,\gamma}\|_{\mathrm{op}}^{3/2}}.$$

Note that this condition is always implied by the condition on $\mathbf{x}^\top P_{K,\gamma}\mathbf{x}$ in the statement of the proposition. $\qquad\square$

## B.4  Bounding the nonlinearity: Proof of Lemma 3.1

Since the origin is an equilibrium point, $G_{\mathrm{nl}}(0,0) = 0$, we can rewrite $G_{\mathrm{nl}}$ as,

$$G_{\mathrm{nl}}(\mathbf{x}, \mathbf{u}) = G_{\mathrm{nl}}(0,0) + \nabla G_{\mathrm{nl}}(\mathbf{x}, \mathbf{u})\big|_{(\mathbf{x},\mathbf{u})=(0,0)} \begin{bmatrix} \mathbf{x} \\ \mathbf{u} \end{bmatrix} + f_{\mathrm{nl}}(\mathbf{x}, \mathbf{u}),$$

$$= A_{\mathrm{jac}}\mathbf{x} + B_{\mathrm{jac}}\mathbf{u} + f_{\mathrm{nl}}(\mathbf{x}, \mathbf{u}),$$

for some function $f_{\mathrm{nl}}$. Taking gradients,

$$\nabla f_{\mathrm{nl}}(\mathbf{x}, \mathbf{u}) = \nabla_{\mathbf{x},\mathbf{u}} G_{\mathrm{nl}}(\mathbf{x}, \mathbf{u})\big|_{(\mathbf{x},\mathbf{u})=(\mathbf{x},\mathbf{u})} - \nabla G_{\mathrm{nl}}(\mathbf{x}, \mathbf{u})\big|_{(\mathbf{x},\mathbf{u})=(0,0)}.$$

Hence, for all $\mathbf{x}$, $\mathbf{u}$ such that the smoothness assumption holds, we get that

$$\|\nabla_{\mathbf{x},\mathbf{u}} f_{\mathrm{nl}}(\mathbf{x}, \mathbf{u})\|_{\mathrm{op}} \ \leq \ \beta_{\mathrm{nl}}(\|\mathbf{x}\| + \|\mathbf{u}\|).$$

Next, by Taylor's theorem,

$$f_{\mathrm{nl}}(\mathbf{x}, \mathbf{u}) = f_{\mathrm{nl}}(0,0) + \int_0^1 \nabla f_{\mathrm{nl}}(t \cdot \mathbf{x}, t \cdot \mathbf{u}) \begin{bmatrix} \mathbf{x} \\ \mathbf{u} \end{bmatrix} dt,$$

and since $f_{\mathrm{nl}}(0,0) = 0$, we can bound,

$$
\begin{aligned}
\|f_{\mathrm{nl}}(\mathbf{x}, \mathbf{u})\| &\leq \int_0^1 \|\nabla f_{\mathrm{nl}}(t \cdot \mathbf{x}, t \cdot \mathbf{u})\|_{\mathrm{op}}\| \begin{bmatrix} \mathbf{x} \\ \mathbf{u} \end{bmatrix} \| dt \\
&\leq \int_0^1 \beta t \cdot (\|\mathbf{x}\| + \|\mathbf{u}\|)^2 dt. \\
&\leq 2\beta(\|\mathbf{x}\|^2 + \|\mathbf{u}\|^2) \int_0^1 t \cdot dt \\
&= \beta(\|\mathbf{x}\|^2 + \|\mathbf{u}\|^2).
\end{aligned}
$$

## C  Implementing $\varepsilon$-Eval, $\varepsilon$-Grad, and Search Algorithms

We conclude by discussing the implementations and relevant sample complexities of the noisy gradient and function evaluation methods, $\varepsilon$-Grad and $\varepsilon$-Eval. We then use these to establish the runtime and correctness of the noisy binary and random search algorithms.

We remark that throughout our analysis, we have assumed that $\varepsilon$-Grad and $\varepsilon$-Eval succeed with probability 1. This is purely for the sake of simplifying our presentation. As established in Fazel et al. [2018], the relevant estimators in this section all return $\varepsilon$ approximate solutions with probability $1 - \delta$, where $\delta$ factors into the sample complexity polynomially. Therefore, it is easy to union bound and get a high probability guarantee. We omit these union bounds for the sake of simplifying the presentation.

### C.1  Implementing $\varepsilon$-Eval

**Linear setting.**    From the analysis in Fazel et al. [2018], we know that if $J_{\mathrm{lin}}(K \mid \gamma) < \infty$, then

$$J_{\mathrm{lin}}(K \mid \gamma) \approx \frac{1}{N} \sum_{i=1}^N J_{\mathrm{lin}}^{(H)}(K \mid \gamma, \mathbf{x}^{(i)}), \quad \mathbf{x}^{(i)} \sim r \cdot \mathcal{S}^{d_x - 1}, \tag{C.1}$$

where $J_{\mathrm{lin}}^{(H)}$ is the length $H$, finite horizon cost of $K$:

$$J_{\mathrm{lin}}^{(H)}(K \mid \gamma, \mathbf{x}) = \sum_{j=0}^{H-1} \gamma^t \cdot \left(\mathbf{x}_t^\top Q \mathbf{x}_t + \mathbf{u}_t^\top R \mathbf{u}_t\right), \quad \mathbf{u}_t = K \mathbf{x}_t, \quad \mathbf{x}_{t+1} = A \mathbf{x}_t + B \mathbf{u}_t, \quad \mathbf{x}_0 = \mathbf{x}.$$

More formally, if $J_{\mathrm{lin}}(K \mid \gamma)$ is smaller than a constant $c$, Lemma 26 from Fazel et al. [2018] states that it suffices to set $N$ and $H$ to be polynomials in $\|A\|_{\mathrm{op}}, \|B\|_{\mathrm{op}}, J_{\mathrm{lin}}(K \mid \gamma)$ and $c/\varepsilon$ in order to have an $\varepsilon$-approximate estimate of $J_{\mathrm{lin}}(K \mid \gamma)$ with high probability.

On the other hand, if $J_{\mathrm{lin}}(K \mid \gamma)$ is larger than $c$ (recall that the costs should never be larger than some universal constant times $M_{\mathrm{lin}}$ during discount annealing), the following lemma proves we can detect that this is the case by setting $N$ and $H$ to be polynomials in $\|A\|_{\mathrm{op}}, \|B\|_{\mathrm{op}}$ and $c$. This argument follows from the following two lemmas:

**Lemma C.1.** *Fix a constant $c$, and take $H = 4c$. Then for any $K$ (possibly even with $J_{\mathrm{lin}}(K \mid \gamma) = \infty$),*

$$\mathbb{P}_{\mathbf{x} \sim \sqrt{d_x} \cdot \mathcal{S}^{d_x - 1}} \left[ J_{\mathrm{lin}}^{(H)}(K \mid \gamma, \mathbf{x}) \geq \frac{\min\left\{\frac{1}{2} J_{\mathrm{lin}}(K \mid \gamma), c\right\}}{d_x} \right] \geq \frac{1}{10}.$$

Setting $c = \alpha M_{\mathrm{lin}}$ for some universal constant $\alpha$, we get that all calls to the $\varepsilon$-Eval oracle during discount annealing can be implemented with polynomially many samples. Using standard arguments, we can boost this result to hold with high probability by running $N$ independent trials, where $N$ is again a polynomial in the relevant problem parameters.

**Nonlinear setting.** Next, we sketch why the same estimator described in Eq. (C.1) also works in the nonlinear case if we replace $J_{\text{lin}}^{(H)}(K \mid \gamma)$ with the analogous finite horizon cost, $J_{\text{nl}}^{(H)}(K \mid \gamma, r_\star)$, for the nonlinear objective. By Lemma B.3 and Proposition 3.3 if the nonlinear cost is small, then costs on the nonlinear system and the linear system are *pointwise* close. Therefore, previous concentration analysis for the linear setting from Fazel et al. [2018] can be easily carried over in order to implement $\varepsilon$-Eval where $N$ and $H$ are depend polynomially on the relevant problem parameters.

On the other hand if the nonlinear cost is large, then the cost on the linear system must also be large. Recall that if the linear cost was bounded by a constant, then the costs of both systems would be pointwise close by Proposition 3.3. By Proposition C.5, we know that the $H$-step nonlinear cost is lower bounded by the cost on the linear system. Since we can always detect that the cost on the linear system is larger than a constant using polynomially many samples as per Lemma C.1, with high probability we can also detect if the cost on the nonlinear system is large using again only polynomially many samples.

## C.2 Implementing $\varepsilon$-Grad

**Linear setting.** Just like in the case of $\varepsilon$-Eval, Fazel et al. [2018] (Lemma 30) prove that

$$\nabla J_{\text{lin}}(K \mid \gamma) \approx \frac{1}{N}\sum\nolimits_{i=1}^{N} \nabla J_{\text{lin}}^{(H)}(K \mid \gamma, \mathbf{x}^{(i)}), \quad \mathbf{x}^{(i)} \sim r \cdot \mathcal{S}^{d_x-1} \tag{C.2}$$

where $N$ and $H$ only need to be made polynomial large in the relevant problem parameter and $1/\varepsilon$ in order to get an $\varepsilon$ accurate approximation of the gradient. Note that we can safely assume that $J_{\text{lin}}(K \mid \gamma)$ is finite (and hence the gradient is well defined) since we can always run the test outlined in Lemma C.1 to get a high probability guarantee of this fact. In order to approximate the gradients $\nabla J_{\text{lin}}^{(H)}(K \mid \gamma, \mathbf{x})$, one can employ standard techniques from Flaxman et al. [2005]. We refer the interested reader to Appendix D in Fazel et al. [2018] for further details.

**Nonlinear setting.** Lastly, we remark that, as in the case of $\varepsilon$-Eval, Lemma B.6 establishes that the gradients between the linear and nonlinear system are again pointwise close if the cost on the linear system is bounded. In the proof of Theorem 2, we established that during all iterations of discount annealing, the cost on the linear system during executions of policy gradients is bounded by $M_{\text{nl}}$. Therefore, the analysis from Fazel et al. [2018] can be ported over to show that $\varepsilon$-approximate gradients of the nonlinear system can be computed using only polynomially many samples in the relevant problem parameters.

## C.3 Auxiliary results

**Lemma C.2.** *Fix any constant $c \geq 1$. Then, for*

$$P_H = \sum_{j=0}^{H-1} \left((\sqrt{\gamma}(A+BK))^t\right)^\top (Q + K^\top RK)(\sqrt{\gamma}(A+BK))^t$$

*the following relationship holds*

$$J_{\text{lin}}^{(H)}(K \mid \gamma) \geq \min\left\{\frac{1}{2}J_{\text{lin}}(K \mid \gamma), c\right\}, \quad \forall H \geq 4c,$$

*where $J_{\text{lin}}(K \mid \gamma)$ may be infinite.*

*Proof.* We have two cases. In the first case $A_{\text{cl}} := \sqrt{\gamma}(A+BK)$ has an eigenvalue $\lambda \geq 1$. Letting $\mathbf{v}$ denote such a corresponding eigenvector of unit norm, one can verify that $\text{tr}(P_H) \geq \mathbf{v}^\top P_H \mathbf{v} \geq \sum_{i=0}^{H-1} \lambda_i^2 \geq H$, which is at least $c$ by assumption.

In the second case, $A_{\text{cl}}$ is stable, so $P_{K,\gamma}$ exists and its trace is $J_{\text{lin}}(K \mid \gamma)$ (see Fact A.2). Then, for $Q_K = Q + K^\top RK$,

$$P_H = \sum_{i=0}^{H-1} A_{\text{cl}}^{i\top} Q_K A_{\text{cl}}^i,$$

we show that if $\mathrm{tr}(P_H) \leq c$, then $\mathrm{tr}(P_H) \geq \frac{1}{2}\mathrm{tr}(P_{K,\gamma})$.

To show this, observe that if $\mathrm{tr}(P_H) \leq c$, then $\mathrm{tr}(P_H) \leq H/4$. Therefore, by the pidgeonhole principle (and the fact that $\mathrm{tr}(Q_K) \geq 1$), there exists some $t \in \{1, \ldots, H\}$ such that $\mathrm{tr}(A_{\mathrm{cl}}^{t\top} Q_K A_{\mathrm{cl}}^t) \leq 1/4$. Since $Q_K \succeq I$, this means that $\mathrm{tr}(A_{\mathrm{cl}}^{t\top} A_{\mathrm{cl}}^t) = \|A_{\mathrm{cl}}^t\|_{\mathrm{F}}^2 \leq 1/4$ as well. Therefore, letting $P_t$ denote the $t$-step value function from Eq. (C.3), the identity $P_{K,\gamma} = \sum_{n=0}^{\infty} A_{\mathrm{cl}}^{nt\top} P_t A_{\mathrm{cl}}^{nt}$ means that

$$\mathrm{tr}(P_{K,\gamma}) = \mathrm{tr}\left(\sum_{n=0}^{\infty} A_{\mathrm{cl}}^{nt\top} P_t A_{\mathrm{cl}}^{nt}\right)$$

$$\leq \mathrm{tr}(P_t) + \|P_t\| \sum_{n \geq 1} \|A_{\mathrm{cl}}^{nt}\|_{\mathrm{F}}^2$$

$$\leq \mathrm{tr}(P_t) + \|P_t\| \sum_{n \geq 1} \|A_{\mathrm{cl}}^t\|_{\mathrm{F}}^{2n}$$

$$\leq \mathrm{tr}(P_t) + \|P_t\| \sum_{n \geq 1} (1/2)^n$$

$$\leq 2\mathrm{tr}(P_t) \leq 2\mathrm{tr}(P_H),$$

where in the last line we used $t \leq H$. This completes the proof. $\qquad\square$

**Lemma C.3.** *Let* $\mathbf{u}, \mathbf{v} \overset{i.i.d}{\sim} \mathcal{S}^{d_x-1}$ *then,* $\mathbb{P}\left[(\mathbf{u}^\top \mathbf{v})^2 \geq 1/d_x\right] \geq .1$.

*Proof.* The statement is clearly true for $d_x = 1$, therefore we focus on the case where $d_x \geq 2$. Without loss of generality, we can take $\mathbf{u}$ to be the first basis vector $\mathbf{e}_1$ and let $\mathbf{v} = Z/\|Z\|$ where $Z \sim \mathcal{N}(0, I)$. From these simplifications, we observe that $(\mathbf{v}^\top \mathbf{u})^2$ is equal in distribution to the following random variable,

$$\frac{Z_1^2}{\sum_{i=1}^d Z_i^2},$$

where each $Z_i^2$ is a chi-squared random variable. Using this equivalency, we have that for arbitrary $M > 0$,

$$\mathbb{P}\left[(\mathbf{u}^\top \mathbf{v})^2 \geq 1/d_x\right] = \mathbb{P}\left[Z_1^2 \geq \frac{\sum_{i=2}^{d_x} Z_i^2}{d_x - 1}\right]$$

$$= \mathbb{P}\left[Z_1^2 \geq \frac{\sum_{i=2}^{d_x} Z_i^2}{d_x - 1} \,\middle|\, \sum_{i=2}^{d_x} Z_i^2 \leq M\right] \mathbb{P}\left[\sum_{i=2}^{d_x} Z_i^2 \leq M\right]$$

$$+ \mathbb{P}\left[Z_1^2 \geq \frac{\sum_{i=2}^{d_x} Z_i^2}{d_x - 1} \,\middle|\, \sum_{i=2}^{d} Z_i^2 > M\right] \mathbb{P}\left[\sum_{i=2}^{d_x} Z_i^2 > M\right]$$

$$\geq \mathbb{P}\left[Z_1^2 \geq \frac{\sum_{i=2}^{d_x} Z_i^2}{d_x - 1} \,\middle|\, \sum_{i=2}^{d} Z_i^2 \leq M\right] \mathbb{P}\left[\sum_{i=2}^{d_x} Z_i^2 \leq M\right]$$

$$\geq \mathbb{P}\left[Z_1^2 \geq \frac{M}{d_x - 1}\right] \mathbb{P}\left[\sum_{i=2}^{d_x} Z_i^2 \leq M\right].$$

Setting $M = 2(d_x - 1)$, we get that,

$$\mathbb{P}\left[(\mathbf{u}^\top \mathbf{v})^2 \geq 1/d_x\right] \geq \mathbb{P}\left[Z_1^2 \geq 2\right] \mathbb{P}\left[\sum_{i=2}^{d_x} Z_i^2 \leq 2(d_x - 1)\right].$$

From a direct computation,

$$\mathbb{P}\left[Z_1^2 \geq 2\right] \geq .15.$$

To bound the last term, if $Y$ is a chi-squared random variable with $k$ degrees of freedom, by Lemma 1 in Laurent and Massart [2000],

$$\mathbb{P}[Y \geq k + 2\sqrt{kx} + x] \leq \exp(-x).$$

Setting $x = 2\sqrt{2(d_x - 1)k} + 2d_x + k - 2$ we get that $k + 2\sqrt{kx} + x = 2(d_x - 1)$. Substituting in $k = d_x - 1$, we conclude that,

$$\mathbb{P}\left[\sum_{i=2}^{d_x} Z_i^2 \leq 2(d_x - 1)\right] \geq 1 - \exp(-2\sqrt{2}(d_x - 1) - 3d_x + 3),$$

which is greater than .99 for $d_x \geq 2$. $\qquad\square$

**Lemma C.1 (restated).** *Fix a constant $c$, and take $H = 4c$. Then for any $K$ (possibly even with $J_{\mathrm{lin}}(K \mid \gamma) = \infty$),*

$$\mathbb{P}_{\mathbf{x} \sim \sqrt{d_x} \cdot \mathcal{S}^{d_x - 1}}\left[J_{\mathrm{lin}}^{(H)}(K \mid \gamma, \mathbf{x}) \geq \frac{\min\left\{\frac{1}{2}J_{\mathrm{lin}}(K \mid \gamma), c\right\}}{d_x}\right] \geq \frac{1}{10}.$$

*Proof.* Observe that for the finite horizon value matrix $P_H = \sum_{i=0}^{H-1}(A+BK)^{i\top}(Q+K^\top RK)(A+BK)^i$, we have $J_{\mathrm{lin}}^{(H)}(K \mid \gamma, \mathbf{x}) = \mathbf{x}^\top P_H \mathbf{x}$. We now observe that, since $P_H \succeq 0$, it has a (possibly non-unique) top eigenvector $v_1$ for which

$$\begin{aligned}
J_{\mathrm{lin}}^{(H)}(K \mid \gamma, \mathbf{x}) = \mathbf{x}^\top P_H \mathbf{x} &\geq \langle v_1, \frac{\mathbf{x}}{\sqrt{d_x}}\rangle^2 \cdot d_x \|P_H\| \\
&\geq \langle v_1, \frac{\mathbf{x}}{\sqrt{d_x}}\rangle^2 \cdot \mathrm{tr}(P_H) \\
&= \underbrace{\langle v_1, \frac{\mathbf{x}}{\sqrt{d_x}}\rangle^2}_{:=Z} \cdot J_{\mathrm{lin}}^{(H)}(K \mid \gamma)
\end{aligned}$$

Since $\mathbf{x}/\sqrt{d_x} \sim \mathcal{S}^{d_x - 1}$, Lemma C.3 ensures that $\mathbb{P}[Z \geq 1/d_x] \geq 1/10$. Hence,

$$\mathbb{P}_{\mathbf{x} \sim \sqrt{d_x} \cdot \mathcal{S}^{d_x - 1}}\left[J_{\mathrm{lin}}^{(H)}(K \mid \gamma, \mathbf{x}) \geq \frac{J_{\mathrm{lin}}^{(H)}(K \mid \gamma)}{d_x}\right] \geq \frac{1}{10}.$$

The bound now follows from invoking Lemma C.2 to lower bound $J_{\mathrm{lin}}^{(H)}(K \mid \gamma, \mathbf{x}) \geq \min\left\{\frac{1}{2}J_{\mathrm{lin}}(K \mid \gamma), c\right\}$ provided $H \geq 4c$.

$\qquad\square$

In the following lemmas, we define $P_H$ to be the following matrix where $K$ is any state-feedback controller.

$$P_H = \sum_{j=0}^{H-1}\left((\sqrt{\gamma}(A+BK))^t\right)^\top (Q + K^\top RK)(\sqrt{\gamma}(A+BK))^t \tag{C.3}$$

Similarly, we let $J_{\mathrm{nl}}^{(H)}(K \mid \gamma, \mathbf{x}_0)$ be the horizon $H$ cost of the nonlinear dynamical system:

$$J_{\mathrm{nl}}^{(H)}(K \mid \mathbf{x}, \gamma) := \sum_{t=0}^{H-1} \mathbf{x}_t^\top Q \mathbf{x}_t + \mathbf{u}_t^\top R \mathbf{u}_t \tag{C.4}$$

$$\text{s.t } \mathbf{u}_t = K\mathbf{x}_t, \quad \mathbf{x}_{t+1} = \sqrt{\gamma} \cdot G_{\mathrm{nl}}(\mathbf{x}_t, \mathbf{u}_t), \quad \mathbf{x}_0 = \mathbf{x}. \tag{C.5}$$

And again overloading notation like before, we let $J_{\mathrm{nl}}^{(H)}(K \mid \gamma, r) := \mathbb{E}_{\mathbf{x} \sim r \cdot \mathcal{S}^{d_x - 1}}\left[J_{\mathrm{nl}}^{(H)}(K \mid \gamma, \mathbf{x})\right] \times \frac{d_x}{r^2}$.

**Lemma C.4.** *Fix a horizon $H$, constant $\alpha \in (0, d_x)$, $\mathbf{x}_0 \in \mathbb{R}^{d_x}$, and suppose that*

$$\|\mathbf{x}_0\|^2 \cdot \frac{\alpha}{2H^2\beta_{\mathrm{nl}}^2(1 + \|K\|^2)d_x} \leq r_{\mathrm{nl}}^4.$$

*Furthermore, define $\mathcal{X}_\alpha := \{\mathbf{x}_0 \in \mathbb{R}^{d_x} : \langle \mathbf{x}_0, \mathbf{v}_{\max}(P_H)\rangle^2 \geq \alpha\|\mathbf{x}_0\|^2/d_x\}$. Then, if $\mathbf{x}_0 \in \mathcal{X}_\alpha$, it holds that*

$$J_{\mathrm{nl}}^{(H)}(K \mid \gamma, \mathbf{x}_0) \geq \min\left\{\frac{\alpha\|\mathbf{x}_0\|^2}{4d_x^2} \cdot J_{\mathrm{lin}}^{(H)}(K \mid \mathbf{x}_0, \gamma), \sqrt{\frac{\alpha}{d_x}}\frac{\|\mathbf{x}_0\|}{2H\beta_{\mathrm{nl}}(1 + \|K\|)}\right\}.$$

*Proof.* Fix a constant $r_1 \leq r_{\mathrm{nl}}$ to be selected. Throughout, we use the shorthand $\beta_1^2 = \beta_{\mathrm{nl}}^2(1 + \|K\|^2)$ We consider two cases.

**Case 1:** The initial point $\mathbf{x}_0$ is such that it is always the case that $\|\mathbf{x}_t\| \leq r_1$ for all $t \in \{0, 1, \ldots, H-1\}$ and $J_{\mathrm{nl}}^{(H)}(K \mid \gamma, \mathbf{x}_0) \leq r_1^2$. Observe that we can write the nonlinear dynamics as

$$\mathbf{x}_{t+1} = \sqrt{\gamma}(A + BK)\mathbf{x}_t + \mathbf{w}_t,$$

where $\mathbf{w}_t = \sqrt{\gamma} \cdot f_{\mathrm{nl}}(\mathbf{x}_t, K\mathbf{x}_t)$. We now write:

$$\mathbf{x}_t = \mathbf{x}_{t;0} + \mathbf{x}_{t;w}, \text{ where}$$

$$\mathbf{x}_{t;0} = \sqrt{\gamma}(A + BK)^t \mathbf{x}_0$$

$$\mathbf{x}_{t;w} = \sum_{i=0}^{t-1} \gamma^{i/2}(A + BK)^i \mathbf{w}_{t-i}.$$

Then, setting $Q_K = Q + K^\top R K$,

$$J_{\mathrm{nl}}^{(H)}(K \mid \gamma, \mathbf{x}_0) = \sum_{t=0}^{H-1} \mathbf{x}_t^\top Q_K \mathbf{x}_t$$

$$= \sum_{t=0}^{H-1} \left( \mathbf{x}_{t;0}^\top Q_K \mathbf{x}_{t;0} + 2\mathbf{x}_{t;0}^\top Q_K \mathbf{x}_{t;w} + \mathbf{x}_{t;w}^\top Q_K \mathbf{x}_{t;w} \right)$$

$$\geq \frac{1}{2} \sum_{t=0}^{H-1} \mathbf{x}_{t;0}^\top Q_K \mathbf{x}_{t;0} - \sum_{t=0}^{H-1} \mathbf{x}_{t;w}^\top Q_K \mathbf{x}_{t;w}$$

$$= \frac{1}{2}\mathbf{x}_0^\top P_H \mathbf{x}_0 - \sum_{t=0}^{H-1} \mathbf{x}_{t;w}^\top Q_K \mathbf{x}_{t;w},$$

where (a) the last inequality uses the elementary inequality $\langle \mathbf{v}_1, \Sigma \mathbf{v}_1 \rangle + \langle \mathbf{v}_2, \Sigma \mathbf{v}_2 \rangle + 2\langle \mathbf{v}_1, \Sigma \mathbf{v}_2 \rangle \geq \frac{1}{2}\langle \mathbf{v}_1, \Sigma \mathbf{v}_1 \rangle - \langle \mathbf{v}_2, \Sigma \mathbf{v}_2 \rangle$ for any pair of vectors $\mathbf{v}_1, \mathbf{v}_2$ and $\Sigma \succeq 0$, and (b) the last inequality recognizes how

$$\sum_{t=0}^{H-1} \mathbf{x}_{t;0}^\top Q_K \mathbf{x}_{t;0} = J_{\mathrm{lin}}^{(H)}(K \mid \mathbf{x}_0, \gamma) = \mathbf{x}_0^\top P_H \mathbf{x}_0$$

for $P_H$ defined above in [Eq. (C.3)]. Moreover, for any $t$,

$$\sum_{t=0}^{H-1} \mathbf{x}_{t;w}^\top Q_K \mathbf{x}_{t;w} = \sum_{t=0}^{H-1} \sum_{i=0}^{t-1} \sum_{j=0}^{t-1} \mathbf{w}_{t-i}^\top \gamma^{(i+j)/2}((A+BK)^i)^\top Q_K (A+BK)^j \mathbf{w}_{t-j}$$

$$\leq H \sum_{t=0}^{H-1} \sum_{i=0}^{t-1} \mathbf{w}_{t-i}^\top \gamma^i ((A+BK)^i)^\top Q_K (A+BK)^i \mathbf{w}_{t-i}$$

$$= H \sum_{t=0}^{H-2} \mathbf{w}_t^\top \left( \sum_{i=0}^{H-t-1} \gamma^i ((A+BK)^i)^\top Q_K (A+BK)^i \right) \mathbf{w}_t$$

$$= H \sum_{t=0}^{H-2} \mathbf{w}_t^\top P_{H-t-1} \mathbf{w}_t \leq H^2 \|P_H\|_{\mathrm{op}} \max_{t \in \{0, \ldots, H-2\}} \|\mathbf{w}_t\|^2.$$

Now, because $\|\mathbf{x}_t\| \leq r_1 \leq r_{\mathrm{nl}}$, [Lemma 3.1] lets use bound $\|\mathbf{w}_t\|^2 \leq \beta_{\mathrm{nl}}^2(1 + \|K\|^2)\|\mathbf{x}_t\|^4 \leq \beta_1^2 r_1^4$, where we adopt the previously defined shorthand $\beta_1^2 = \beta_{\mathrm{nl}}^2(1 + \|K\|^2)$. Therefore,

$$J_{\mathrm{nl}}^{(H)}(K \mid \gamma, \mathbf{x}_0) \geq \frac{1}{2}\mathbf{x}_0^\top P_H \mathbf{x}_0 - H^2 \beta_1^2 r_1^4 \|P_H\|_{\mathrm{op}}.$$

Next, if $\mathbf{x}_0 \in \mathcal{X}_\alpha$,

$$J_{\mathrm{nl}}^{(H)}(K \mid \gamma, \mathbf{x}_0) \geq \frac{\alpha}{2d_x}\|\mathbf{x}_0\|^2 \|P_H\| - H^2 \beta_1^2 r_1^4 \|P_H\|_{\mathrm{op}}$$

$$= \|P_H\| \left( \frac{\alpha}{2d_x}\|\mathbf{x}_0\|^2 - H^2 \beta_1^2 r_1^4 \right).$$

In particular, selecting $r_1^4 = \frac{\alpha}{4\beta_1^2 d_x H^2}\|\mathbf{x}_0\|^2$ (which ensures $r_1 \leq r_\star$ by the conditions of the lemma), it holds that,

$$J_{\mathrm{nl}}^{(H)}(K \mid \gamma, \mathbf{x}_0) \geq \frac{\|P_H\|\alpha}{4d_x}\|\mathbf{x}_0\|^2 \geq \frac{\mathrm{tr}(P_H)\alpha}{4d_x^2}\|\mathbf{x}_0\|^2 = \frac{\alpha J_{\mathrm{lin}}^{(H)}(K, \gamma)}{4d_x^2}\|\mathbf{x}_0\|^2.$$

**Case 2:** The initial point $\mathbf{x}_0$ is such that it is always the case that either $\|\mathbf{x}_t\| \geq r_1$ for all $t \in \{0, 1, \ldots, H-1\}$ or $J_{\mathrm{nl}}^{(H)}(K \mid \gamma, \mathbf{x}_0) \geq r_1^2$. Therefore, in either case, $J_{\mathrm{nl}}^{(H)}(K \mid \gamma, \mathbf{x}_0) \geq r_1^2$. For our choice of $r_1$, this gives

$$J_{\mathrm{nl}}^{(H)}(K \mid \gamma, \mathbf{x}_0) \geq \sqrt{\frac{\alpha\|x_0\|^2}{4d_x H^2 \beta_1^2}} = \sqrt{\frac{\alpha\|x_0\|^2}{4d_x H^2 \beta_{\mathrm{nl}}^2(1 + \|K\|^2)}} \geq \sqrt{\frac{\alpha}{d_x}}\frac{\|x_0\|}{2H\beta_{\mathrm{nl}}(1 + \|K\|)}.$$

Combining the cases, we have

$$J_{\mathrm{nl}}^{(H)}(K \mid \gamma, \mathbf{x}_0) \geq \min\left\{\frac{\alpha\|\mathbf{x}_0\|^2}{4d_x^2} \cdot J_{\mathrm{lin}}^{(H)}(K \mid \mathbf{x}_0, \gamma), \sqrt{\frac{\alpha}{d_x}}\frac{\|x_0\|}{2H\beta_{\mathrm{nl}}(1 + \|K\|)}\right\}.$$

$\square$

**Proposition C.5.** *Let $c$ be a given (integer) tolerance $c$ and $\mathcal{X}_\alpha := \{\mathbf{x} \in \mathbb{R}^d : \langle \mathbf{x}, \mathbf{v}_{\max}(P_H)\rangle^2 \geq \alpha\|\mathbf{x}\|^2/d_x\}$ be defined as in Lemma C.4. Then, for $\alpha \in (0, 2]$, $H = 4c$, and $\mathbf{x}_0 \in \mathcal{X}_\alpha$ satisfying,*

$$\|\mathbf{x}_0\|^2 \leq c^2 r_{\mathrm{nl}}^4, \quad and \quad \|\mathbf{x}_0\| \leq \frac{d_x}{64c^2\beta_{\mathrm{nl}}(1 + \|K\|)}, \tag{C.6}$$

*it holds that:*

$$\frac{J_{\mathrm{nl}}^{(H)}(K \mid \gamma, \mathbf{x}_0)}{\|\mathbf{x}_0\|^2} \geq \frac{\alpha}{8d_x^2}\min\left\{J_{\mathrm{lin}}(K \mid \gamma), c\right\}.$$

*Moreover, for $r \leq \min\{cr_{\mathrm{nl}}^2, \frac{d_x}{64c^2\beta_{\mathrm{nl}}(1+\|K\|)}\}$,*

$$J_{\mathrm{nl}}^{(H)}(K \mid \gamma, r) \geq \frac{1}{80d_x^2}\min\left\{J_{\mathrm{lin}}(K \mid \gamma), c\right\}.$$

*Proof.* From Lemma C.4, it holds that for $H = 4c$,

$$\|\mathbf{x}_0\|^2 \cdot \frac{\alpha}{32c^2\beta_{\mathrm{nl}}^2(1 + \|K\|^2)d_x} \leq r_{\mathrm{nl}}^4 \tag{C.7}$$

and hence

$$J_{\mathrm{nl}}^{(H)}(K \mid \gamma, \mathbf{x}_0) \geq \min\left\{\frac{\alpha\|\mathbf{x}_0\|^2}{4d_x^2} \cdot J_{\mathrm{lin}}^{(H)}(K \mid \mathbf{x}_0, \gamma), \sqrt{\frac{\alpha}{d_x}}\frac{\|\mathbf{x}_0\|}{8c\beta_{\mathrm{nl}}(1 + \|K\|)}\right\}.$$

Note that, due to $\alpha/d_x \leq 1$, $\beta_{\mathrm{nl}}^2 \geq 1$, Eq. (C.7) holds as soon as $\|\mathbf{x}_0\| \leq c^2 r_{\mathrm{nl}}^4$. By Lemma C.2, it then follows that:

$$J_{\mathrm{nl}}^{(H)}(K \mid \gamma, \mathbf{x}_0) \geq \min\left\{\frac{\alpha\|\mathbf{x}_0\|^2}{4d_x^2} \cdot \min\left\{\frac{1}{2}J_{\mathrm{lin}}(K \mid \gamma), c\right\}, \sqrt{\frac{\alpha}{d_x}}\frac{\|\mathbf{x}_0\|}{8c\beta_{\mathrm{nl}}(1 + \|K\|)}\right\}$$

$$\geq \min\left\{\frac{\alpha\|\mathbf{x}_0\|^2}{8d_x^2}\min\left\{J_{\mathrm{lin}}(K \mid \gamma), c\right\}, \sqrt{\frac{\alpha}{d_x}}\frac{\|\mathbf{x}_0\|}{8c\beta_{\mathrm{nl}}(1 + \|K\|)}\right\}$$

$$= \frac{\alpha\|\mathbf{x}_0\|^2}{8d_x^2}\min\left\{J_{\mathrm{lin}}(K \mid \gamma), c, \left(\frac{\alpha\|\mathbf{x}_0\|^2}{8d_x^2}\right)^{-1}\sqrt{\frac{\alpha}{d_x}}\frac{\|\mathbf{x}_0\|}{8c\beta_{\mathrm{nl}}(1 + \|K\|)}\right\}.$$

Simplifying, the last term gives

$$\left(\frac{\alpha\|\mathbf{x}_0\|^2}{8d_x^2}\right)^{-1}\sqrt{\frac{\alpha}{d_x}}\frac{\|\mathbf{x}_0\|}{8c\beta_{\mathrm{nl}}(1 + \|K\|)} = \frac{1}{c\|\mathbf{x}_0\|}\cdot\left(\frac{d_x^3}{\alpha}\right)^{1/2}\frac{1}{64\beta_{\mathrm{nl}}(1 + \|K\|)} \geq \frac{1}{c\|\mathbf{x}_0\|}\cdot\frac{d_x}{64\beta_{\mathrm{nl}}(1 + \|K\|)},$$

| Noisy Binary Search | Noisy Random Search |
|---|---|
| **Require:** $\overline{f}_1$ and $\overline{f}_2$ as defined in Lemma C.6. | **Require:** $\overline{f}_1, \overline{f}_2$ as defined in Lemma C.7. |
| **Initialize:** $b_0 \leftarrow 0, u_0 \leftarrow 1, c \geq \overline{f}_2 + 2\alpha$ for $\alpha := \alpha = \min\{|f_1 - \overline{f}_1|, |f_2 - \overline{f}_2|\}$, $\varepsilon \in (0, \alpha/2)$ | **Initialize:** $c \geq \overline{f}_2 + 2\alpha$ for $\alpha := \min\{|f_1 - \overline{f}_1|, |f_2 - \overline{f}_2|\}$, $\varepsilon \in (0, \alpha/2)$ |
| **For** $t = 1, \ldots$ 
 1. Query $a_t \leftarrow \varepsilon\text{-Eval}(f, x_t, c)$ where $$x_t = \frac{b_t + u_t}{2}$$ 2. If $a_t > \overline{f}_2 + \varepsilon$, update $u_t \leftarrow x_t$ and $b_t \leftarrow b_{t-1}$ 
 3. Else if, $a_t < \overline{f}_1 + \varepsilon$, update $b_t \leftarrow x_t$ and $u_t \leftarrow u_{t-1}$ 
 4. Else, break and return $x_t$ | **For** $t = 1, \ldots$ 
 1. Sample $x$ uniformly at random from $[0, 1]$ 
 2. Query $a \leftarrow \varepsilon\text{-Eval}(f, x, c)$ 
 3. If $a \in [\overline{f}_1, \overline{f}_2]$, break and return $x$, |

Figure 2: Algorithms for 1 dimensional search used as subroutines for Step 4 in the discount annealing algorithm.

where we use $d_x/\alpha \geq 1$. Thus, for

$$\|\mathbf{x}_0\| \leq \frac{d_x}{64c^2\beta_{\mathrm{nl}}(1 + \|K\|)},$$

the third term in the minimum is at most $c$, so that

$$J_{\mathrm{nl}}^{(H)}(K \mid \gamma, \mathbf{x}_0) \geq \frac{\alpha\|\mathbf{x}_0\|^2}{8d_x^2} \min\left\{J_{\mathrm{lin}}(K \mid \gamma), c\right\}.$$

Lastly, using Lemma C.3,

$$J_{\mathrm{nl}}^{(H)}(K \mid \gamma, r) = \frac{d_x}{r^2} \cdot \mathbb{E}_{\mathbf{x}_0 \sim r \cdot \mathcal{S}^{d_x - 1}} J_{\mathrm{nl}}^{(H)}(K \mid \gamma, \mathbf{x}_0)$$

$$\geq \mathbb{P}[x_0 \in \mathcal{X}_1] \cdot \frac{1}{8d_x^2} \min\left\{J_{\mathrm{lin}}(K \mid \gamma), c\right\}$$

$$\geq \frac{1}{80d_x^2} \min\left\{J_{\mathrm{lin}}(K \mid \gamma), c\right\}.$$

$\square$

### C.4 Search analysis

**Lemma C.6.** *Let $f : [0, 1] \to \mathbb{R} \cup \{\infty\}$ be a nondecreasing function over the unit interval. Then, given $\overline{f}_1, \overline{f}_2$ such that $[\overline{f}_1, \overline{f}_2] \subseteq [f_1, f_2]$ and for which there exist $[x_1, x_2] \subseteq [0, 1]$ such that for all $x' \in [x_1, x_2]$, $f(x') \in [\overline{f}_1, \overline{f}_2]$, binary search as defined in Figure 2 returns a value $x_\star \in [0, 1]$ such that $f(x_\star) \in [f_1, f_2]$ in at most $\lceil \log_2(1/\Delta) \rceil$ many iterations where $\Delta = x_2 - x_1$.*

**Lemma C.7.** *Let $f : [0, 1] \to \mathbb{R} \cup \{\infty\}$ be a function over the unit interval. Then, given given $\overline{f}_1, \overline{f}_2$ such that $[\overline{f}_1, \overline{f}_2] \subseteq [f_1, f_2]$ and for which there exist $[x_1, x_2] \subseteq [0, 1]$ such that for all $x' \in [x_1, x_2]$, $f(x') \in [\overline{f}_1, \overline{f}_2]$, with probability $1 - \delta$, noisy random search as defined in Figure 2 returns a value $x_\star \in [0, 1]$ such that $f(x_\star) \in [f_1, f_2]$ in at most $1/\Delta \log(1/\delta)$ many iterations where $\Delta = x_2 - x_1$.*

The analysis of the correctness and runtime of this classical algorithms for search problems in 1 dimension are standard. We omit the proofs for the sake of concision.

# D  Additional Experiments

## D.1  Cart-pole dynamics

The state of the cart-pole is given by $\mathbf{x} = (x, \theta, \dot{x}, \dot{\theta})$, where $x$ and $\dot{x}$ denote the horizontal position and velocity of the cart, and $\theta$ and $\dot{\theta}$ denote the angular position and velocity of the pole, with $\theta = 0$ corresponding to the upright equilibrium. The control input $u \in \mathbb{R}$ corresponds to the horizontal force applied to the cart. The continuous time dynamics are given by

$$\begin{bmatrix} m_p + m_c & -m_p l \cos(\theta) \\ -m_p l \cos(\theta) & m_p l^2 \end{bmatrix} \begin{bmatrix} \ddot{x} \\ \ddot{\theta} \end{bmatrix} = \begin{bmatrix} u - m_p l \sin(\theta)\dot{\theta}^2 \\ m_p g l \sin(\theta) \end{bmatrix}$$

where $m_p$ denotes the mass of the pole, $m_c$ the mass of the cart, $l$ the length of the pendulum, and $g$ acceleration due to gravity. For our experiments, we set all parameters to unity, i.e. $m_p = m_c = l = g = 1$.

## D.2  $\mathcal{H}_\infty$ synthesis

Consider the following linear system

$$\mathbf{x}_{t+1} = A\mathbf{x}_t + B\mathbf{u}_t + \mathbf{w}_t, \quad \mathbf{z}_t = \begin{bmatrix} Q^{1/2}\mathbf{x}_t \\ R^{1/2}\mathbf{u}_t \end{bmatrix}, \tag{D.1}$$

where $\mathbf{w}$ denotes an additive disturbance to the state transition, and $\mathbf{z}$ denotes the so-called performance output. Notice that $\|\mathbf{z}_t\|_2^2 = \mathbf{x}_t^\top Q \mathbf{x}_t + \mathbf{u}_t^\top R \mathbf{u}_t$. The $\mathcal{H}_\infty$ optimal controller minimizes the $\mathcal{H}_\infty$-norm of the closed-loop system from input $\mathbf{w}$ to output $\mathbf{z}$, i.e. the smallest $\eta$ such that

$$\sum_{t=0}^{T} \|\mathbf{z}_t\|_2^2 \leq \eta \sum_{t=0}^{T} \eta \|\mathbf{w}_t\|_2^2$$

holds for all $\mathbf{w}_{0:T}$, all $T$, and $\mathbf{x}_0 = 0$. In essence, the $\mathcal{H}_\infty$ optimal controller minimizes the effect of the worst-case disturbance on the cost. In this setting, additive disturbances $\mathbf{w}$ serve as a crude (and unstructured) proxy for modeling error. We synthesize the $\mathcal{H}_\infty$ optimal controller using Matlab's `hinfsyn` function. For the system (D.1) with perfect state observation, we require only static state feedback $\mathbf{u}_t = K\mathbf{x}_t$ to implement the controller.

## D.3  Difference between LQR and discount annealing as a function of discount

In Figure 3 we plot the error $\|K_{\text{pg}}^*(\gamma) - K_{\text{lin}}^*(\gamma)\|_F$ between the policy $K_{\text{pg}}^*(\gamma)$ returned by policy gradient and the optimal policy $K_{\text{lin}}^*(\gamma)$ for the (damped) linearized system, as a function of the discount $\gamma$ during the discount annealing process. Observe that for small radius of the ball of initial conditions ($r = 0.05$), the optimal controller from policy gradient remains very close to the exact optimal controller for the linearized system; however, for larger radius ($r = 0.7$) the controller from policy gradient differs significantly.

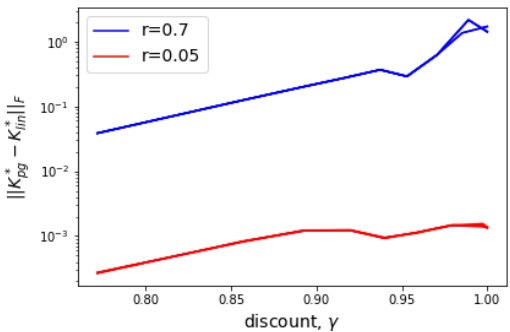

Figure 3: Error between the policy returned by policy gradient and optimal LQR policy for the (damped) linearized system during discount annealing, for two different radius values, $r = 0.05$ and $r = 0.7$, of the ball of initial conditions. Five independent trials are plotted for each radius value. The values across trials highly overlap.