# OpenReview forum: "Stabilizing Dynamical Systems via Policy Gradient Methods"
_NeurIPS.cc/2021/Conference — NeurIPS 2021 Poster_

### Official Review · Reviewer_9Hp6 · 2021-07-16

**Rating:** 7
**Confidence:** 4

**Summary:**

This work gives a method to compute stabilizing controllers given access to noisy gradient evaluations with respect to the policy. The challenge is that if a controller does not stabilize the system, then the cost is infinite and the gradient is undefined. They key insight is to use the discount factor as  a free parameter. Namely, for sufficiently small discount factors, the cost is finite and the gradient can be evaluated. So, the algorithm alternates between taking gradient steps and increasing the discount factor. When a discount factor of 1 is reached, stabilization has been obtained and the problem reduces to a policy gradient method. Bounds on convergence and applications to nonlinear systems are given.



**Limitations And Societal Impact:**

The authors have discussed the limitations. I do not forsee any negative societal impact.

**Main Review:**

This paper is well-written and insightful. The proofs appear to be rigorous. However, as discussed below, the basic idea of using the discount factor for model-free stabilization has appeared independently in the literature. The proofs appear to be very different from the existing work, and the similarities appear to be coincidental. So, overall, this is nice work, the contribution feels a bit smaller as a result.

A very similar model-free methodology has already been devised for computing stabilizing controllers and solving LQR in:

Lamperski, Andrew. "Computing Stabilizing Linear Controllers via Policy Iteration." 2020 59th IEEE Conference on Decision and Control (CDC). IEEE, 2020.

Both are model free, and both interleave policy improvement steps with increases in the discount factor. Both give stabilizing controllers and converge to the LQR solution.

The prior work uses policy-iteration style updates in an off-policy manner. In particular, it can be used on a single trajectory or an offline data set. The current submission uses gradient evaluations, which effectively require restating the system for each new gradient. The current submission does give more explicit convergence bounds and gives applications to nonlinear systems.

Edit based on Response: Based on the response, I can see greater merit and distinction in the work and I will raise my score appropriately.

**Time Spent Reviewing:**

1.5

---

> ### Author Response · Authors · 2021-08-06
> **Author Response**
>
>
> We greatly appreciate you bringing Lamperski 2020 (henceforth Lam2020) to our attention. We were unaware of this recent piece of interesting, related work and surprised to see how our algorithms are quite similar in spirit. We will certainly attribute the idea of iteratively increasing the discount factor to their work and not to claim novelty for this algorithmic principle.
>
> As you point out, there are a number of differences between our work and Lam2020:
>
> 1. Even though Lam2020 does not require knowledge of ‘A,B,Q,R’, it requires a linear inversion step (Eq 16 therein), which estimates a matrix of similar dimension. In this sense, it estimates a convex representation of the Q-function dynamics under a controller policy K. It then uses this representation to perform a closed-form policy improvement step based on the Riccati update. In contrast, our approach is based on policy gradients, which does not require any explicit estimation of the Q-function and is more consistent with popular approaches in the "model-free policy improvement" space.
>
> 2. We provide precise, finite time rates for convergence to stabilizing controller systems, while this other algorithm only has an asymptotic guarantee for linear systems. We note that establishing finite-time rates has been a central focus of the recent literature (as pointed out by the other 2 reviewers).
>
> 3. Our analysis extends to nonlinear systems, whereas Lam2020 applies only to linear systems.
>
> 4. We rigorously address the required tolerances required to implement our method, and show that it can be implemented with access to finite samples from a simulator (e.g. the binary search and policy updates can be updated in a fashion robust to error) .
>
> 5. We conduct experiments evaluating our method and illustrate its empirical effectiveness at a common, nonlinear control benchmark.
>
> With regards to the multiple rollouts, it is true that restarts are indeed necessary to estimate gradients. However, as noted by Lam2020 (see Remark 1), a similar access model is likely needed for his algorithm to work for linear systems that are not controllable.
>
> We look forward to incorporating this paper to our related work and better situating our contributions within the broader literature. Thank you again for pointing this out to us and helping us improve the quality of our manuscript. Please let us know if you have any further questions or concerns about our work.

---

> > ### Comment · Reviewer_9Hp6 · 2021-08-25
> > **I see the points and the merit of the method.**
> >
> > Based on the comment I will raise my score a bit.
> >
> > * Much more precise guarantees than Lam2020
> > * The policy gradient approach has the desirable property for high-dimensional systems that a matrix of size (n+p)x(n+p) need not be estimated. (Here n is the state dimension and p is the input dimension.) Indeed, the gradient estimates are just of size nxp, which could be dramatically smaller.
> > * The methodology seems likely to generalize reasonably well to stochastic settings.

---

### Official Review · Reviewer_YL9s · 2021-07-21

**Rating:** 7
**Confidence:** 3

**Summary:**

This paper studies how to stabilize dynamic systems using policy gradient methods. In the linear setup, a global stabilization problem is addressed. In the nonlinear setup, a local stabilization problem is solved. The basic idea is that a zero control can be used when the discount  factor is sufficiently small, and one can iteratively apply a discounted LQR formulation to increase the discount factor. The core algorithm is "Discount Annealing." Both theory and simulations are provided to justify the proposed method. Overall, this paper provides the first sample complexity result for solving stabilization via policy gradient methods.

**Ethical Concerns:**

I have not seen any such issues.

**Limitations And Societal Impact:**

Yes.

**Main Review:**

Originality: In my opinion, this paper is quite original and considers an important question. Stabilization is a fundamental question for control, and how to solve it via policy gradient is unclear. This paper makes progress in answering such an important question, and I can see there is unique contribution in this paper.

References: This paper missed a large body of literature on policy optimization of linear control problems. The following papers are relevant, and the authors should discuss them.

1. T. Rautert and E. W. Sachs, “Computational design of optimal output feedback controllers,” SIAM Journal on Optimization, 1997.

2. K.M°artensson and A. Rantzer, “Gradient methods for iterative distributed control synthesis,” CDC, 2009.

3. D. Malik, A. Pananjady, K. Bhatia, K. Khamaru, P. Bartlett, and M. Wainwright. "Derivative-free methods for policy optimization: Guarantees for linear quadratic systems," AIstats, 2019.

4. K. Zhang, B. Hu, and T. Basar, “Policy optimization for H2 linear control with H-infinity robustness guarantee: Implicit regularization and global convergence,” L4DC, 2020.

5. L. Furieri, Y. Zheng, and M. Kamgarpour, “Learning the globally optimal distributed LQ regulator,” L4DC, 2020.

6. J. P. Jansch-Porto, B. Hu, and G. Dullerud, “Convergence guarantees of policy optimization methods for Markovian jump linear systems,” ACC, 2020.

7. B. Gravell, P. M. Esfahani, and T. Summers, “Learning optimal controllers for linear systems with multiplicative noise via policy gradient,” IEEE TAC, 2020.

8. K. Zhang, B. Hu, and T. Basar, “On the stability and convergence of robust adversarial reinforcement learning: A case study on linear quadratic systems,” NeurIPS, 2020.

9. I. Fatkhullin and B. Polyak, “Optimizing static linear feedback: Gradient method,” arXiv preprint arXiv:2004.09875, 2020.

10. H. Mohammadi, M. Soltanolkotabi, and M. R. Jovanovic, “On the linear convergence of random search for discrete-time LQR,” IEEE Control Systems Letters, 2020.

11. H. Mohammadi, A. Zare, M. Soltanolkotabi, and M. R. Jovanovic. "Convergence and sample complexity of gradient methods for the model-free linear quadratic regulator problem." IEEE TAC, 2021.

12. Zheng, Y., Tang, Y. and Li, N. "Analysis of the optimization landscape of linear quadratic Gaussian (LQG) control," L4DC, 2021

13. Zhang, K., Zhang, X., Hu, B. and Başar, T. "Derivative-free policy optimization for risk-sensitive and robust control design: Implicit regularization and sample complexity."  arXiv, 2021.


Quality: This is a solid paper, although it is unclear how the authors draw the conclusion that their method improves the H-infinity control method. I have a few more questions here.
1) How can the authors prove that gamma keeps increasing to 1 without approaching some limit point in the middle? I guess the proof will require some lower bound on how much progress on gamma can be made at each iteration. Is that true? Any uniform lower bound here?
2) In the nonlinear setting, it seems that based on Lyapunov's indirect method, one can just linearize the dynamics and then stabilize the linear systems. Right? What is the point of developing new theory for "local" stabilization of nonlinear systems?
3) Any evidence supporting the claim that the proposed approach is competitive with established robust control procedures? I mean, I can't find how the authors set up robust control baselines in their numerical study.
4) For simulations, how does the proposed approach work on systems which have modes blowing up to infinity very quickly? In this case, although the theory states that a very small discount factor can be used such that the zero control gain lead to a finite cost, practically the system states in the simulations can blow up quickly to cause numerical issues, right? Can the authors provide an example on this?

Clarity: The paper is well written. The idea is quite clear.

Significance: I think the paper is significant in that it tries to answer a fundamental question with relatively simple ideas. In my opinion, such a solution for stabilization is novel and interesting.

================
Post-rebuttal: I am convinced to increase my score to 7.

**Time Spent Reviewing:**

3 hours

---

> ### Author Response · Authors · 2021-08-06
> **Author Response**
>
>
> Thank you for the constructive comments. We sincerely appreciate you taking the time to compile such a comprehensive list of references that will improve the quality of our work. We look forward to citing these in the updated manuscript.
>
> Many of the contributions of these papers - refining sample complexities, derivative-free methods, robust and distributed control - appear to be somewhat orthogonal to the stabilization question considered in this paper. However, we are excited to incorporate these and to elaborate on the connections between these references and our work. We believe this recent interest in model-free methods for control makes our study of model-free stabilization (as opposed to model-free policy improvement) even more timely.
>
> With regards to your other comments:
>
> 1. Yes, you are absolutely correct. For the case of linear systems, in the proof of Theorem 1, we show that the discount factor can always be increased by a multiplicative constant that is uniformly lower bounded by a term (strictly greater than 1) that depends on the trace of the optimal value function for the undiscounted LQR problem (see L223 as well as pages 14 & 15 in the supplementary material). A similar phenomenon holds for the nonlinear case.
>
> 2. Yes, precisely the argument is based on linearization of the dynamics (see L136:145). We remark quite openly (but will further emphasize in the updated version) that Jacobian linearization + robust control does in fact work to stabilize the system around its equilibrium. However, the focus of our work is to address the mathematical challenges of establishing formal guarantees that policy gradients, with samples drawn from the true nonlinear dynamics (and not its Jacobian linearization), will successfully stabilize the system. We consider the resolution of these technical questions (for model-free control) one of our main contributions.
>
> 3. Thank you for pointing this out, we will specify the precise formulation of the H-infinity control problem in the revised manuscript. In brief, we compared against the optimal controller that minimizes the H-infinity norm of the transfer function from a disturbance (that enters through the input channel) to the performance output (that encodes the LQR cost function). It was not our intention to make a general claim that discount annealing provides better robustness guarantees than H infinity control in general. Only to remark that it has a larger region of attraction than this common control benchmark for the particular case of the nonlinear pendulum. Investigating the extent to which this phenomenon holds more generally is a valuable and important direction for future work.
>
> 4. We agree that there are likely to be numerical issues if one uses finite precision arithmetic to simulate highly unstable systems with modes that blow up to infinity (exponentially fast). We believe these concerns are inherent and fundamental when attempting to stabilize unknown dynamical systems. Importantly, in the nonlinear setting, we propose an interaction model where one can simulate damped system rollouts (not just discounted costs). In the linear setting, this would address the issue of numerical instability.

---

> > ### Comment · Reviewer_YL9s · 2021-08-31
> > **Follow-up**
> >
> > Thanks for addressing my comments. I am convinced that this is an important work and will further increase my score.

---

### Official Review · Reviewer_uRW9 · 2021-07-22

**Rating:** 7
**Confidence:** 3

**Summary:**

This paper proposes a model-free algorithm for stabilizing linear and nonlinear dynamical systems. The algorithm solves a series of discounted LQR problems using policy gradient, and provably converges to a stabilizing controller for LTI systems; for nonlinear systems, the algorithm recovers a stabilizing controller in a small ball around the origin (assuming origin is the equilibrium point). This work overcomes a significant limitation of prior work on policy gradient methods for control, which requires a stabilizing controller a priori. Experiments on a simulated nonlinear system suggest that the algorithm can produce a controller in a few iterations that is stabilizing in a ball of reasonable radius around the origin.


**Limitations And Societal Impact:**

Yes.

**Main Review:**

The paper considers an important problem, since many control methods require a stabilizing controller even when the system is unknown. Moreover, the paper leverages existing techniques but uses new insights to overcome the limitations of said techniques. The theoretical results appear to be correct but I wasn’t able to check all the details.

However, the significance of the result can be strengthened by comparing with other stabilization methods for linear systems with similar access models. For example, “Finite-Time Adaptive Stabilization of Linear Systems” and “Randomized algorithms for data-driven stabilization of stochastic linear systems”  give algorithms for stabilization. How does this paper compare with the techniques used there?

I’m also not completely convinced of model-free methods’ advantages over mode-based methods for linear systems, since linear systems don’t have arbitrary dynamics.

Some limitations of the work:
The setting for linear systems assumes there is no noise, which is restrictive. Can the results be extended to accommodate stochastic noise?
The experimental section could be improved by including other, and possibly more difficult, tasks.

Post rebuttal: I have read the authors' response, and will increase my score.


**Time Spent Reviewing:**

5

---

> ### Author Response · Authors · 2021-08-06
> **Author Response**
>
> Thank you for the insightful comments on our work. We appreciate you bringing these two papers to our attention. We were previously unaware of this important related work and will certainly include them in our updated manuscript.
>
> Relative to our results, which focus on the theoretical foundations of model-free control, these papers focus on developing model-based approaches for stabilizing linear dynamical systems. The algorithms they propose first form system estimates via least squares and then synthesize a stabilizing controller.
>
> More broadly, the goal of our work was not to resolve the question of whether model-based or model-free approaches are superior, but rather to strengthen our theoretical understanding of policy gradient methods and model free procedures in control. Given the increasing popularity of policy gradients amongst practitioners, our aim was to provide the first finite time guarantees for stabilizing dynamical systems using this class of algorithms.
>
> The observation that our model-free method outperforms model-based approaches for nonlinear systems is not a central claim of the paper, but rather a curious finding for the specific case of the nonlinear cartpole. It corroborates the results of Gu et al. 2020 that model-free methods can refine model-based methods by finding local optima (whereas solutions based on Jacobian linearizations are only an approximation thereof). We believe that analyzing the extent to which this phenomenon holds more generally is an important direction for future work.
>
> Regarding noisy dynamics, we chose to focus on the noiseless setting for conceptual simplicity and in order to leverage previous analysis. Since our algorithm is based on a reduction to the convergence of policy gradients for LQR, any extension of these guarantees to LQR with noisy dynamics could be directly incorporated into our analysis to show that discount annealing provably stabilizes dynamical systems with stochastic noise. Thank you for flagging this concern, we will add a more extended, formal discussion regarding extensions to noisy dynamics in the updated manuscript.
>
> Concerning further experiments, we would be happy to apply the method to other systems, especially if the reviewer has a particular notion of “task difficulty” in mind. The cartpole considered in the existing numerical experiments represents a challenging stabilization problem (the dynamics of the linearization are non-minimum phase), however, “difficulty” can be increased by considering systems of larger state dimension. Other challenging control tasks, such as those involving trajectory optimization (e.g. swingup of the cartpole) go beyond the scope of this paper, which is focused on stabilization.

---

> > ### Comment · Reviewer_uRW9 · 2021-09-02
> > **Post Rebuttal**
> >
> > I have read the authors' response, and I will slightly raise my score. My concerns regarding the motivation is addressed, and I hope the authors can incorporate more experiments to illustrate the behavior of the algorithm on different systems.

---

### Decision · Program_Chairs · 2021-09-27

**Decision:**

Accept (Poster)

**Comment:**

All reviewers find an interesting and important idea in this paper. Although some reviewers point out that the similar idea is found in some existing works, the authors clarify the significant difference or their work from those. As a total, this is a nice paper that could appear in NeurIPS. Therefore, my recommendation is acceptance (poster) for this paper.